

Sedimentary mechanisms of a modern banded iron formation on
Milos Island, Greece
[1,2]Ernest Chi Fru*, [3]Stephanos Kilias, [4]Magnus Ivarsson, [1]Jayne E. Rattray, [3]Katerina
Gkika, [2]Iain McDonald, [5]Qian He, [1]Curt Broman
[1]Department of Geological Sciences, 10691, Stockholm University, Sweden.
[2]School of Earth and Ocean Sciences, Cardiff University, Park Place, CF10  3AT,
Cardiff, UK.
[3]Department of Economic Geology and Geochemistry, Faculty of Geology and
Geoenvironment, National and Kapodistrian University of Athens, Panepistimiopolis,
Zographou, 15784, Athens, Greece.
[4]Department of Palaeobiology, Swedish Museum of Natural History, Box 50007,
Stockholm, Sweden.
[5]School of Chemistry, Cardiff University, Park Place, CF10 3AT, Cardiff, UK.
*Corresponding author
Tel: +44(0)29 208 70058
Email: ChiFruE@cardiff.ac.uk
Short title: A modern banded iron formation



**Abstract.** An Early Quaternary shallow submarine hydrothermal iron formation (IF)
in the Cape Vani sedimentary basin (CVSB) on Milos Island, Greece, displays banded
rhythmicity similar to Precambrian banded iron formation (BIF). Sedimentary,
stratigraphic reconstruction, biogeochemical analysis and micro-nanoscale
mineralogical characterization confirms the Milos rocks as modern Precambrian BIF
analogues. Spatial coverage of the BIF-type rocks in relation to the economic grade
Mn ore that brought prominence to the CVSB implicates tectonic activity and
changing redox in the deposition of the BIF-type rocks. Field-wide stratigraphic and
biogeochemical reconstruction demonstrates two temporal and spatially isolated iron
deposits in the CVSB with distinct sedimentological character. Petrographic screening
suggest the previously described photoferrotrophic-like microfossil-rich IF (MFIF),
accumulated on basement andesite in a ~150 m wide basin, in the SW margin of the
basin. A strongly banded non-fossiliferous IF (NFIF) caps the Mn-rich sandstones at
the transition to the renowned Mn-rich formation. Geochemical evidence relates the
origin of the NFIF to periodic submarine volcanism and water column oxidation of
released Fe(II) in conditions apparently predominated by anoxia, similar to the MFIF.
This is manifested in the lack of shale-normalized Ce anomalies. Raman spectroscopy
pairs hematite-rich grains in the NFIF with relics of a carbonaceous material carrying
an average $\delta^{13}C_{org}$ signature of ~-25‰. However, a similar $\delta^{13}C_{org}$ signature in the
MFIF is not directly coupled to hematite by mineralogy. The NFIF, which post dates
large-scale Mn deposition in the CVSB, is composed primarily of amorphous Si
(opal-$SiO_2 \cdot nH_2O$) while crystalline quartz ($SiO_2$) predominates the MFIF. An
intricate interaction between tectonic processes, changing redox, biological activity
and abiotic Si precipitation, formed the unmetamorphosed BIF-type deposits.

**Keywords:** Banded iron formation; BIF analogue; Hydrothermal activity; Iron
cycling; Silica cycling.



## 1 Introduction

Recently, an Early Quaternary iron formation (IF), ~2.0 million years old, displaying
banded rhythmicity typical of Precambrian banded iron formations (BIF) (James,
1954; Gross, 1980; Simonson, 1985, 2003; Bekker et al., 2010), was serendipitously
discovered in the Cape Vani sedimentary basin (CVSB) on Milos Island, Greece (Chi
Fru et al., 2013, 2015). Before this discovery, Cape Vani was long known to host Mn
oxide ores of economic potential (Hein et al., 2000; Liakopoulos et al., 2001; Glasby
et al., 2005; Kilias et al., 2007). Milos is an emergent volcano on the Hellenic
Volcanic Arc (HVA) where arc-volcanism and seafloor hydrothermal activity occur
in thinned pre-Alpine to Quaternary continental crust (Kilias et al., 2013) (Fig. 1). The
first reported IF from Cape Vani is unmetamorphosed and contains diverse
microfossils encrusted by hematite, with ferrihydrite proposed as a primary precursor
mineral (Chi Fru et al., 2013, 2015). Field stratigraphy, Rare Earth Elements (REEs),
stable isotopes, petrographic and microfossil studies point to microbial Fe deposition
in a semi-enclosed, shallow submarine basin under conditions analogous to those that
formed the Precambrian Algoma-type BIFs near volcanic centers (Chi Fru et al.,
2015). These earlier reports assumed a one-time basin-wide depositional event and a
common origin for all Fe-rich sedimentary rocks in the CVSB.
However, it remains unclear what sedimentary processes caused the distinct
deposition of the BIF-type rocks in a basin where Mn precipitation was apparently
widespread at various intervals. Moreover, it is not known how the Mn ores relate
temporally and spatially to Fe deposition in the ~1 km long CVSB. This knowledge
may provide clues to processes that triggered large-scale deposition of similar
Proterozoic Fe-Mn-rich deposits (Roy, 2006; Tsikos et al., 2010). Here, new
sedimentological, petrological and biogeochemical evidence describes cycles of





periodic precipitation of shallow submarine Si and Fe-rich sedimentary rocks and the
plausible mechanisms that enabled their temporal and spatial separation from the Mn
deposits in the CVSB. The data reveal a much more complex depositional system not
only controlled by microbial Fe(II) oxidation as previously proposed (Chi Fru et al.,
2013, 2015), but illuminates episodic volcanism coupled to changing redox conditions
as a central mechanism in the formation of the banded iron rocks.

## 93    1.1 Geological setting

The geology and, iron and manganese mineralization of the CVSB have been
described in detail (Plimer, 2000; Hein et al., 2000; Liakopoulos et al., 2001;
Skarpelis and Koutles, 2004; Glasby et al., 2005; Stewart and McPhie, 2006; Kilias,
2011; Alfieris et al., 2013: Chi Fru et al., 2013, 2015; Papavassiliou et al., 2017).
Briefly, the Milos IF is part of the CVSB, a recently emergent sedimentary rift basin
located NW of Milos Island, along the HVA in the Aegean Sea, Greece (Fig. 1). It
hosts a fossil analogue of active shallow-submarine hydrothermal activity on the coast
of Milos Island (Dando et al., 1995). The CVSB developed within a 2.7 to 1.8 Ma
shallow-submarine rhyolitic-dacitic volcanic center, filled up mainly by a ~35-50 m
thick stratigraphic succession of volcaniclastic/epiclastic sandstones and sandy tuffs
spanning Upper Pliocene to Lower Pleistocene, 35-40% of which is hydrothermally
mineralized by Mn oxides and barite (Hein et al., 2000; Liakopoulos et al., 2001;
Skarpelis and Koutles, 2004; Papavassiliou et al., 2017). Sedimentologic and
ichnologic data, including sedimentary structures, lamellibranch, echinoid and
brachiopod fossils, the gastropod mollusk fossil, Haustator biplicatus (Bronn, 1831),
and microbially induced sedimentary structures (e.g., Kilias, 2011), suggest that most
of the CVSB sandstones/sandy tuffs hosting the Mn-rich deposit, are foreshore to



shoreface shallow submarine deposits, formed at a maximum depth of 200 mbsl. Over
the last 0.8 Myr, fluctuating water depths due to sea-level changes of up to 120 m and
volcanic edifice building, has resulted in tectonic uplift of ~250 m (Papanikolaou et
al., 1990). The CVSB fill, currently 35 m above sea level, is tectonically northbound
by intrusive rhyolite, framed by elevated andesitic-dacitic centres, with the Cape Vani
and the Katsimoutis dacitic lava domes being the most prominent (Fig. 1).

**2 Methodology**
**2.1 Sample preparation**
Sedimentary structures, grain-size trends, lateral facies variations, vertical stacking
trends, and key stratigraphic surfaces form the basis for facies analysis. Prior to
mineralogical and geochemical analysis, exposed rock surface layers were sawn and
removed.    GeoTech Labs (Vancouver, Canada) produced doubly polished thin
sections for mineralogical and textural analysis, while geochemical analysis was
performed on pulverized powders and acid-digests (Chi Fru et al., 2013, 2015).

**2.2 Mineralogical analysis**
**2.2.1 X-Ray Diffraction analysis**
A PANalytical Xpert-pro diffractometer at room temperature, 45 kV, 40 mA and
1.5406 Å wavelength and Cu-Kα radiation and Ni-filter, was used for Powder X-Ray
Diffraction (PXRD) analysis. Samples were analyzed between 5-80° in step sizes of
0.017° with continuous mode scanning step time of 50.1650 s while rotating. Raman
spectroscopy was performed with a confocal laser Raman spectrometer (Horiba
instrument LabRAM HR 800), equipped with a multichannel air-cooled (-70°C) 1024
x 256 pixel charge-coupled device (CCD) array detector as previously described (Chi



Fru et al. 2013, 2015). Spectral resolution was ~0.3 cm$^{-1}$/pixel.   Accuracy was
determined by a repeated silicon wafer calibration standard at a characteristic Raman
line of 520.7 cm$^{-1}$.

**2.2.2 Transmission electron microscopy**
Specimens for Transmission electron microscopy (TEM) were prepared from the
crushed rock specimen powder. This was followed by dry-dispersal onto a 300 mesh
holey carbon TEM Cu grid. Microscopy was conducted using a JEOL 2100 TEM
with a $LaB_6$ source in the School of Chemistry, Cardiff University, operated at
200kV. The X-EDS analysis was performed with an Oxford Instrument SDD detector
X-Max$^N$ 80 T.

**2.2.3 Scanning electron microscopy**
Scanning Electron Microscopy-Energy Dispersive Spectroscopy (SEM-EDS) analysis
was done on a FEI QUANTA FEG 650 ESEM. Images were captured at 5 kV and
EDS data collected at 20 kV, using an Oxford T-Max 80 detector (Oxford
Instruments, UK). The analyses were performed in low vacuum to minimize surface
charging of uncoated samples. EDS elemental maps were collected for 30 min or until
the signal had stabilized, indicated by a clear distribution trend. The data were further
processed with the Oxford Aztec software.

**2.3 Geochemical analysis**
**2.3.1 Laser ablation ICP-MS and trace element analysis**
Laser Ablation-Inductively Coupled Plasma-Mass Spectrometry (LA-ICP-MS) was
performed at Cardiff University on polished thin sections. The LA-ICP-MS system



comprised a New Wave Research UP213 laser system coupled to a Thermo X Series
2 ICP-MS. The laser was operated using a frequency of 10 Hz at pulse energy of
~5mJ for an 80μm diameter beam using lines drawn perpendicular to the layering and
at a movement speed of 26 microns sec$^{-1}$. Samples were analyzed in time resolved
analysis (TRA) mode using acquisition times of between 110 and 250 seconds;
comprising a 20 second gas blank, 80-220 second ablation and 10 second washout.
Dwell times varied from 2 msec for major elements to 35 msec for low abundance
trace elements. Blank subtraction was carried out using the Thermo Plasmalab
software before time resolved data were exported to Excel.
Separated and independently pulverized banded layers were digested by lithium
borate fusion followed by major, trace and rare earth element (REE) analyses using
ICP-ES/MS and XRF at AcmeLabs® (http://acmelab.com). Geochemical data were
compared with previously published results for the more widely investigated Mn
deposits (Hein et al., 2000; Liakopoulos et al., 2001; Glasby et al., 2005).

**2.3.2 Isotope analysis**
C, N and S isotopic composition for the pulverized samples was determined as
previously described (Chi Fru et al., 2013, 2015), following combustion in a Carlo
Erba NC2500 analyzer and analyzed in a Finnigan MAT Delta V mass spectrometer,
via a split interface to reduce gas volume. Reproducibility was calculated to be better
than 0.15‰ for $\delta^{13}C$ and $\delta^{15}N$ and 0.2‰ for $\delta^{34}S$. Total C and N concentrations were
determined simultaneously when measuring the isotope ratios. The relative error was
<1% for both measurements. For carbon isotopic composition of organic carbon,
samples were pre-treated with concentrated $HNO_3$ prior to analysis.



### 2.4 Organic geochemistry analysis


Lipid biomarker and compound specific $\delta^{13}C$ analyses were executed on powdered
samples of sectioned bands from which exposed surface layers had been removed.
Modern sediments from Spathi Bay, 36°40'N, 24°31'E, southeast of Milos Island,
collected by push coring at 12.5 m below the seafloor were freeze-dried prior to
extraction to aid the identification of potential syngenetic biomarkers in the
Quaternary rocks. Between 4-6 g of ground samples were ultrasonically extracted
using 3×Methanol, 3×(1:1) Methanol:Dichloromethane (DCM), and 3×DCM and
extracts were combined and dried under $N_2$. Samples were subsequently re-dissolved
in DCM then methylated following the method of Ichihara and Fukubayashi (2010).
The resulting residue was silylated using, 20 μl pyridine and 20 μl BSTFA and heated
at 60°C for 15 min. Total lipid extracts were analyzed using a Shimadzu QP 2010
Ultra gas chromatography mass spectrometer (GC/MS). Separation was performed on
a Zebron ZB-5HT column (30 m x 0.25 mm x 0.10 μm) with a helium carrier gas
flow at 1.5 ml min$^{-1}$. Samples were injected splitless, onto the column at 40°C with
the subsequent oven temperature program ramped to 180°C at a rate of 15°C min$^{-1}$,
followed by ramping to 325°C at a rate of 4°C min$^{-1}$ and a final hold for 15 min. The
MS was set to scan from 50 to 800 m/z with an event time of 0.70 sec and a scan
speed of 1111 u/sec. All peaks were background subtracted and identification
confirmed using the NIST GC/MS library and literature spectra. Contamination was
not introduced into the samples, as blank samples worked up concurrently with the
rock fractions had results comparable to the ethyl acetate instrument blank.

### 2.5 Chemical weathering analysis




Chemical index of alternation (CIA) was used to determine whether variations in
chemical weathering intensities would in addition to hydrothermal activity deliver
materials into the depositional basin from the continent, according to the
formula:   $CIA = Al_2O_3/(Al_2O_3 + CaO + Na_2O + K_2O) \times 100$   .   Extensively
applied, the CIA index reveals subtle changes in weathering fluxes (Nesbit and
Young, 1982; Maynard, 1993; Bahlburg & Dobrzinski, 2011), where increasing CIA
values generally indicate amplified chemical dissolution of rocks and selective release
of dissolvable CaO, $Na_2O$ and $K_2O$ into solution (Nesbit & Young, 1982; Maynard,
1993; Bahlburg & Dobrzinski, 2011). The broken rock particles enriched in the
poorly soluble $Al_2O_3$ fraction, settle to the seafloor as weathered sediments carrying a
chemical composition different from the source. In the absence of chemical
dissolution, no net chemical change is expected in the composition of sediments
compared to source and thus a low CIA index.  CIA indices for detritus of 0-55, 55-75
and >75, are considered unweathered, unweathered to slightly weathered and
weathered to highly weathered, respectively (Nesbit & Young, 1982; Maynard, 1993;
Bahlburg & Dobrzinski, 2011). The redox conditions under which sediments formed
were obtained from REE composition normalized to the North American Shale
Standard (NASC) (Groment et al., 1984).

**3 Results**
**3.1 Lithostratigraphy**
Field-wide sedimentological and lithostratigraphical mapping of the CVSB in the
summer and fall of 2014, enabled the assessment of the lateral and vertical coverage
of the Milos iron oxide-rich facies relative to the Mn-rich sandstones that dominate
the Early Quaternary sedimentary basin (Fig. 2). Six stratigraphic sections,





representing marine siliciclastic lithofacies sequences, were investigated along a ~1
km SW-NE trending portion of the CVSB infill (Supplementary Figs 1-7). Sequence
stratigraphy was conducted on outcrops and vertical shafts and tunnels left behind by
extinct Mn mining activity. Two of those sections; Section A located at
36°44'17.85''N, 24°21'17.72''E and Section B located at 36°44'35.11''N,
24°21'11.25''E, contain stratigraphic units composed of layered, bedded, or
laminated rocks that contain ≥15 % Fe, in which the Fe minerals are commonly
interlayered with quartz or chert, in agreement with the definition of Precambrian
BIFs (James, 1954; Gross, 1980; Bekker et al., 2010). These IFs are descriptively
referred to here as microfossiliferous iron formation (MFIF) according to Chi Fru et
al. (2013, 2015), and non-microfossiliferous iron formation (NFIF) (this study),
respectively (Fig. 2). The MFIF and the NFIF occupy at most ~20% of the entire
CVSB infill.   The stratigraphy and sedimentary lithofacies are illustrated below,
using lithofacies codes modified after Bouma (1962), Miall (1978, 1985), Lowe
(1982), Mutti (1992) and Shanmugam (2016).
Further field stratigraphic survey revealed considerable lithologic variability
within three fault-bounded volcanosedimentary sub-basins in the CVSB (Fig. 2),
which for the sake of simplicity are referred to as Basin 1—host of the MFIF; Basin
2—host of economic grade Mn ore; and Basin 3—host of the NFIF (Fig. 2).  Each
section is framed by distinct marginal normal faults that strike in the NW-SE and NE-
SW to NNE-SSW directions, distinguishable by distinct lateral sedimentary facies
exhibiting unique vertical sequence stratigraphy (Fig. 2; Supplementary Figs 1-7).
Faulting in the CVSB is related to major geographical activation of extensional
structures at intervals that shaped Milos into a complex mosaic of neotectonic units
(Papanikolaou et al., 1990).




### 3.1.1 Section A (36°44'17.85''N, 24°21'17.72''E)

Informally known as "Little Vani", Section A is the type section containing the MFIF
at the base. It crops out in the W-SW edge of the CVSB (Figs 1 & 2) as a ~6-7 m high
cliff resting stratigraphically on submarine dacitic and andesitic lavas and domes.
This section extends laterally in the N-NE direction for an estimated 300–500 m.

The MFIF is correlatively interpreted to be in direct stratigraphic contact with
Late Pliocene-Early Pleistocene (2.5–1.5 Ma) basement submarine dacitic-andesitic
rocks. Lithologically, the MFIF comprises laminated and massive fine-grained red
and white weathered ferruginous jaspelitic red chert layers (Chi Fru et al., 2013,
2015). The chert layers contain morphologically distinct Fe minerals dispersed in a
fine-grained siliceous matrix (Fig. 3), marked by the notable absence of pyrite and an
extremely low S content (Chi Fru et al., 2013, 2015). Layers are tabular and typically
laterally continuous at scales of several meters, whereas wave and current structures
(e.g., cross-lamination), are generally absent from the MFIF. The hematite-rich MFIF
laminae (Table 1) are built by massive encrustation of anoxygenic photoferrotrophic-
like microbial biofilms by precipitated Fe (Chi Fru et al., 2013). The base of the MFIF
outcrop, is visibly mineralized by black diffused bands/veins composed of Mn oxides
(Fig. 4 & Table 1).

A markedly reddish 2-3 m-thick section immediately overlies the MFIF,
comprising a distinct package of Fe-rich beds that transition up the section from fine
to reddish medium and coarse-grained to pebble–cobble conglomerate volcaniclastic
sandstone beds (Figs 4A & 5). The lower 1-2 m consist of fine-grained sandstone
beds that are well to moderately sorted, containing a 20-40 cm thick portion
dominated by plane parallel-laminated sandstone/sandy tuff, massive to plane



parallel-laminated sandstone/sandy tuff, and massive sandstone/sandy tuff lithofacies
(Fig. 5; Supplementary Fig. 1). The fabric of these Fe-rich sandstone  facies consists
of sub-angular to sub-rounded and 100–600 μm fine to medium-grained volcaniclastic
K-feldspar grains, making up to 75% of the total rock, with variable amounts of
quartz and clay mineral grains.

The latter are overlain  by a ~1-1.5 m  sequence of poorly-sorted tabular clast-

supported pebble-to-cobble conglomerate beds with an erosional base, grading
upward  into coarse to medium-grained Sh beds, arranged in alternating conglomerate
cycles (Fig. 5), averaging 20-40 cm in thickness.  The cobble/pebble conglomerate
clasts include intraformational volcanic rocks (dacite, andesite), allochthonous
volcaniclastic sandstone, and volcaniclastic microclasts (e.g. K-feldspar), cemented
by hematite (Fig. 5; Chi Fru et al., 2013; Kilias et al., 2013). Towards the
westernmost edge of the "Little Vani" section, there is a facies change from the
graded Gcm/Sh rhythms to a predominantly Fe-rich conglomerate Gcm bed (Fig. 4A),
termed the conglomerate-hosted IF (CIF) in Chi Fru et al. (2015), with a maximum
thickness of ~0.5 m and a cobble size range of ~10 cm. The Fe-rich conglomerate bed
transitions upward into medium-grained pebbly reddish ferruginous Sm with thin
volcanic rock and sandstone pebble lenses. This in turn grades upwards into a very-
fine-grained greenish glauconite-bearing plane parallel-laminated sandstone to
siltstone bed; characterized by soft-sediment deformation structures, such as flame
structures, convolute bedding and lamination structures, loop bedding, load casts, and
pseudonodules (Supplementary Figs 1-2).

The "Little Vani" section is eventually capped along an erosional surface by

an overlying 1-2 m thick section dominated by medium to fine-grained and
moderately to poorly-sorted reddish Fe-rich tabular sandstone beds, 10–40 cm thick,



topped by patchy sub-cm to cm-thick Mn-rich sandstones (Fig. 5; Supplementary Figs
1-2). Dominant lithofacies of the Fe-rich sandstone cap include planar and hummocky
cross-bedding, exhibiting bioturbation in places. The Fe-rich lithofacies cap is
laterally discontinuous, thinning out basinwards towards the N-NE, and can be
observed smoothly grading into a 1-2 m thick section composed of cm to sub-cm-
thick Mn–rich volcaniclastic sandstone lithofacies, described below in Section B. No
Fe-rich hydrothermal feeder veins are obvious in the MFIF, however feeder veins and
Mn horizons can be observed to truncate laminations in the MFIF, and up through the
whole "Little Vani" section (e.g., Figs 4C & 5).

**3.1.2 Interpretation of Section A**
The MFIF rests directly on the submarine dacites-andesites that were deposited in
relatively shallow but dominantly below a wave base submarine setting (Stewart and
McPhie, 2006). The fine-grained, finely laminated nature of the MFIF, and, the lack
of evidence of current or wave structures (e.g., symmetric ripples or hummocky
cross-stratification), coupled to the absence of volcanogenic detrital particles and
intraclast breccia structures, indicate a low energy sedimentation environment at ca. $\geq$
100-200 m depth, marked by negligible volcanic interference (e.g., Tice and Lowe,
2006; Trower and Lowe, 2016; Konhauser et al., 2017). This interpretation is
supported by the observed enrichment of Fe in the MFIF; a characteristic of relatively
deeper water lithofacies (Trower and Lowe, 2016; Konhauser et al., 2017). This view
is compartible with the proposition that hematite enrichment in the MFIF was under
the control of photoferrotrophic biofilms (Chi Fru et al., 2013) known to thrive at
lower light intensities (Kappler et al., 2005; Li et al., 2013; Konhauser et al., 2017).
The quiet environmental conditions would have ensured the formation of such stable



photoferrotrophic biofilms over extended periods of time that would have facilitated
the oxidation of hydrothermally released Fe(II) and the depositon of Fe(III) minerals.
The overlying lithofacies sequence record a switch to faster accumulation of
volcaniclastic turbidites on the quiet MFIF deposit, with the fine, medium to coarse-
grained sandstone lithofacies typifying deposition during low and high density
turbiditic flows in the middle to inner parts of a turbidite fan-like environment (Lowe,
1982; Mutti 1992; Talling et al., 2012; Orme and Laskowski, 2016; Shanmugam,
2016; Wang et al., 2017). Massive conglomerates containing both allochthonous
sandstone clasts and intraformational andesite-dacite are interpreted as channelized
submarine debris flows or slump deposits sourced from adjacent topographic highs
(Lowe, 1982; Stewart and McPhie, 2006; Orme and Laskowski, 2016). Also,
deposition from a waning low density turbidity current is indicated by the upward
fining bed of pebbly Fe-rich sandstone, greenish glauconite bearing sandstone and
laminated siltstone.  Up section, the abundance of parallel and cross stratified Fe-rich
and Mn-rich sandstone facies along an erosional surface, reflect a change in
deposition to a high energy, shallow submarine shoreface/foreshore setting, above a
wave base.
In summary, stratigraphic observations in the "Little Vani" section  indicate that
the MFIF constitutes the older IF deposited on the CVSB basement lavas. Following
MFIF deposition, there have been a series of upward lithologic changes which reflect
gradual shoaling, accompanied by tectonic instability, topographic growth, submarine
erosion,  and massive sediment supply by density/gravity flows of volcanogenic
debris. It is proposed that these changes were controlled by a combination of
submarine volcano-constructional processes, synvolcanic rifting and volcano-tectonic
uplift, resulting in cyclic changes in depositional water depth (Stewart and McPhie,



2006; Papanikolaou et al., 1990; Steele et al., 2000; Trower and Lowe, 2016; Wang et
al., 2017).

**3.2 Section B (36°44'35.11''N, 24°21'11.25''E)**
This ~8-10 m-thick fault-bounded stratigraphic section, here referred to as "Magnus
Hill", is the type section that contains the NFIF (Figs 2 & 7; Supplementary Figs 3-4).
Two lithostratigraphic units—a lower unit A and an upper unit B—are identified in
this study. Unit A is made up of a lower sandstone facies that is ~4-5 m thick,
dominated by a Mn-oxide cement exhibiting a grayish to black, coarse to very coarse-
grained volcaniclastic sandstone beds, overlain by reddish brown Fe-rich massive
sandstone beds (Fig. 8 & Supplementary Figs 3-4). Unit B, ~5 m thick,
unconformambly overlies unit A and comprises two distinct packages of beds that
transition up section from brownish gravel-to-pebble conglomerate beds (0.5-1.0 m
thick), in contact with the very fine-grained NFIF deposit (Supplementary Fig. 8 & 9).
The NFIF is capped by patchy cm-thick crustiform Mn oxides. Bifurcating feeder
veins composed of barite, quartz and Mn and Fe-oxide minerals cut through the
underlying sandstone beds (Supplementary Fig. 4).

Sandstone beds are moderately to well-sorted and 5-15 cm thick, and Mn-

mineralized lithofacies include plane parallel-laminated sandstone, plane parallel
laminated to rippled sandstone, planar cross-bedded sandstone, and massive
sandstone. Secondary lithofacies include thinly bedded (1-5 cm thick) greenish
glauconite-bearing heterolithic sandstone and thin (< 5 cm thick) white to pale-brown
sandy tuff beds interbedded with the other Lithofacies. The sandstone facies host the
main economic grade Mn oxide ores in the CVSB, which typically construct
texturally diverse cements associated with a variety of volcaniclastic detritus (i.e., K-



feldspar, lithic fragments, altered volcanic glass, quartz, sericitized plagioclase,
chloritized biotite) and authigenic barite and or glauconite. This constitutes part of a
separate study devoted to the Mn ores and will not be dealt with further here, as the
focus of the current study is on the IFs. Kilias (2011), however, suggested that many
of the sedimentary structures identified within the Mn–mineralized sandstone
lithofacies are associated with microbial mat growth.

The NFIF is composed of strongly banded Fe-rich rocks (Fig. 7) exposed on

the topmost part of "Magnus Hill". About 2-3 m thick, the NFIF consists of mm to
sub-mm thick, dark grey and brown Fe-rich bands, interbanded with reddish brown
Si-rich layers (Figs 7 & 9-11; Supplementary Figs 10-11).   Sedimentary structures in
the NFIF are predominantly characterized by rhythmic mm to sub-mm thick bedding
(e.g., Fig. 7). The iron oxide-rich bands made up mainly of hematite (Table 1 & Fig.
10C) are typically composed of very fine-grained angular to sub-angular volcanic dust
material (i.e., fine volcanic ash with particle size under 0.063 mm, K-feldspar,
tridymite and cristobalite (Table 1) in an amorphous Si and crystalline hematite
matrix (Fig. 12). The predominantly amorphous Si-rich bands are typically planer,
finely laminated and composed of microcrystalline to cryptocrystalline ferruginous
chert.

The NFIF is directly overlain by a  ~1 m thick laminated to massive well-

indurated, nodular-pisolitic ironstone bed (Fig. 8A, C & D) that locally preserves a
sub-horizontal fabric reflecting the bedding in the original sediment or contain various
ferruginous clasts such as fragments, nodules, pisoliths, and ooliths set in a hematite-
rich siliceous matrix (Fig. 8C). Scattered cm-scale pisoliths display a crude concentric
internal layering, characterized by open and vermiform voids filled by cauliflower-
like Mn oxides overprint (Fig. 8D).




### 3.2.1 Interpretation of Section B

We interpret the ferruginous NFIF lithofacies to represent the deepest water deposits
in the "Magnus Hill" section based on its very fine-grained sedimentary composition,
fine laminations and a paucity of intraclast breccias (e.g., Trower and Lowe, 2016,
and references therein). These combined with the lack of evidence for wave and
current-formed sedimentary structures (e.g., hummocky cross-stratification, trough,
ripple cross-stratification, and erosional contacts), indicate quiet water low energy
sedimentation, below a likely fair-weather wave base (Simonson and Hassler, 1996;
Trendall, 2002; Krapež et al., 2003; Trower and Lowe, 2016; Konhauser et al., 2017).
This interpretation is consistent with (1) up section lithofacies change from
predominantly sandstone facies of the lower unit to conglomerate facies (Fig. 8B),
probably related to a series of channel deposits in an inner-turbidite fan-like setting
(Orme and Laskowski, 2016). This sedimentary sequence shows overall deepening
from a tidal to shoreface zone depositional environment to an offshore zone during
periods of high sea level stand (Trower and Lowe, 2016); (2) conclusions of previous
workers suggest that lithofacies with Fe-rich composition similar to the NFIF, were
deposited from seawater in a basinal settings (Lowe and Byerly, 1999; Tice and
Lowe, 2006). The hypothesized deepening of the "Magnus Hill" section is generally
consistent with the interpretation that active rifting was occurring during the filling of
the CVSB (Papanikolaou et al., 1990; Stewart and McPhie, 2006; Liakopoulos et al.,
2001; Papavassileiou et al., 2017), resulting in the transition from a relatively shallow
and deeper water setting represented by the sandstone and conglomeratic deposits, to
a relatively deeper quiet water environment, characterized by the finely laminated
NFIF facies (Trower and Lowe, 2016).



Sedimentary structures and microbial mat fabrics (Kilias et al., 2011) in
lithostratigraphic unit A are interpreted to record a variation between storm-
dominated shallow-marine (lower shoreface), stable shallow-marine environment
with low sedimentation rate in an upper to middle shoreface, and tide-influenced
environments (e.g. Noffke et al., 2003; Ramos et al. 2006; Kilias, 2011; Ossa et al.,

2016).

We interpret that each graded Fe oxide-rich band of the NFIF (Supplementary
Figs 8 & 9), represents an individual fallout deposit from a proximal pyroclastic
eruption (Stiegler et al., 2011; Trower and Lowe, 2016). This interpretation is
supported by normal grading in fine volcanic ash content that reflects their likely
origin as pyroclastic fallout deposits in an otherwise quiet water setting (Lowe, 1999).
For example, tridymite is a stable $SiO_2$ polymorph formed at low pressures of up to
0.4 GPa and at temperatures of ~870-1470 $^{\circ}C$ (Swamy et al., 1994; Koike et al., 2013;
Morris et al., 2016). The coincidence of trimydite formation with silicic volcanism is
in agreement with the widespread distribution of andesite, dacitite and rhyolitic lava
domes in the CVSB. For example, vapour phase production of tridymite together with
sanidine identified in this study (Fig. 10) and iron oxides is principally associated
with rhyolite ash flow (Breitkreuz, 2013; Galan et al., 2013). Similarly, Cristobalite is
a $SiO_2$ polymorph associated with high temperature rhyolitic eruptions (Horwell et al.,
2010).  Finally, in situ carbonaceous laminations are absent, suggesting that benthic
microbial mat growth had no influence on deposition of the NFIF (Trower and Lowe,
2016). Ironstones overlying the NFIF are difficult to interpret with the existing data,
but may represent primary granular iron formations (GIF); i.e., a facies transition
from BIF-style to GIF-style IF (e.g., Bekker et al., 2010), or supergene ferruginous
duricrust formation resulting from subaerial weathering (Anand et al., 2002).






**3.3 Geochemistry**


The SEM-EDS-electron micrographs of the NFIF thin sections reveal distinct Fe and
Si-rich layers alternating periodically with each other in a fine sediment matrix as
shown by the grain size (Figs 9 & 11 & Supplementary Figs 9-11). Laser ablation
ICP-MS line analysis indicates Si and Fe count intensities in the Milos BIF-type are
comparable to the 2.5 Ga Precambrian BIF reference from the Kuruman IF formation,
Transvaal Supergroup, South Africa (Fig. 11). The laser ablation ICP-MS data further
show that dramatic fluctuations in Fe concentrations control the Si to Fe ratio in both
types of rocks, despite the thousands of millions of years gap between them.
No other Fe(III)(oxyhydr)oxide minerals have been identified in the Cape Vani
Fe-rich facies different from hematite. Electron imaging of the NFIF Fe-rich bands
suggests Si, Al and K-rich phases are mostly associated with the volcaniclastic
material predominated by K-feldspar clasts (Fig. 9; Supplementary Figs 10 & 11). A
unique feature of the NFIF is that the hematite in the Fe-rich bands occurs in tight
association with a carbonaceous material (Fig. 10C), but not for the hematite in the
Fe-rich sandstones and in the MFIF. This is also the case for the CIF overlying the
MFIF. Hematite showing a fluffy texture and at times presenting as framboidal
particles, is sprinkled in the Si-rich cement containing traces of Al and K in the MFIF
rocks (Fig. 3). Lack of association of the framboidal-iron-rich particles with S,
following SEM-EDS analysis, rules out a pyrite affiliation. This is consistent with the
non-sulfidic conditions proposed for the deposition of the Milos BIF-type rocks, as
are their Precambrian predecessors. TEM analysis suggests platy nano-Fe oxide-rich
particles predominate in the NFIF and MFIF, confirmed by overlaid X-ray Energy
Dispersive spectra taken from selected areas (Fig. 12) and consistent with the XRD





data showing hematite in both samples. The platy hematite needles in the Milos BIF-
type rocks are morphologically, and by size, comparable to hematite needles reported
in the ~2.5 Ga Kuruman BIFs (Sun et al., 2015).

Unlike the iron-rich bands, volcaniclasts in the Si-rich bands are much smaller

in size, occurring mainly as fine-grained (Supplementary Fig. 8-11), signifying
predominant precipitation during periods of weaken hydrothermal activity. The $SiO_2$
matrix in both the MFIF are fine-grained, occurring mainly as amorphous opal in the
NFIF and crystalline quartz in the MFIF (Figs 10B & 12A-B), while in the MFIF it is
mainly present as crystalline quartz (Fig. 12C-D). Relative concentrations of Al, K
and Ti in the samples are generally low, with bulk-measured concentrations in both
the Si-/Fe-rich bands, together with the $SiO_2$ and $Fe_2O_3$ content, strongly covarying
with continental crust concentrations (Fig. 13A). Mn impregnation of the MFIF,
preserved in the form of replacement layers mostly identified as cryptomelane
[$K(Mn^{4+},Mn^{2+})_8O_{16}$) (Table 1), is below detection in the NFIF. Rare hausmannite
($Mn^{2+}Mn^{3+}_2O_4$) was detected in a few cases in the MFIF (Fig. 10D).

Trends of major elements from which CIA indices were calculated (Fig. 13B),

covary with those of the continental crust (Fig. 13A). Continental crust averages, refer
to the zone from the upper continental crust to the boundary with the mantle (Rudnick
& Gao, 2003). The calculated CIA indices average 52 with one outlier at 22 (Fig.
13B). No distinct relationship could be established between the CIA indices and the
respective IFs or between the distinct alternating Si- and Fe-rich bands (Fig. 13).
Highly weathered clay minerals resulting from the chemical decomposition of
volcanic rocks, e.g., kaolinite representing maximum CIA values of 100 or 75-90 for
illite, are absent in the analyzed materials. The absence of carbonates in the rocks
strengthened the CIA indices, since CIA indices are expected to be lower when Ca

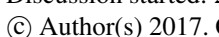

carbonates are present (Bahlburg and Dobrzinski, 2011). $TiO_2$ content—a detrital
proxy—is mostly constant and covaries with the CIA values (Fig. 13B), suggesting
little variability and limited continental weathering input. A fairly strong negative
linear correlation was found between $SiO_2$ and $Fe_2O_3$ values normalized to $TiO_2$
(inset, Fig. 13B).

Shale-normalized REE values ($REE_{(SN)}$) for both the MFIF and NFIF are

consistent with previous reports (Chi Fru et al., 2013, 2015), and show patterns
typical of marine sedimentary environments affected by hydrothermal activity
throughout Earth's history (e.g., Planavsky et al., 2010). There is a notable absence of
significant negative $Ce_{(SN)}$ anomaly for both the MFIF and NFIF (Fig. 14A-B). These
observations are statistically corroborated by true Ce anomalies, calculated as Ce/Ce*
($Ce_{(SN)}/0.5Pr_{(SN)} + 0.5La_{(SN)}$) and Pr/Pr* ($Pr_{(SN)}/0.5Ce_{(SN)} + 0.5Nd_{(SN)}$) and considered
significant when Ce/Ce* and Pr/Pr*are less than and greater than 1, respectively (Bau
et al., 1996; Planavsky et al., 2010) (Fig. 14B). Further, the Eu/Eu* anomalies
averages for the MFIF and NFIF and the distinct Fe-/Si-rich bands, suggest a ~2×
higher Eu/Eu* signal for the Si-rich bands relative to the Fe-rich bands and between
the MFIF and NFIF deposits (Fig. 14C). Average Pr and Yb shale normalized ratios
(Pr/Yb*)—a light vs. heavy REE enrichment proxy (Planavsky et al., 2010)—indicate
similar enrichment levels of light and heavy REE in both the NFIF and MFIF, as well
as in the Fe- and Si-rich bands (Fig. 14C).

**531    3.4 Lipid biomarker distribution and chemotaxonomy**

Bulk $\delta^{13}C_{org}$ averaged −25.4‰ (SD:±0.22), −25.2‰ (±0.26) NFIF Fe-/Si-rich bands
and −25.6‰ (SD:±0.12) for bulk MFIF, respectively (Table 2). A fractionation effect
between the alternating Fe-/Si-rich layers ($\Delta^{13}C_{\text{Fe-rich NFIF-Si-rich NFIF}}$) is estimated to be





~0.23‰ (SD±0.036), while $\Delta^{13}C_{\text{Fe-rich NFIF-MFIF}}$ and $\Delta^{13}C_{\text{Si-rich NFIF-bulk MFIF}}$, is 0.13‰
(SD:±0.11) and 0.36‰ (SD:±0.14), respectively. These differences are small and
within the margin of error of analysis, suggesting no strong distinction in $\delta^{13}C_{org}$
preserved in the different IFs and their various facies. They are interpreted to mean
similar carbon fixation processes operated during intervals of predominant Si and
Fe(III)(oxyhydr)oxides deposition in both IFs. Attempts to discriminate between these
environments by lipid biomarker analysis revealed mainly $C_{16}$-$C_{19}$ fatty acid methyl
esters (FAME) in the Fe-rich NFIF bands and in bulk MFIF, while the Si-rich NFIF
bands contain mainly $C_{12}$-$C_{21}$ FAMEs, suggesting either selective preservation (lipid
recovery was lower in the Fe-rich MFIF bands) or shifts to different potential
biological populations during the deposition of the different layers.  Preserved lipids
discriminate against typical microbial lipid biomarkers like hopanoids, while C3 plant
FAME are detected in all studied materials (Fig. 15). The anaerobic bacteria indicator,
$10MeC_{16:0}$ FAME, was however identified in a few bands.

**4 Discussion**
**4.1 Sedimentological processes**
The three sub-basin interpretation of the CVSB is consistent with previous proposals
suggesting that sedimentation within the CVSB was characterized by active
synvolcanic rifting which must have been important in shaping basin topography and
the creation of sub-basin architecture (Papanikolaou et al., 1990; Stewart and McPhie,
2006; Liakopoulos et al., 2001; Papavassiliou et al., 2017).  Moreover, this tectonic
regime would suggest that the location(s) of volcanism were continually changing
relative to the two stratigraphic sections, which themselves were also being affected,
i.e. changes in depositional water depth and sedimentation style or and/or that local



submarine or subaerial topographic highs impeded the lateral continuity of
sedimentary units (Stewart and McPhie, 2006; Trower and Lowe, 2016). Chi Fru et al.
(2015) have suggested there is an upward deepening of the overall depositional
setting recorded in the "Little Vani" section, consistent with rifting during CVSB
infilling time.
The CVSB floored by dacitic/andesitic lava domes and overlain by
vocaniclastic infill, dates back to Upper Pliocene-Lower Pleistocene. A complex
mosaic of lithologically diverse sedimentary units (blocks), confined by neotectonic
marginal faults, characterizes the CVSB (Fig. 2). The most pronounced of these faults
being the NW-trending Vromolimni-Kondaros fault (Papanikolaou et al., 1990) that
has been proposed as the trigger of the hydrothermal activity that deposited Mn ore in
the CVSB (Papanikolaou et al., 1990; Liakopoulos et al., 2001; Alfieris et al., 2013;
Papavassiliou et al., 2017). The stratigraphically tight coupling between Mn and Fe
deposition, linked by Fe oxide minerals in feeder-veins, and positive Eu anomalies
(Fig. 14) indicating vent-sourced Fe (Maynard, 2010), associate Fe mineralization to
fault-triggered hydrothermalism in the CVSB. This is consistent with models of
geothermal fluid circulation along fault lines as conduits for the Mn-rich fluids that
formed the Milos Mn ore deposit (Hein et al., 2000; Liakopoulos et al., 2001; Glasby
et al., 2005; Kilias, 2012; Papavassiliou et al., 2017). More importantly, the overall
complex neotectonic structure of the CVSB (Papanikolaou et al., 1990) would explain
the creation of restricted basins, with sedimentological, lithological and geothermal
conditions that enabled the development of unique biogeochemical circumstances in
which the NFIF and MFIF formed.
The presence of the three depositional basins is supported by the fact that the
sequence lithologies in each fault-bound unit are characterized exclusively by



occurrences of specific and variably thick stratigraphic packages that tend to be
absent in others. For example, the MFIF occurs restricted to basin 1 and the NFIF to
Basin 3. Basin 2 is further distinguished by 35-50 m thick interbedded ore-grade Mn-
mineralized and glauconitic sandstones/sandy tuffs, much less developed in Basins 1
and 3 (Fig. 2). The presence or absence of a stratigraphic sequence, together with its
thickness variation, are interpreted as a result of local syntectonic sediment formation
conditions in each basin as a result of block tectonic movements along fault lines
(Papanikolaou et al., 1990; Nijman et al., 1998). It may also be attributed to unique
basin scale water column redox conditions (e.g. Bekker et al., 2010, and references
therein), post-depositional erosion and changing sea level stand (Cattaneo & Steel,

2000).

The lack of hydrothermal feeder veins or seafloor exhalative structures (i.e.,

chimneys) in the MFIF and NFIF lithologies, suggests that hydrothermal Fe(II) was
delivered by diffuse flow and that the Milos-IF formed on the seafloor. Importantly, a
number of studies propose that the main Mn deposit in Basin 2 formed in two stages.
First boiling hydrothermal fluids precipitated sulfide at depth, leading to first
generation microbial-induced deposition of Mn oxides as pyrolusite and ramsdellite.
Tectonic uplift resulted in the replacement of the first generation Mn minerals by
second-generation Mn oxides, including cryptomelane (Hein et al., 2000;
Liakopoulous et al., 2001; Papavassiliou et al., 2017). Cryptomelane replacement of
the original Fe(III)(oxyhydr)oxides in MFIF therefore suggests that deposition of the
MFIF is coeval with first stage Mn deposition in Basin 2. This observation also
indicates that at this time, two active fault-bounded basins probably existed in the
CVSB; i.e., Basins 2 and 3 (Fig. 2). For example, the underlying Mn-enriched
sandstone lithology in Basin 3, stratigraphically correlated to the sandstone Mn



deposit in Basin 1 and the MFIF Fe-Si-rich rocks in Basin 2, justify this proposal (Fig.

2).

A subsequent geomorphological/chemical reconfiguration formed Basin 3,

orchestrating the deposition of the NFIF in a deeper, small-restricted basin (Fig. 2).
The deepening of Basin 3 is strongly demonstrated by an underlying fine upward
grading of a transgressive-type Fe-rich lag deposit, that transitions into the NFIF. This
uplifting into shallower water event that prompted second generation deposition of
Mn oxides in Basin 2 and the substitution of Fe(III)(oxyhydr)oxides by Mn in sub-
Basin 1, potentially triggered this environmental change in Basin 3. The MFIF and
NFIF sequences are therefore temporally and spatially distinct (Fig. 2).

**4.2 Formation Mechanism of The Milos BIFS**
**4.2.1 Paragenetic sequence**
It is stressed that the previously generalized model proposed for biological deposition
of the Milos IF, refers exclusively to parts of what is now designated as MFIF (Chi
Fru et al., 2013). The NFIF is strongly banded, but does not display the typical
microfossils seen in the MFIF, where diffused microbanding apparently relates to the
distribution of microbial mats in thin sections (Chi Fru et al., 2013, 2015). The
distinction of microcrystalline quartz and amorphous silica phases in the MFIF and
NFIF, respectively, together with nano-crystalline hematite particles, suggests a
primary amorphous silica origin in both deposits, diagenetically transformed to quartz
in the MFIF. The difference in silica crystallinity between the IFs is concurrent with
the older age predicted for the MFIF relative to the NFIF, from reconstructed
sequence stratigraphy (Fig. 2). Hematite in BIFs is generally interpreted, based on
thermodynamic stability, to be a transformation of various primary Fe(III) minerals,



with ferrihydrite often proposed as the principal precipitate from the water column
(Glasby and Schulz, 1999; Bekker et al., 2010; Johnson et al., 2008; Percoits et al.,
2009). It is thought that acidic pH yields mainly goethite while hematite is produced
at circumneutral pH (Schwertmann and Murad, 2007). The notable absence of
diagenetic magnetite and Fe carbonates (siderite and ankerite), point to negligible
coupling of primary Fe(III) oxyhydroxides reduction to organic matter oxidation by
the dissimilatory iron-reducing bacteria during burial diagenesis (Johnson et al.,
2008). Minor occurrence of iron-silicate phases (Chi Fru et al., 2015) indicates an
origin of the hematite precursor in seawater independent of the iron silicate proposed
in some cases (Fischer and Knoll, 2009; Rasmussen et al., 2013, 2014). The up to 50
wt% Fe content recorded in the Fe-rich bands, indicate that large amounts of
dissolved Fe(II) was intermittently sourced and deposited as primary Fe(III) minerals,
through various oxidative processes in the depositional basin.

Importantly, the CIA index does not support mass weathering and

mineralization of terrestrial Fe and Si, in agreement with the absence of rivers
draining into the CVSB (Chi Fru et al., 2013). The specific identification of plant
biolipids would at face value imply post-depositional contamination. However,
samples were sawn to remove exposed layers and only the laminated bands for the
NFIF were analyzed, while modern sediments from Spathi bay, located Southeast of
Milos Island where hydrothermal activity is presently ensuing at 12.5 m below sea
level, revealed similar plant lipids as recorded in the Quaternary IF (Fig. 15G). Post-
depositional contamination with terrestrial plant lipids is therefore ruled out for the
idea that recalcitrant plant biomass probably entered the sediments via seawater
entrainment at the time of deposition (see Naden et al., 2005). This finding
necessitates the careful interpretation of bulk $\delta^{13}C_{org}$ values obtained from both the



modern and ancient Milos sediments, involving in situ and ex situ biological
contributions to $^{13}C_{org}$ fractionation by various known carbon fixation pathways
(Preuß et al., 1989; Berg et al., 2010).
Such indication of mixing of the hydrothermal fluids with seawater may be
interpreted to negate a reducing depositional environment as suggested by the Ce
anomalies. However, Pichler & Veizer (1999) demonstrated that in the unconfined
seafloor shallow hydrothermal vent fields at Tatum Bay, Papua New Guinea,
experiencing little or no water column stratification, as low as 11% seawater is
involved in the precipitation of Fe(III)(oxyhydr)oxides from hydrothermal fluids and
at maximum 57%. It is therefore suggested that seawater mixing during deposition
was at the lower limits. This is demonstrated by the REE analysis and the presence of
anaerobic bacteria biomarkers in the NFIF formation, coupled to sediment lithology
and stratigraphy, as explained below.

**4.2.2 Tectono-sedimentary processes and band formation**
Fluctuation in hydrothermal activity is proposed to account for the banding in the
NFIF (Fig. 16), under redox depositional conditions inferred to be mainly reducing
for both investigated IFs, consistent with previous reports (Chi Fru et al., 2013, 2015).
Positive Eu anomalies indicate a hydrothermal origin for all but one of the sample
suite (Fig. 14A). However, statistically calculated Eu/Eu* anomalies ($Eu_{(SN)}$/
$(0.66Sm_{(SN)} + 0.33Tb_{(SN)})$) to correct for differences in Gd anomalies commonly
encountered in seawater (Planavsky et al., 2010) are in the range of 0.1-0.58,
averaging 0.42. The values are closer to the anoxic water column values calculated for
Archean IFs, compared to Paleoproterozoic IFs (Planavsky et al., 2010), which may
be due to their deposition in an active volcanic center like most of the Archean



Agloma BIFs (Bekker et al., 2010; Chi Fru et al., 2015). The lack of statistically
significant true negative Ce anomalies (Fig. 14B) is interpreted to indicate a reducing
depositional environment for both IFs.

CIA indices traditionally provide relative information on contributions from

chemical weathering to sediment deposition, linked to operative hydrological and
climatological patterns on land. This information is often gleaned from ancient and
modern soils and from reworked siliciclastic deposits in marine basins (Maynard, 1993;
Bahlburg & Dobrzinski, 2011). The calculated CIA indices, however, are closer to the
range obtained for unweathered and or only minimally weathered volcanic rocks (e.g.,
Nesbitt & Young, 1982; Bahlburg & Dobrzinski, 2011), thus pointing to a
predominantly volcanic and/or hydrothermal provenance for the clastic sedimentary
materials in the IFs (also see Alfieris et al., 2013).

It has been suggested that the release of reduced submarine hydrothermal fluids

contributed towards maintaining water column anoxia during the deposition of
Precambrian BIFs (Bekker et al., 2010). The calculated Eu anomalies (Fig. 14) and
petrographic data showing volcaniclastic detritus (i.e., K-feldspar, sanidine, tridymite,
cristobalite) as key rock components are in agreement with a submarine hydrothermal
source for the investigated IFs. The coarse volcaniclastic detritus embedded in the Fe-
rich bands compared to the finer particles in the Si-rich layers, highlights rapid
oxidation of Fe(II) that coincided with periodic cycles of hydrothermal/volcanic
discharge of new materials into the water column. However, the fine-grained nature
of both the MFIF and NFIF deposits suggests that deposition likely occurred away
from where such activity was occurring or that volcanic/hydrothermal discharge of Fe
and Si was non-eruptive and disruptive. The Fe-rich bands repetitively revealed
hematite grains cementing the denser volcaniclastic fragments that gradually diminish



upwards into a zone of fine-grained hematite before transitioning into Si-rich bands
consisting mainly of finer volcaniclastic detritus. These observations provide four
valuable interpretational considerations for proposing a model for the formation of the
alternating Si and Fe-rich bands.
1. The Si and Fe oxides-rich bands are a primary precipitate formed in the water
column, by a process in which the precipitation of amorphous Si occurred
during quiescent non-volcanic intervals, after the oxidation and precipitation
of reduced Fe intermittently introduced into the water column by
volcanic/hydrothermal activity to form the Fe oxides.
2. The repetitive zonation of distinct particle sizes, suggests density gradient
sedimentation that requires a water column-like environment, rather than
diagenetic alteration of pre-formed sediments by hydrothermal fluids.
3. The lack of statistically significant Ce anomaly across the Si and Fe-rich units
does not support sediment diagenesis as an alternative model for explaining
the origin of the Milos IF, in favor for a primary water column source. This is
because the oxidation of ferrous Fe supplied by reduced hydrothermal fluids
to iron oxides, requires coincidental interaction with a sizeable pool of
oxygen (Johnson et al., 2008). Otherwise, light-controlled photoferrotrophy—
an extremely rare sediment characteristic—precipitates Fe oxides in the
absence of oxygen (Weber et al., 2006).
4. The style of deposition of the MFIF and NFIF is distinct from the post-
depositional infilling of a porous sandstone sediment matrix during the
formation of the Mn ores. Instead the deposition of the MFIF and NFIF in
restricted portions in the basins not associated with previously accumulated
sandstones, and the difficulty and lack of evidence to provide a viable



biogeochemical mechanism for the formation of the even bands of alternating
Si and Fe-rich layers of several meters high and wide, does not support post-
depositional pore filling of a porous sandstone matrix by Fe, as a potential
pathway to the formation of the Milos IF.

**4.2.3 Biological involvement**
Hematite precipitation in the MFIF on microbial filaments (Chi Fru et al., 2013) was
previously used to propose a generalized basin-scale mechanism for the deposition of
Fe-rich rocks in Cape Vani. However such filaments are absent in the NFIF, while
pure hematite grains are tightly bound to relics of an organic matter signal carrying a
maximum $\delta^{13}C_{org}$ signature of -25‰ (Table 2). Similar processes are recorded in
modern marine sediments where interactions between Fe and free organic matter has
been reported to enable the preservation up to 21.5wt% of total organic carbon over
geological time scales (Lalonde et al., 2012). Moreover, Fe generally traps and
preserves organic matter at redox interfaces (Riedel et al., 2013). The data appear to
suggest that the mechanism of Fe(III) (oxyhydr)oxide precipitation and preservation
varied between the two IFs. The lack of similar photoferrotrophic-like filamentous
fossils reported in the MFIF (Chi Fru et al., 2013), in the NFIF, does not however rule
out the potential role of microbial involvement in Fe(II) oxidation, since diverse
microbial taxa carry out this process, several of which are non-filamentous (Chi Fru et
al., 2012). However, our data is insufficient to enable clear quantification of the levels
of abiotic vs. biotic contribution to Fe(II) oxidation in the NFIF. Nevertheless, the
inferred predominantly anoxic depositional conditions as explained above, together
with the identification of anaerobic bacteria biomarkers in the laminated bands,
intuitively favor significant contribution of anaerobic biological Fe(II) oxidation in



the precipitation of primary Fe(III)(oxyhydr)oxides in the NFIF. See Weber et al.,
2006, for a review of potential biological pathways to anaerobic Fe(II) oxidation.
Briefly, anaerobic microbial Fe(II) oxidation can proceed via nitrate reduction
and by photoferrotrophy to deposit Fe(III)(oxyhydr)oxides. These mechanisms have
been linked to microbial contribution to BIF formation (Weber et al., 2006; Kappler et
al., 2005) and also for the MFIF (Chi Fru et al., 2013). However, it is also possible
that microaerophilic neutrophilic Fe(II)-oxidizing bacteria likely played an important
role, assuming a depositional setting analogous to the Santorini caldera and Kolumbo
shallow submarine volcanoes, where such low-$O_2$-dependent microbial Fe(II)
oxidation has been identified to actively precipitate Fe(III) (oxyhydr)oxides (Kilias et
al., 2013; Camilli et al., 2015). It appears that in the MFIF, precipitating
Fe(III)(oxyhydr)oxide minerals were bound and preserved free of organic carbon or
that such organic carbon was diagenetically degraded. As was previously shown,
Fe(III)(oxyhydr)oxides completely replaced the organic content of the filamentous
microfossils in the MFIF (Chi Fru et al., 2013).
The $10MeC_{16:0}$ FAME identified in the rocks has been reported in anaerobic
organisms coupling nitrite reduction to methane oxidation (Kool et al., 2012), in
sulfate and iron-reducing bacterial species such as Desulfobacter, Desulfobacula
(Bühring et al., 2005; Dowling et al., 1986; Taylor and Parkes, 1983), Geobacter,
Marinobacter and the marine denitrifier, Pseudomonas nautical (Kool et al., 2006;
Bühring et al., 2005; Dowling et al., 1986). It had previously been proposed that post-
depositional denitrification was a potential pathway for early organic matter removal,
justified by the low rock organic carbon and nitrogen content in the Milos BIF-type
rocks (Chi Fru et al., 2013, 2015; Table 2). Equally, the detected $10MeC_{16:0}$ FAME
has also been found in anaerobic oxidation of methane (AOM) communities (Alain et



al., 2006; Blumenberg et al., 2004), originating from sulfate reducing bacteria.
However, bulk sediment $\delta^{13}C_{org}$ of −20‰ does not reflect AOM activity that is
expected to produce bulk $\delta^{13}C_{org}$ values that are $\leq$−30‰. Low $10MeC_{16:0}$ FAME
concentrations frustrated attempts at acquiring its compound specific isotopic
signature to enable further biomolecular level reconstruction of active microbial
metabolisms to explain Fe deposition mechanisms.

It is nevertheless puzzling why potential microbial biomarkers typical of marine

or hydrothermal vent environments are hardly preserved in the rocks, given that
microfossil evidence indicates a vast community of diverse prokaryotic assemblages
in the adjacent MFIF (Chi Fru et al., 2013, 2015). Moreover, sediments of the modern
Milos hydrothermal system and elsewhere on the HVA, are ubiquitously colonized by
microbial life, characterized by the marked large-scale absence or low abundance of
higher life forms, including plants (Kilias et al., 2013; Camilli et al., 2015; Oulas et
al., 2015). One possibility could be the discriminatory preservation of lipids related to
their selectivity and reactivity towards Fe(III)(oxyhydr)oxides and clays or different
pathways to diagenetic degradation (e.g., Canuel & Martens, 1996; Lü et al., 2010;
Riedel et al., 2013). As noted, the carbonaceous materials in the BIF-type NFIF rocks
occur in tight association with hematite.

Importantly, prokaryotic biomarkers are suggested to poorly preserve in these

young BIF analogues. This raises the possibility that this may provide an important
explanation for why lipid biomarkers are yet to be extracted from Precambrian BIFs.
Moreover, the data are compatible with low $C_{org}$ recorded in BIFs of all ages. They
suggest these unique BIF features may not be entirely related to metamorphic
degradation of organic matter, since the Milos BIF-type rocks are unmetamorphosed,
yet have vanishing $C_{org}$ levels similar to the ancient metamorphosed BIFs.



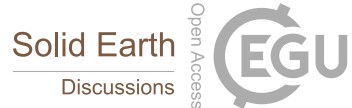


### 4.2.4 Mn layers and the deposition of the Si-Fe-rich facies

Cryptomelane $[K(Mn^{4+},Mn^{2+})_8O_{16}]$, commonly occurring in oxidized manganese
deposits resulting from mineral replacements and as open space fillings
(Papavassiliou et al., 2016), common in MFIF, supports the idea of post-depositional
impregnation of the base of the MFIF by Mn-rich fluids. Microscopic analysis
supports the epigenetic origin of the Mn in the MFIF by revealing Mn oxides growing
along fractures, impregnating and replacing Fe minerals (Fig. 4B-F). The
macroscopically evident thinning out to disappearance of such Mn-rich horizons up
the MFIF, coupled by their development along microfractures emphasizes this
epigenetic Mn origin. Mn is not a common feature of the NFIF, even though it sits on
top of a thin sandstone layer impregnated by Mn, that locally forms the cap of the
main Mn ore at Cape Vani. The generally accepted view is that Mn rich hydrothermal
fluids rose and mineralized the Cape Vani sandstones (Hein et al., 2000; Liakopoulos
et al., 2001; Glasby et al., 2005). Based on the stratigraphic location of the MFIF
which pre-dates the Mn-rich sandstones, it is proposed that impregnation of the MFIF
by Mn was coeval with large-scale Mn ore mineralization of the Cape Vani
sandstones, implying the entire basin was likely oxygenated at the time. The lack of
Ce anomalies suggests that both the MFIF and the NFIF formed in anoxic settings.
Similar data for the Mn oxides have suggested formation in oxic settings (Glasby et
al., 2005; Chi Fru et al., 2015). However, more sensitive proxies are needed to resolve
and confirm the stratigraphic and REEs-dependent interpretation of potential redox
conditions. This implies that Mn epigenetically replaced the MFIF, either because the
basin was tectonically uplifted into a high–energy oxygenated shallow water setting
or that sea level dropped, leading to partial metasomatism of the base of MFIF, when



oxygenated seawater mixed with hydrothermal fluids and precipitated Mn. The lack
of significant Ce anomalies in the dataset also indicates that for the final deposition of
the NFIF, an eventual deepening event or sea level rise, or both, were tectonically
triggered, resulting in deoxygenation of parts of the CVSB.
All of this is feasible with the three-basin-fault-bounded hypothesis as a
requirement for movement along fault lines in response to temporal tectonic
activation. Importantly, a deepening event is suggested by the sudden change from the
underlying Mn-rich layer into the conglomeratic deposit (Fig. 8B) often associated
with sedimentary features that form during sea level rise and the landward migration
of the shoreline (Cattaneo & Steel, 2000). The upward sequential transition from the
Mn-rich facies through pebbly to fine-grained sediment to the NFIF, strongly implies
that the underlying Mn-rich facies and NFIF layers formed in shallower and deeper
waters, respectively, or that they are separated by an erosional unconformity. This
study proposes that the NFIF that overlies the transgressive-type conglomeratic lag
along an erosional contact surface was likely deposited during maximum flooding,
when the basin became stagnant and stratified, and subsequently was uplifted to
emergence. Similar transgression-type lithologies are indicated to have regulated
primary sedimentation styles during the deposition of nearshore Paleoproterozoic
BIFs (Pufahl and Fralick, 2004; Pufahl et al., 2014). Moreover, deposition of BIFs in
sandstone/grainstone-dominated environments has also been suggested for
Precambrian IFs (Simonson, 1985; Simonson and Goode, 1989; Pufahl and Fralick,

2004).

Uplifting is suggested by potential the weathering of the NFIF to form the
ferruginous duricrust cap. Comparable ferruginous layers on Precambrian BIFs are
linked to pervasive subaerial chemical weathering, via the dissolution of the silica-



rich layers and precipitation of relatively stable Fe oxides in the spaces between more
resistant hematite crystals (e.g., Dorr, 1964; Shuster et al., 2012; Levett et al., 2016).
This collective evidence supports the existence of a geodynamic tectonic system
capable of producing shallow oxic to deeper anoxic basin conditions at different times
that would explain the existence of Mn and Fe oxide layers within the same
sedimentary sequence. For example, it is common knowledge that both Fe and Mn
oxides will precipitate in the presence of oxygen (Roy, 1997, 2006), with kinetic rates
usually being faster for the oxidation of reduced Fe than reduced Mn. In the Fe(II)-
rich conditions that prevail in anoxic settings, abiotic reactions between Fe(II) and Mn
oxides, produce Fe(III) leading to the dissolution of the Mn oxides to form reduced
Mn, implying Mn oxides should not accumulate (Dieke, 1985). Moreover, under these
conditions, biological precipitation of Fe(III) can occur rapidly, leaving dissolved Mn
in solution to be deposited when oxygen becomes available. Given that the
hydrothermal fluids of the Hellenic Volcanic Arc are commonly enriched in both
reduced Fe and Mn, the deposition of the MFIF and NFIF therefore implies there was
an existing mechanism that enabled the kinetic discrimination and deposition of the
oxides of Fe and Mn into separate settings, most likely dependent on prevailing redox
conditions. The accumulation of the ferruginous duricrust layer, overprinted by redox
sensitive Mn-nodules, above the NFIF indicates a new shallowing event might have
terminated the formation of the NFIF.

**4.2.5 Modern analogues on the HVA**
Mechanistic explanation for the development of potential stratified waters and
reducing conditions during the deposition of the Milos BIF is problematic. However



evidence is available from present shallow submarine hydrothermal analogues in the
central part of the HVA, to which the CVSB belongs. These include:

(1) The crater floor of the Kolumbo shallow-submarine volcano (~600×1200

$m^3$), which rises from 504 to 18 m below sea level near Santorini, (Sigurdsson et al.,
2006; Carey et al., 2013; Kilias et al., 2013).

(2) The N part of Santorini's submerged caldera walls, which rises from 390 m

below sea level to over 300 m above sea level (Druitt et al., 1999; Friedrich et al.,
2006; Nomikou et al., 2013; Camilli et al., 2015).

(3) The coastal embayments at the Kameni emergent volcanic islands in the

centre of the Santorini caldera (Hanert, 2002; Nomikou et al., 2014; Robbins et al.,

2016).

The benthic waters within Kolumbo's crater potentially sustain $O_2$ depleted

conditions via stable $CO_2$-induced water column densification, and accumulation of
acidic water (pH~5), extending ~10 m above the $CO_2$ venting crater floor (Kilias et
al., 2013). This phenomenon is believed to lead not only to obstruction of vertical
mixing of bottom acidic water, but also to $O_2$ deprivation by precluding efficient
transfer of oxygenated surface seawater into the deeper crater layer. In addition,
diffuse $CO_2$ degassing is believed to be linked to the formation of Fe microbial mats
and amorphous Fe(III) oxyhydroxides on the entire Kolumbo crater floor (Kilias et
al., 2013). Prerequisites for the $O_2$ depleted conditions to happen are the closed
geometry of the Kolumbo crater and the virtually pure $CO_2$ composition of the
released hydrothermal vent fluids that produce oxygen stratification along a stable
$CO_2$-pH gradient.

A similar scenario is reported for the Santorini caldera, where large (~5 m

diameter) $CO_2$-rich, acidic (pH, ~5.93) hydrothermal seafloor pools and flow



channels, develop within m-thick microbial Fe-mats on the seafloor slope at 250-230
m below sea level. Persistent hypoxia exists in these pools, representing concentrated
seafloor $CO_2$ accumulation centers generated by hydrothermal venting (Camilli et al.,
2015). Here, the dissolved $O_2$ content (~80 μM or less) in the pools is ~40 % depleted
relative to the surrounding ambient seawater (Camilli et al., 2015). These hypoxic
conditions are comparable to or even lower than those measured in the $CO_2$-rich
oxygen minimum zones of coastal oceans, relative to seawater existing in equilibrium
with atmospheric $pO_2$ and $pCO_2$ pressures (Paulmier et al., 2008, 2011; Franco et al.,
2014). These conditions enable strong redox stratification of the pool waters, in which
unique Si- and Fe-rich microbial mats are associated with amorphous opal and
Fe(III)(oxyhydro)xides (Camilli et al., 2015). Importantly, the Fe microbial mats in
these $CO_2$-rich hypoxic pools are affiliated with specific microaerophilic Fe(II)-
oxidizing bacteria that accumulate Fe(III) oxyhydroxides (Camilli et al., 2015; Oulas
et al., 2015). These Fe bacteria are implicated in the deposition of the Precambrian
BIFs (Konhasuer et al., 2002; Planavsky et al., 2009; Bekker et al., 2010).

Hypoxia is also associated with the water column of the Fe(III)-rich coastal

embayments and their hydrothermal vents (≤1.0 m water depth), Kameni islands,
considered a modern analogue environment for the precipitation of Precambrian BIFs
(Hanert, 2002; Robbins et al., 2016 and references therein). Venting fluids are warm
(20-40 ℃), acidic to circumneutral (pH 5.5-6.9), enriched in $CO_2$, Fe and Si
(Georgalas & Liatsikas, 1936, Böstrom et al., 1990; Handley et al., 2010; Robbins et
al., 2016). Water column stratification is expressed as decreasing $O_2$ with depth that is
positively related to Fe(III)(oxyhydr)oxide density and microaerophilic Fe(II)-
oxidizing bacterial prevalence (Hanert, 2002). Robbins et al. (2016) found that
Fe(III)-rich suspended particulate material in these "Fe bays" may be associated with



anoxia, extending up to the air-seawater interface, near the hydrothermal vents
(Hanert, 2002). They consist of ferrihydrite, goethite and microaerophilic Fe(II)
oxidizers.

The biogeochemical occurrence of these phenomena within the localized

confines of the Santorini caldera and Kolumbo crater, may however be difficult to
achieve in ordinary shallow submarine hydrothermal settings, such as those occurring
on the coast of present day Milos. The same may be true for Tatum Bay, where non-
volcanic and unconfined diffuse hydrothermalism is widespread (Dando et al., 1996;
Pichler & Dix, 1996; Pichler & Veizer, 1999; Stüben et al., 1999; Rancourt et al.,
2001; Varnavas et al., 2005).

In the Kolumbo and Santorini hydrothermal fields, benthic pH averages 5.5 and

the deposition of carbonates is markedly absent (Kilias et al., 2013, Camilli et al.,
2015; Robins et al., 2016). This conforms to observations in the MFIF and NFIF units
where carbonate mineralization is not detected, thereby suggesting a similar low pH
depositional    environment    for    both    the    MFIF    and    NFIF.    Ubiquitous
Fe(III)(oxyhydr)oxide precipitation and enriched Si content are prevalent in the $CO_2$-
rich-hypoxic shallow submarine Santorini caldera slope pools and the Kameni Fe-
embayments where sulfide precipitation is inhibited (Camilli et al., 2015), or
extremely rare (Robbins et al., 2016). Such sulfide-poor conditions are critical for the
formation of BIFs (Bekker et al., 2010). Moreover, the anoxic amorphous Si-
Fe(III)(oxyhydr)oxide-rich-sulfide-poor shallow submarine environments at Kameni
islands, have been independently proposed as a modern analogue environment for
Precambrian BIF precipitation (Hanert, 2002; Robins et al., 2016).

A high Si-Fe(III)(oxyhydr)oxide content, absence of carbonate and sulfide

mineralization, coupled to a generally low S content have also been demonstrated for




959 the CVSB Fe formations (Chi Fru et al., 2013, 2015). This depositional situation is

960 different, for example, from the unconfined shallow submarine hydrothermal systems

961 in Tatum Bay and Bahia Concepcion Bahia Carlifornia Sur, Mexico, where authigenic

962 carbonate deposition is widespread (Canet et al., 2005; Pichler & Dix, 1996, 2005).

963 Moreover, there is strong geological evidence that within volcanic crater

964 environments associated with high $CO_2$ emission, long-term water column redox

965 stratification is possible under these special conditions. Further evidence is found in

966 volcanic crater lakes (for example the shallow 205 m deep lake Nyos in Cameroon—

967 renowned as one of Earth's three $CO_2$ saturated volcanic lakes (Ozawa et al., 2016;

968 Kling et al., 2005)). Here $CO_2$-induced water column stratification is associated with

969 bottom reducing conditions characterized by a low sulfate and high Fe bottom water

970 content relative to surface concentrations (Tiodjio et al., 2014).

972 **5 Concluding remarks**

973 This study shows the following new insights in light of what was previously known:

974 1. At least two distinct IFs (MFIF and NFIF) formed from hydrothermal mud,

975 within two localized sub-basins in the ~1 km-long CVSB, ~2.66-1.0 Myr ago,

976 controlled by local tectonism.

977 2. A working model that band formation may involve potential

978 Fe(III)(oxyhydr)oxide filling of sediment pores and fractures during

979 diagenesis, is not supported by the data. In addition to the lack of observation

980 of such phenomena, as demonstrated for replacive Mn mineralization,

981 calculated Ce and Eu anomalies, together with preliminary sequential iron

982 extraction analysis (Poulton and Canfield, 2011; data not shown), are





suggestive of anoxic depositional conditions likely induced by the release of

reduced hydrothermal/volcanic fluids into a cutoff sedimentary basin.

3. The precipitation of Fe(III) and Mn oxides require oxygen. In the absence of

oxygen, Mn is not oxidized, while light and photoferrotrophy will oxidize

reduced Fe to Fe(III)(oxyhydr)oxides. Both light and photoferrotrophy are

however extremely rare characteristics of anoxic sediments, but a common

feature of anoxic $Fe^{2+}$-rich waters, where photoferrotrophy is widespread

(Weber et al., 2006). Collectively, these observations provide an important

feasible mechanism for the knife sharp separation of the Mn oxide-rich ores

in the CVSB that are also Fe(III)(oxyhydr)oxide-rich, from the highly

localized MFIF and NFIF deposits that are Fe(III)(oxyhydr)oxide-rich but Mn

oxide-poor.

4. The mechanism of formation of the MFIF and NFIF therefore most likely

involved exhalative release of reduced hydrothermal/volcanic fluids into a

restricted and deoxygenated seafloor water column where the oxidation of

reduced Fe to Fe(III)(oxyhydr)oxides occurred, most likely by the activity of

photoferrotrophs (Chi Fru et al., 2013).

5. Episodic intensification of hydrothermal activity is identified as a main

mechanism for the formation of the millimetric BIF bands (Fig. 16), adding to

the biological mechanism that was inferred from fossil records in the MFIF

(Chi Fru et al., 2013, 2015).

6. Abiotic Si precipitation was apparently much slower relative to Fe(III)

precipitation, resulting in Fe-rich bands in the NFIF forming in association

with large fragments of volcaniclast and the Si-rich bands with fine Si grains.



7. A combination of the above processes produced pulses of Si and Fe in the millimetric Si and Fe-rich bands in the NFIF.

8. The Milos rocks fulfill sedimentological, chemical and mineralogical characteristics that established them as potentially the youngest known BIFs; following the simplistic definition that BIFs are sedimentary rocks composed of alternating layers of Fe and Si containing at least 15% iron.

9. Whether the rocks described here are analogues of Precambrian BIFs or not, and whether the proposed formation mechanisms match those that formed the ancient rocks, is opened to debate. Nonetheless, the present study provides mechanisms by which rocks with Fe and Si-rich bands can be formed in the modern oceans.

*Data availability.* Data can be accessed by request from any of the authors

*Author contributions.* ECF, SK and MI designed the study. ECF, SK, KG and MI performed fieldwork. ECF, JER, KG, IM and QH performed research. ECF, SK, KG, IM, QH and JER interpreted data. ECF and SK wrote paper.

*Competing interests.* The authors declare that they have no conflict of interest.

*Acknowledgments.* Ariadne Argyraki, Nicole Posth, Nolwenn Callac and Eva Zygouri are acknowledged field assistance during sampling and for stimulating intellectual discussions. Special thanks to Christoffer Hemmingsson for contributing to the SEM and XRD analyses. This work is funded by the European Research Council grant No. 336092 to ECF and the Swedish Research Council grant No. 2012-4364 to MI.




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




Table 1. Table 1. Results of X-Ray Radiation (XRD) analysis showing major
mineralogical compositions. NFIF (non-fossiliferous iron formation) and MFIF
(microfossiliferous iron formation), respectively.

| Mineral phase | MFIF1 | MFIF2 | MFIF3 | Fe-rich NFIF2A | Si-rich NFIIF2B | Fe-rich NFIF2C | Si-rich NFIFD | Fe-rich NFIF2E | Fe-rich NFIF2F |
|---|---|---|---|---|---|---|---|---|---|
| Hematite | + | + | - | + | + | + | + | + | + |
| Quartz | + | + | + | - | - | - | - | - | - |
| Sanidine | - | - | - | + | + | + | + | + | + |
| Tridymite | - | - | - | - | + | + | + | + | + |
| Cristobalite | - | - | - | + | - | - | - | - | - |
| Cryptomelane | - | - | + | - | - | - | - | - | - |






Table 2. Stable isotope results. Letters A-F on the NFIF samples represent respective
bands of the sawn rock in Figure 7E.

| Sample | $\delta^{13}C_{org}$ vs PDB (‰) | $C_{org}$ (%) | $\delta^{15}N$ vs air (‰) | N (%) | $\delta^{34}S$ vs CDT (‰) | S (%) |
|---|---|---|---|---|---|---|
| Fe-rich NFIF2A | -25,63 | 0,061 | nd | 0,023 | nd | 0,01 |
| Si-rich NFIF2B | -25,03 | 0,109 | nd | 0,017 | nd | 0,02 |
| Fe-rich NFIF2C | -24,45 | 0,068 | nd | 0,013 | nd | 0,02 |
| Si-rich NFIF2D | -25,04 | 0,076 | nd | 0,015 | nd | 0,02 |
| Fe-rich NFIF2E | -25,19 | 0,042 | nd | 0,009 | nd | 0,01 |
| Si-rich NFIF2F | -25,49 | 0,050 | nd | 0,012 | nd | 0,03 |
| MFIF1 | -25,49 | 0,087 | nd | 0,017 | nd | 0,01 |
| MFIF2 | -26,25 | 0,046 | nd | 0,005 | nd | nd |
| MFIF3 | -25,69 | 0,041 | nd | 0,006 | nd | nd |

ND, Not detected















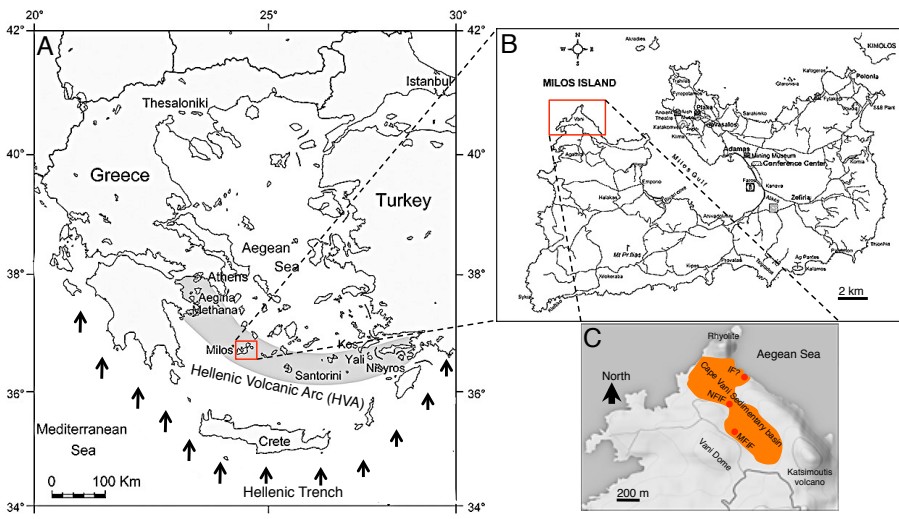


Fig. 1. Geological map of Milos (redrawn from Marschik et al., 2010). (A),
Geotectonic map showing the position of Milos Island, along the Hellenic Volcanic
Arc (HVA). Arrows indicate the direction of subduction of the African plate
underneath the Euroasian plate. (B) Milos Island. (C), The Milos iron formation is
located in the 8-shaped Cape Vani sedimentary basin (CVSB). At least two IFs are
present in the CVSB. These are made up of a non-fossiliferous IF (NFIF) at the
juncture between the two large sedimentary basins and a microfossiliferous iron
formation (MFIF) located at the SW margin in the second basin. A potential third IF
(IF?) is located NE, close to the present day Aegean Sea. It is however not certain if
this deposit is part of the NFIF or not, because of the open mining pit separating the
two.


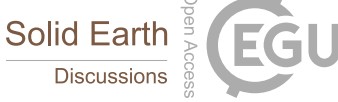

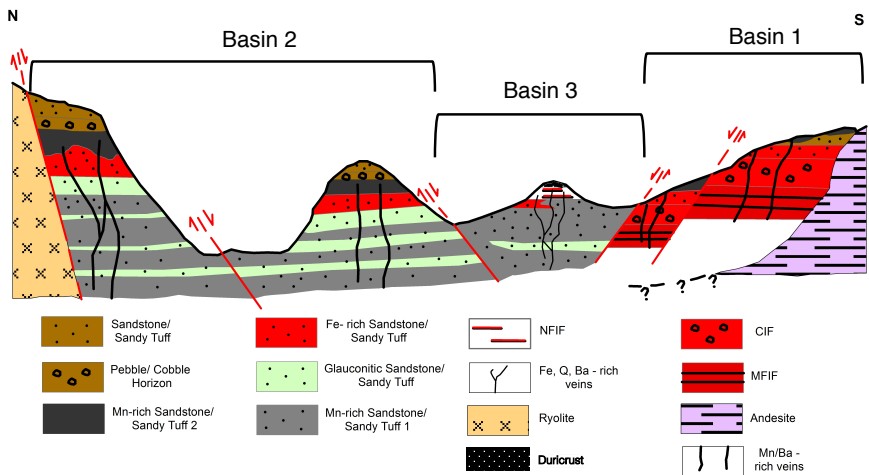


Fig. 2. Generalized schematic north-south geologic cross section through the ~1 km long CVSB showing interpreted geology, relationships between the main lithofacies, main fault locations, the iron and manganese formations, in support of a proposed three-basin hypothesis. Not drawn to scale. Four types of iron-rich sedimentary rocks occur in the CVSB. These include the iron-rich sandstones, the iron-Mn-rich sandstones, the conglomerate hosted iron formation (CIF) and the MFIF and NFIF formations that are depositionally and chemically distinct from the sandstone deposits.







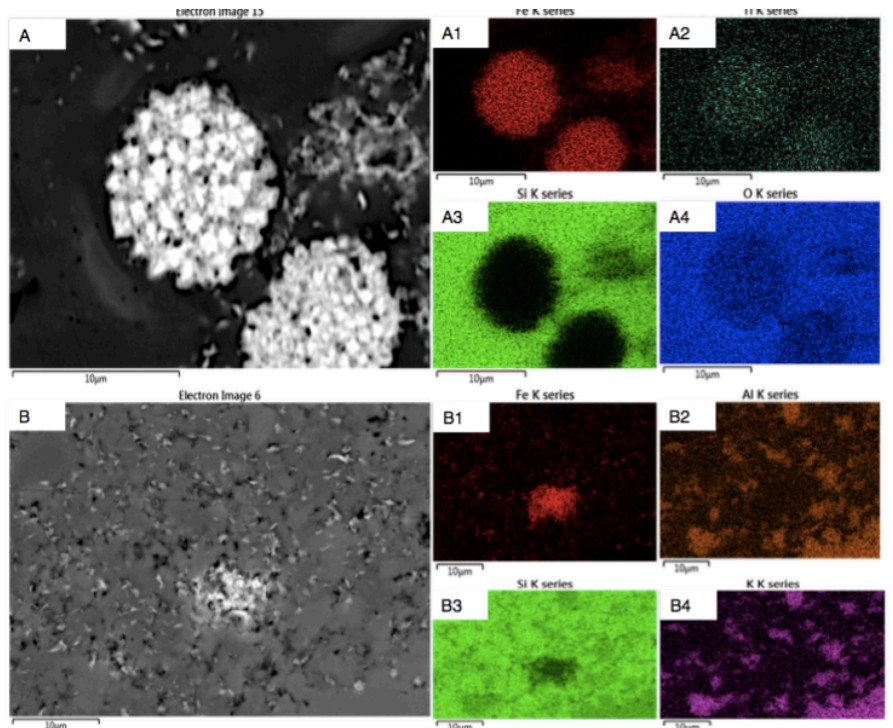


Fig. 3. EDS-electron image showing different Fe-rich mineral phases in a Si-rich
matrix from the MFIF. The bright colours correspond to the analysed elements. (A),
framboidal hematite particles. A1-A4, different element compositions associated with
framboidal particles in panel A. (B), Dispersed fluffy Fe-rich mineral grains. B1-B4,
corresponding elements associated with the micrograph in panel A.






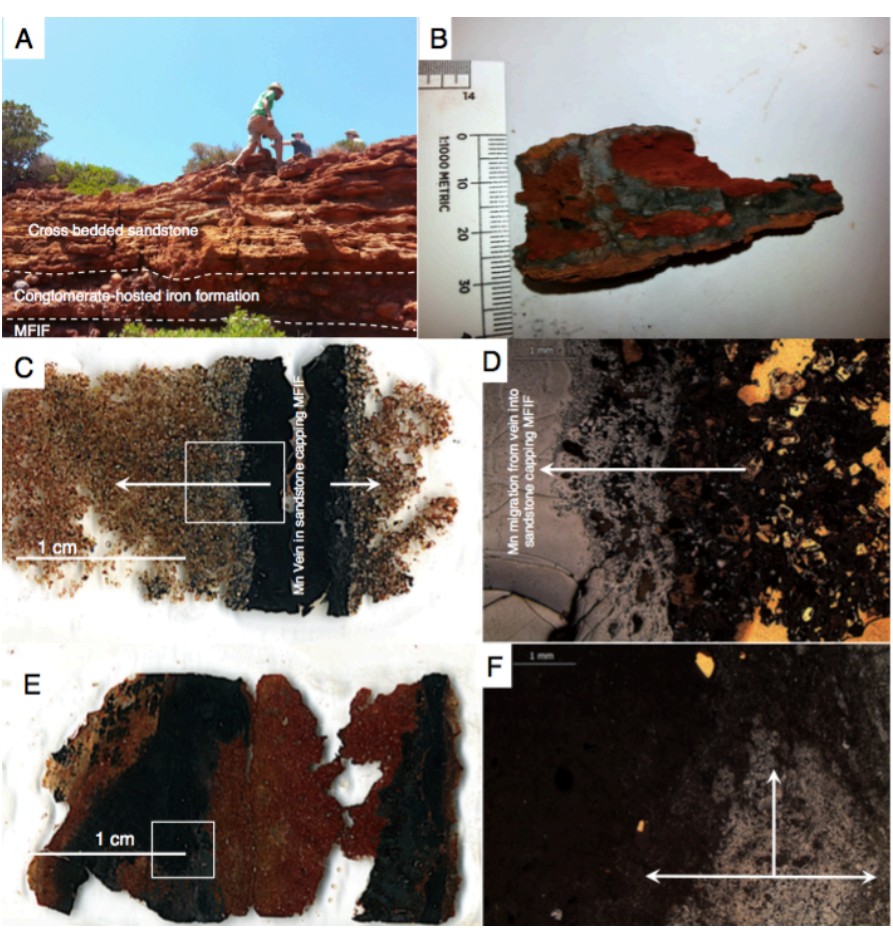


Fig. 4. Sedimentary profile, thin section scans and optical microscope images of the
MFIF. (A), Field photo showing the sedimentary profile of the MFIF chacterized by
the overlying sandstone cap. (B), Photograph showing black diffused Mn-rich bands
near the base of the MFIF. (C), Scanned image of thin section showing a black Mn-
rich vein in the overlying MFIF sandstone showing a gradient of Mn migrating into
the sandstone matrix (white arrows). (D), Light microscopy images showing details in
panel C. (E), Scanned image of an MFIF thin section showing black Mn bands
migration into a red iron-rich background. (F), Amplified light microscope image
showing gray Mn layers migrating into a black Fe-rich matrix. White arrows show
direction of movement. Boxes in C and E are amplified in D and F.






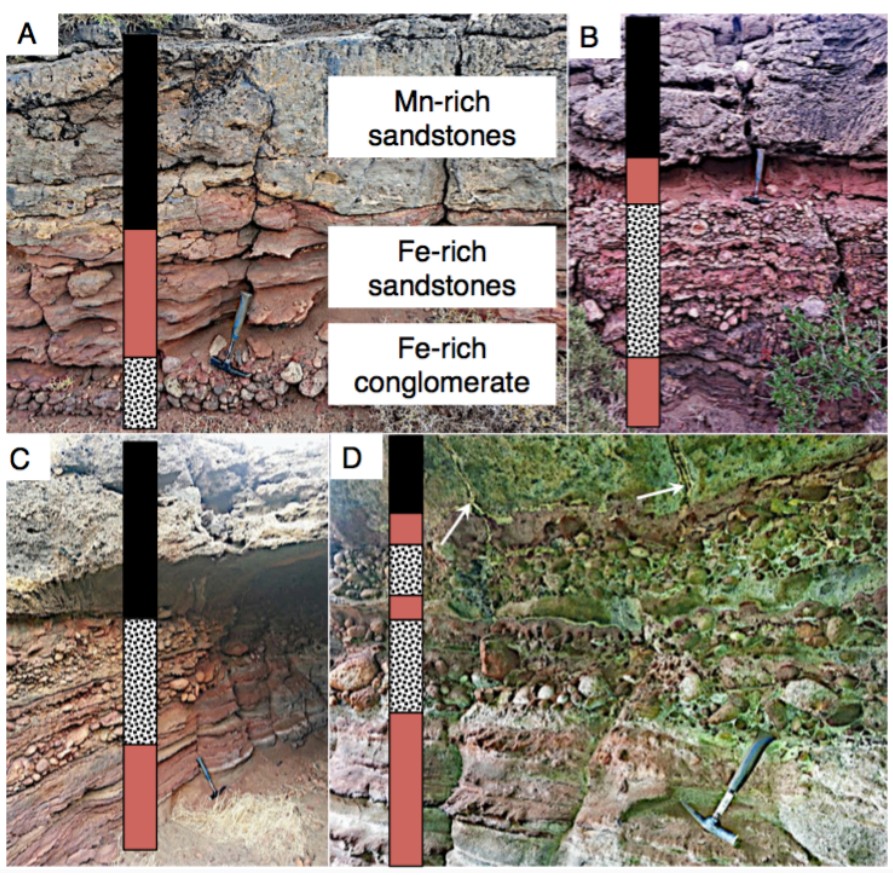

Fig. 5. Sedimentary sequence overlying the MFIF, consisting of thin (< 0.5 m)
polymictic andesite-dacite cobble-pebble, and sandstone-sandy tuff pebble, and Fe-
rich conglomerate facies overlain by thinly laminated Fe-rich sandstone beds. This
vertical sequence is interpreted to represent a progressively deeper water environment
deepening-upward sequence (A) as a result of sea level rise due to tectonic
subsidence. The multiple cycles shown in panels B-D signify several potential
episodes or sea level rise. Arrows in panel D showing hydrothermal feeder veins
feeding the overlying layers. The sequence is overlain by a thin package of parallel
and cross-bedded Mn-sandstone cap.




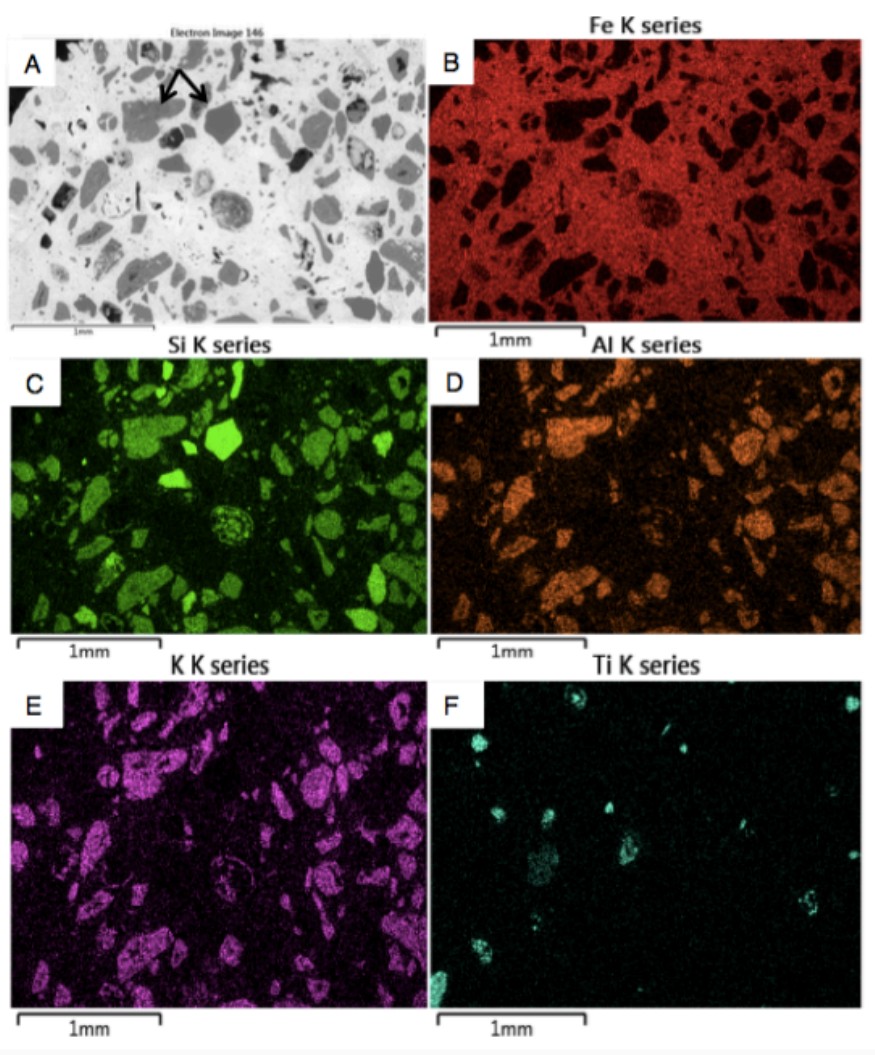


Fig. 6. Scanning electron microscope electron image of the volcaniclastic (K-feldspar)/ iron-rich sandstone layer overlying the MFIF.





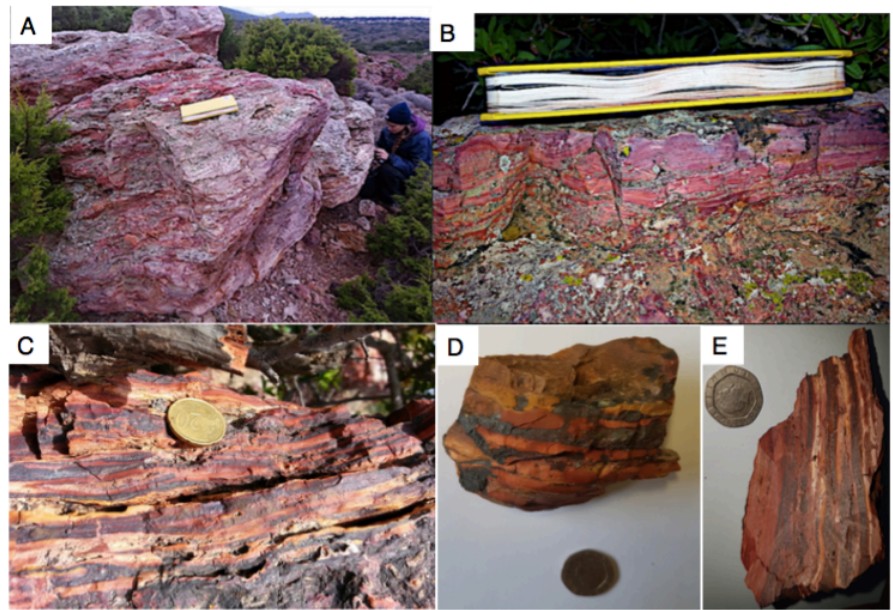


Fig. 7. Typical NFIF banded iron rocks. (A-C), Field photographs. (D), Handheld
banded Fe sample. (E), Sawn NFIF sample with laminated Fe-rich bands alternating
with Si-rich bands.





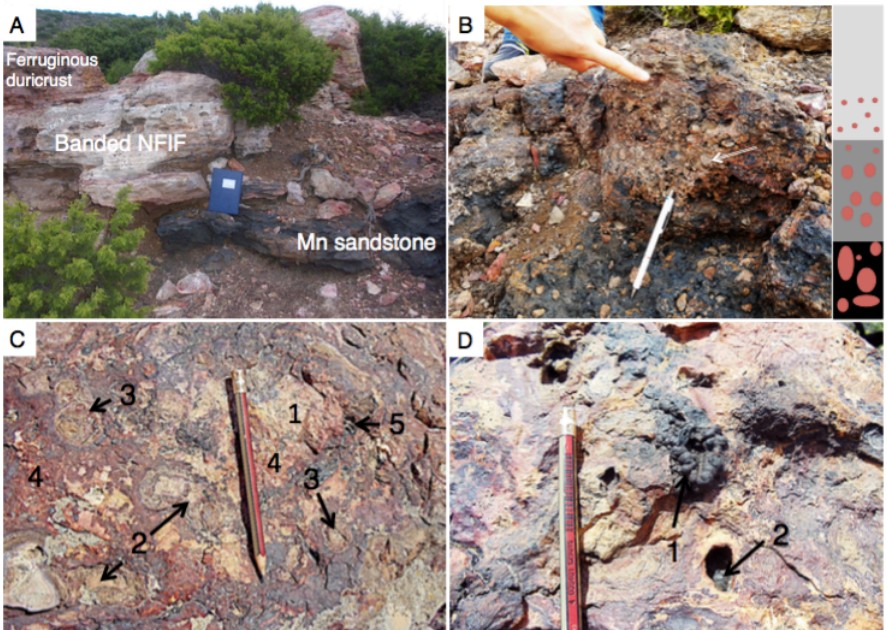

Fig. 8. Field sedimentology and stratigraphy of Section B sequence containing the
NFIF. (A), Sharp boundary between lower Mn sandstone and unconformably
overlying NFIF capped by a ferruginous duricrust. (B), Sandstone-sandy tuff pebble
to gravel conglomerate lag facies, showing an upward fining character and bored
clasts (black), locally overlies the Mn sandstone and capped by a sharp erosional
contact with the overlying NFIF. The tip of the pen (7 cm long) rests on late blue-
black Mn oxide overprint. (C), Ferruginous duricrust that comprises lithic fragments
composed of (1) Fe-nodules (2) and Fe-concretions (3) in a hematite-rich matrix (4).
(D), Matrix dissolution resulting in vermiform Mn nodules (1) and cavity black Mn
oxide (2) infillings, post-dating the ferruginous duricrust formation.




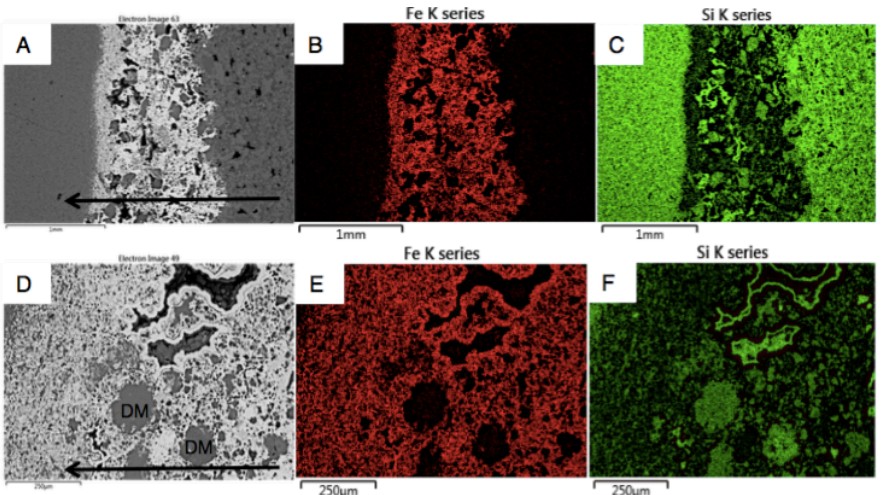


Fig. 9. EDS-electron image showing major elemental composition of typical Fe bands
alternating with Si-rich layers in the NFIF. Volcaniclastic detritus mostly present in
the Fe-rich bands, suggests precipitation during active submarine volcanism. To the
contrary, the Si-rich bands are composed of more fine-grain, signifying deposition
during periods of minimal volcanic activity. Arrows in panels (A) and (B) depict the
direction of sedimentation, which was often seen to proceed from an Fe-rich matrix
mixed with large grains of volcaniclastic detritus (DM) to one composed essentially
of very fine-grained Fe particles before transitioning into the very fine-grained Si-rich
layer. An upward fining of the volcaniclastic particles in the Fe-rich layers transitions
from one made up of volcaniclastic debris and hematite, to a mainly thin hematite-
rich horizon at the top of this mixed layer (see supplementary Figs 8-11 for details).
This concurrent occurrence of volcaniclast and Fe oxides and the upward fining
nature of the Fe-rich layers, suggest the release and oxidation of Fe(II) coincided with
the settling of hydrothermal debris resulting from the introduction of enormous
amount of reduced materials into the water column (Bekker et al., 2010). The iron-
rich layer ceased forming as hydrothermal/volcanic release of Fe subsided, followed
by deposition of the Si-rich layer. This repetitive cycle of events is observed for tens
of metres laterally and vertically, stressing that the layers are not single isolated or
post-depositional replacement events, but chemical precipitates that sequentially
sedimented out of the water column. Red colour in Panels (B) and (C) depict Fe and
green in panels (C) and (F), Si.





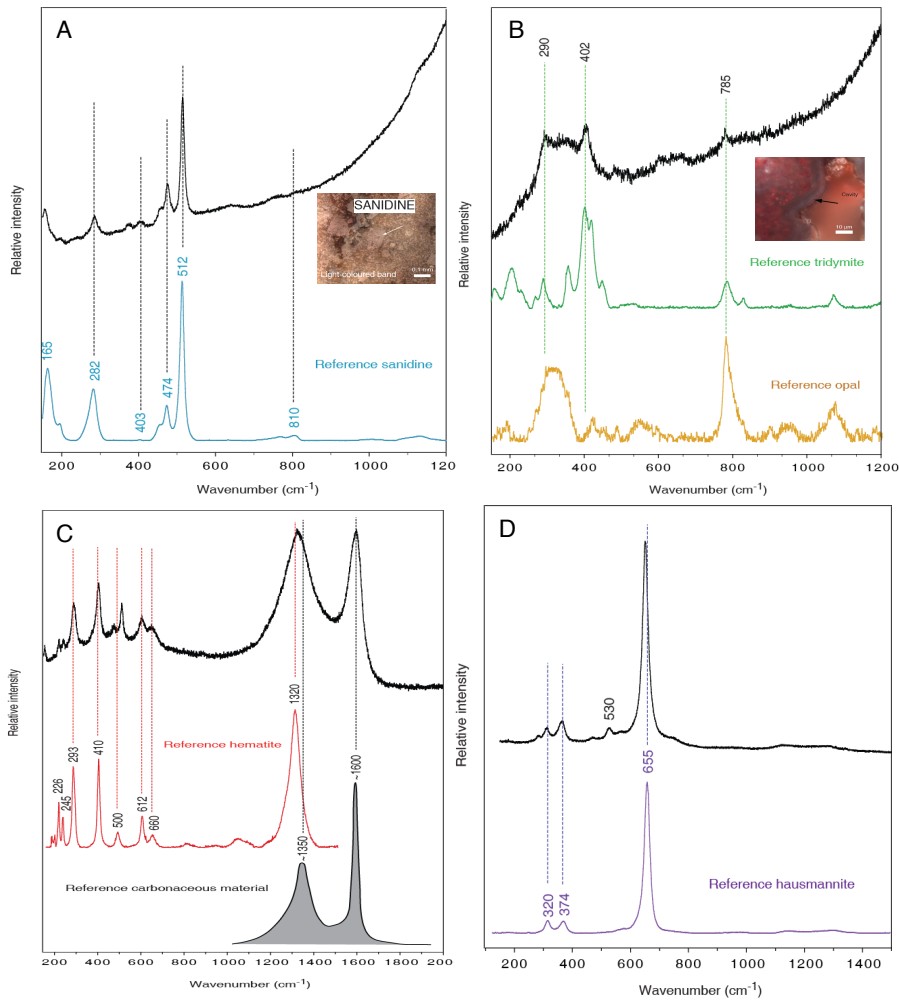


Fig. 10. Raman spectroscopy of the Fe- and/or Si-rich bands from NFIF.





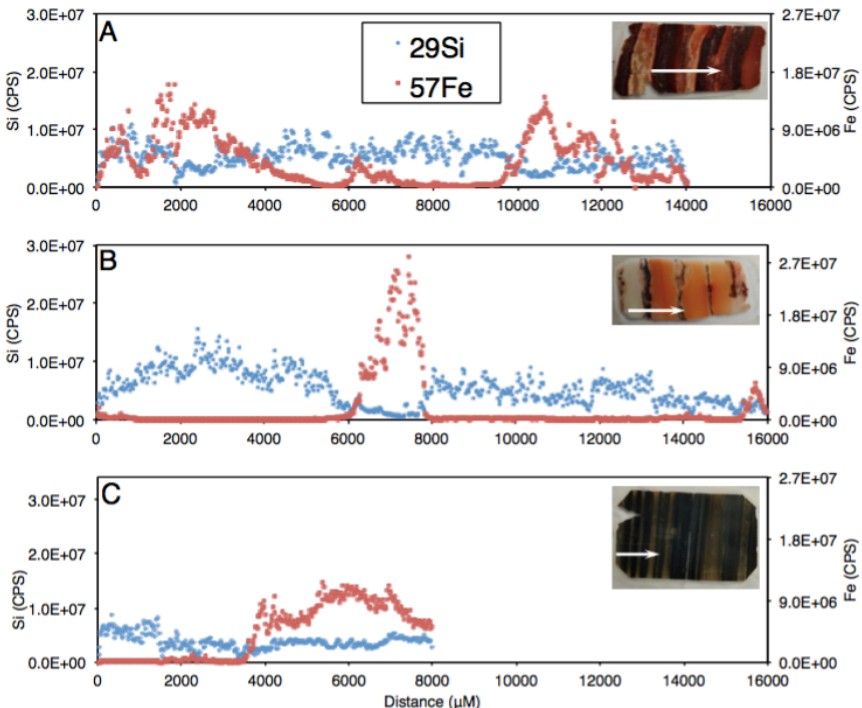

Fig. 11. Fluctuation in Si and Fe content measured by in situ laser ablation ICP-MS analysis. (A), Milos BIF-type rock with evenly distributed Si and iron rich bands. (B), Milos BIF type rock with large Si bands (whitish-brownish strips) and narrow Fe-rich bands (dark strips). (C), An example for the 2.5 Ga Kuruman BIF. Insets are analyzed thin sections. For scale, each thin section is ≈3.3 cm long. White arrow on thin section indicates analyzed area.





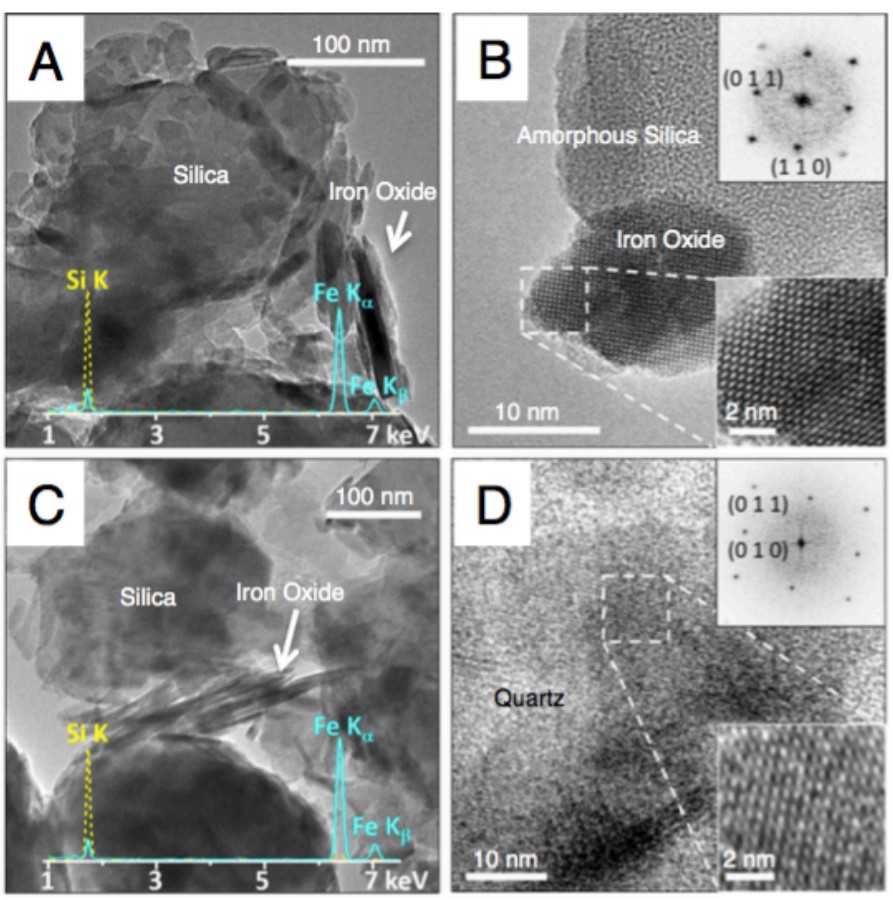


Fig. 12. TEM characterization of an NFIF and MFIF specimen. (A) lower
magnification MFIF TEM-BF image. (B) High resolution images of NFIF showing
amorphous Si and iron oxide crystallline lattice structures. Insets highlight a hematite
particle viewed from the [1-11] axis (Rhombohedral lattice). (C) Lower magnification
MFIF TEM-BF image. (D) High resolution images of MFIF showing crystalline
quartz and iron oxide crystallline lattice structures. Insets in (D) show a quartz crystal
viewed from the [100] axis. Both samples contain silica with a few hundred nm
particle size, and smaller needle-like iron oxide particles. Spectral lines in panels (A)
and (C) are X-ray Energy Dispersive elemental profiles of the individual Fe and Si
mineral phases.





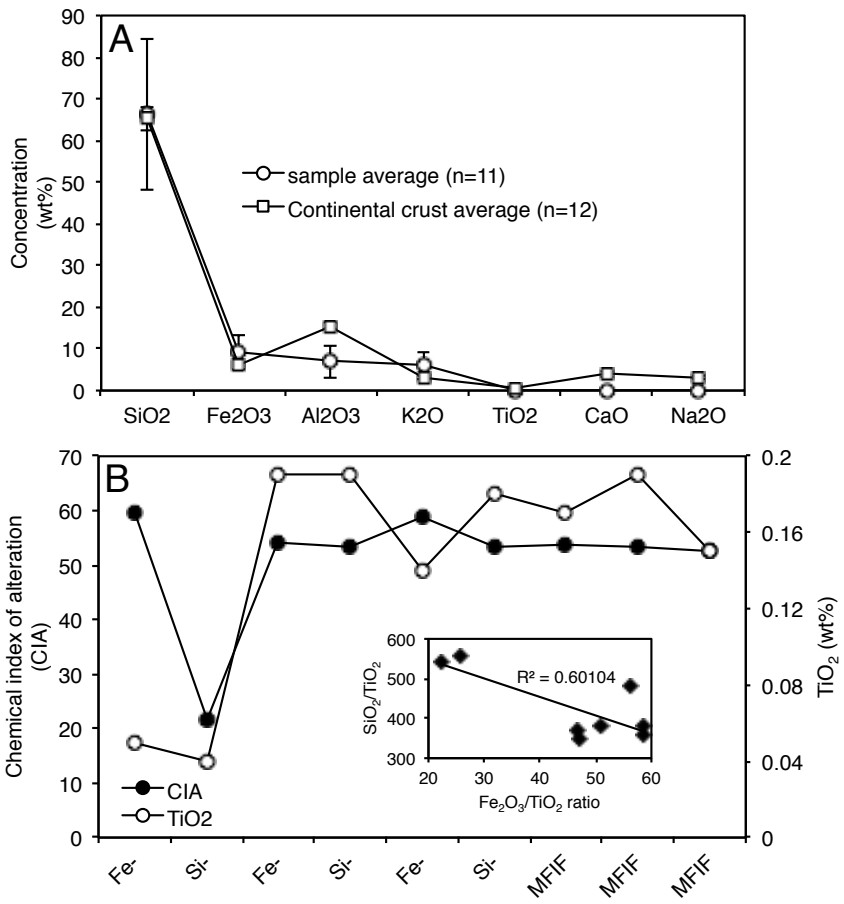


Fig. 13. Bulk average concentrations of major trace elements and chemical
weathering indices. (A), Relationship between average major trace element content
and average continental crust (Rudnick and Gao, 2003). (B), Chemical index of
alteration (CIA). Inset, relationship between $SiO_2$ and $Fe_2O_3$.


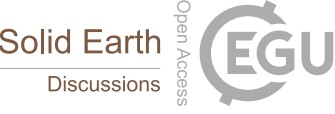



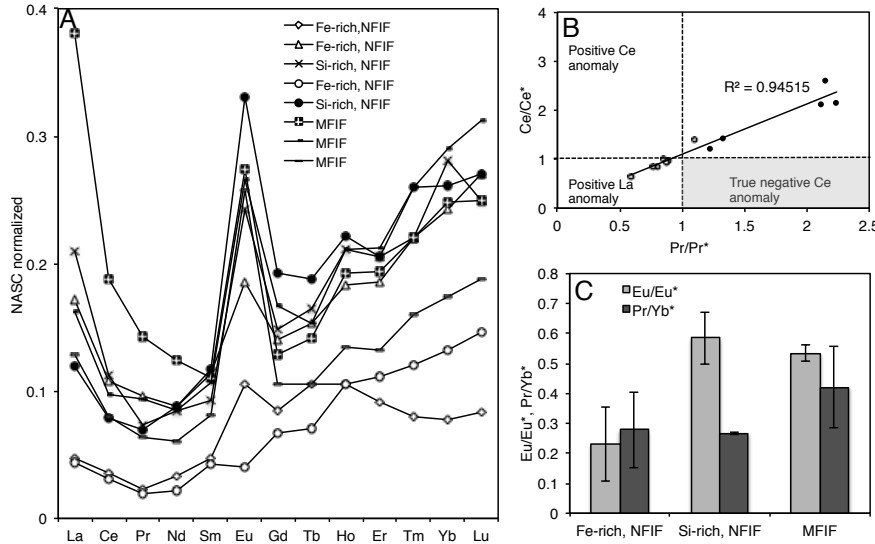


Fig. 14. Rare Earth Element (REE) distribution in samples and calculated Ce and Eu
anomalies. (A), NASC normalized REE distribution in various rock facies. (b), Ce
and Eu anomalies. (C), Eu anomalies and light REE (LREE) vs. heavy REE (HREE).




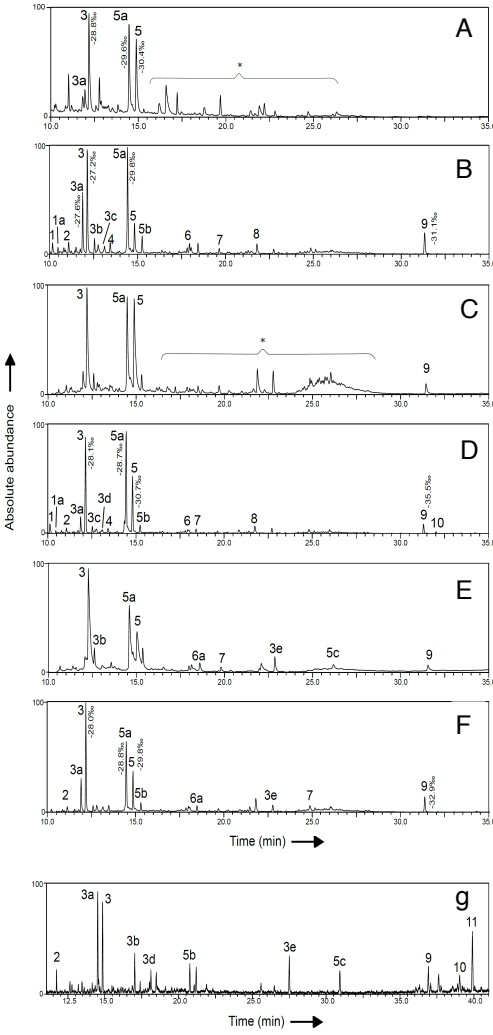


Fig. 15. GC/MS chromatogram sections of total lipid extracts of the BIF (A-F) for
bands excised from the sawn rock in Figure 2c. Panel G illustrates a total lipid extract
of modern sediment from the Milos basin. Values beside peaks indicate the lipid $\delta^{13}C$
values due to the low intensity of the lipids recovered this was not possible for all
peaks. Peaks have been annotated as the following compounds (FAME = fatty acid
methyl ester, Me = methyl group, TMS = trimethylsilyl, TMSE = trimethylsilyl ester):
(1) $C_{14:0}$ FAME, (1a) $C_{14:0}$ 13Me FAME, (2) $C_{15:0}$ FAME, (3) $C_{16:0}$ FAME, (3a) $C_{16:9}$
FAME, (3b) C16:0 TMS, (3c) 10Me $C_{16:0}$ FAME, (3d) $C_{16:9}$ FAME, (3e) $C_{16:0}$ TMSE,
(4) $C_{17:0}$ TMS, (5) $C_{18:0}$ FAME, (5a) $C_{18:9}$ FAME, (5b) $C_{18:0}$ TMS, (5c) $C_{18:0}$ TMSE, (6)



$C_{19:0}$ FAME, (6a) $C_{19:0}$ 18Me TMS, (7) $C_{21:0}$ TMS, (8) $C_{22:0}$ TMS, (9) Cholesterol
TMS, (10) Stigmasterol TMS, (11) beta-Sitosterol (*) Contaminants e.g. phthalates.



Fig. 16. Conceptual model showing the mechanism of band formation in the NFIF
related to changes in the intensity of hydrothermal activity and chemical oxidation of
Fe(II) to Fe(III) in the water column, inferred directly from our data. See Chi Fru et
al. (2013) for a biological model for the formation of the MFIF.