# Peer review of "Sedimentary mechanisms of a modern banded iron formation on"

_Solid Earth, 2017_

## Referee Comment (RC1) · A.J.B. Smith (Referee) · 5 Dec 2017

2017/12/05 Dear Editor – Solid Earth

This letter serves to summarize my review of the manuscript entitled "Sedimentary mechanisms of a modern banded iron formation on Milos Island, Greece", submitted by E Chi Fru et al. for possible publication in Solid Earth.

The manuscript documents Quarternary Fe-rich chemical sediments from the Cape Vani sedimentary basin (CVSB) on Milos Island, Greece. The Fe-rich units show a close associated with Mn-rich units and two subtypes were identified: i) microfossil-rich iron formation (IF); and ii) non-fossiliferous IF. The IFs also occur over a limited lateral extent. Geochemical and mineralogical data suggest that these units are very

similar to Precambrian IFs and could potentially be proxies to the latter. Depositional conditions are also proposed by the authors, which include tectonics, biological activity, changing redox and abiotic Si precipitation. Although there is some overlap with their previous publications in Nature Communications (2013) and Geobiology (2015), the authors state that a major new addition is presenting plausible mechanisms for the temporal and spatial separation between Fe and Mn deposition in the CVSB.

It must be noted upfront that this manuscript covers a fascinating and important geological occurrence, namely a Quarternary, spatially limited IF. With the majority of IF deposited during the Precambrian and, more specifically, prior to 2 Ga, this occurrence shows how unique and isolated conditions can drastically influence localized geology. Further documentation and more field descriptions of this occurrence is therefore always welcome in the literature. The authors should also be commended on the level into which they attempt to present a depositional model for the units, something which is often not even attempted in manuscripts on IFs. The inclusions of the importance of faulting and tectonics is also a great addition into the model, something which often cannot be done in detail on older, Precambrian IF occurrences. The authors also take special care to address a multitude of possible paleoenvironmental conditions, looking at the past and comparing it to the present, and should be complimented on such an even-handed approach. The conclusion on the cause for the banding (main conclusion 5) being caused by episodic hydrothermal intensification is also an important one, and one that I believe is also supported by evidence from Precambrian IFs.

My comments and corrections for the manuscript are presented as: main comments (numbered); section specific comments (bulleted); and detailed comments in a pdf copy of the manuscript. Some repetition will be present as the pdf copy contains all of the comments below to a certain degree in addition to more detailed comments such as minor content, spelling and grammar corrections. The authors must also take note that they need to activate the comment tool in Adobe Acrobat to see all the detailed comments in the pdf copy. The eight most important points that I believe require additions or adjustments to this manuscript are the following: 1. There is inconsistent use of element names and symbols in the manuscript (e.g. line 233 uses "iron" and "Mn" in the same line). The authors should be consistent in their use of either names or symbols for elements. Also, the use of hyphens are also inconsistent and should carefully be revised and updated to ensure format and spelling consistency across the manuscript. 2. It is important that the authors include, even if briefly, the detail on how the age constraints of the CVSB were determined somewhere in the geological setting of the manuscript. 3. The results section on geochemistry (section 3.3; line 461 onwards) contains a lot of interspersed petrography and mineralogy, which I believe is inappropriate. I strongly recommend that the two sets of results be clearly separated and presented in their own result sections. This will also require some reshuffling and/or re-editing of figures to properly fit the order and flow of the revised sections. 4. The use of North American Shale Composite (NASC) to normalize the REE data is strange and a bit dated. Most new publications on IFs from the mid-1990s onwards use Post-Archean Australian Shale (PAAS) for IF REE normalization. I would recommend the authors rather use this standard as it would make the data more comparable to other IF publications. In addition, the assessment of true Ce anomalies presented in this paper is also based on Bau and Dulski (1996), wherein they used PAAS as the shale standard for normalization. The Ce anomalies therefore need to be recalculated and replotted using PAAS. I do realize this might not make much of a difference, but for comparative purposes and for accuracy relating to the original publication this should be done. To make the REE assessment in this paper even more complete, the authors should also plot YSN in the REE diagram between Dy and Ho (REY plots) as proposed by Bau and Dulski (1996) to get a more complete pattern and assessment of the REE trends 5. The conclusion that the IFs were deposited in reducing conditions in a redox stratified sea/ocean, as indicated in figure 16 and stated in lines 685 to 687, is not currently convincing for the following reasons: 1) The lack of a Ce anomaly is not a definite indicator of reducing conditions. This can be buffered by excess Fe(II) in the system (see classic Eh-Ph diagrams by Brookins as well as Smith et al., 2013, Economic Geology, v. 108: 111-134). 2) The Ce anomaly calculations appear to have been done using NASC-normalized REE data instead of PAAS-normalized data. Proposed mechanisms for deeper water anoxia is left for very late in the manuscript, hinging on modern analogues (section 4.2.5), making the lead up at times unconvincing. Some of these arguments are also based on the interpretation of geochemical evidence to suggest anoxia, when micro-oxic conditions, in my opinion, cannot be completely ruled out. A more even handed approach regarding the Eh conditions throughout the manuscript, and taking into account the redox buffering effect between Fe2+ and Mn2+, is likely required throughout. Micro-aerophilic bacteria also likely played a larger role than currently suggested by the authors when considering how recently deposition occurred and how deep the water could have been (well below wave base). I do believe the authors have generally done a good job regarding Eh conditions in their depositional model and that all the necessary information is throughout the manuscript, but in my opinion they are only 70% there in making it convincing and even handed. 6. The formation of the granular Fe-rich beds (lines 456-459) is not satisfactorily addressed. There has to be morphological evidence within the granules for the authors to be able to commit to either a sedimentary or supergene formational mechanism. 7. To me, there is some confusion regarding deep ocean anoxia relative to hydrothermal venting in the depositional model (lines 995-999). This needs to be clarified in the discussion and lead up. Was deep water anoxia semi-permanent or did the venting play a role in establishing it? Here it appears that anoxia was semi-permanent, but the motivation needs to be better conveyed in the lead-up. 8. In the final conclusion (lines 1013-1017) the authors state that "Whether the rocks described here are analogues of Precambrian BIFs or not, and whether the proposed formation mechanisms match those that formed the ancient rocks, is opened to debate." The work here have many similarities to proposed Precambrian BIF depositional models (e.g. Smith et al., 2013; Bekker et al., 2010 and depositional models by Klein and Beukes, Beukes and Gutzmer). The authors should comment on this briefly.

Below follows my general, section-specific comments. Abstract • The sentence running from lines 37 to 38 should be rephrased. As it is currently written it does not clearly convey the stratigraphic relationship between the Fe- and Mn-rich units. Is the transition to the Mn-rich formation upwards or downwards? I think rephrasing this as two sentences briefly providing the bottom-up stratigraphy would clear this up. • The statements in lines 38 to 41 relating to anoxia might not be completely accurate. Refer to point 6 above. • The summary of depositional conditions in lines 46 to 48 is too vague and brief. The authors need to take a few more lines and properly summarize their proposed depositional model. Introduction The introduction and geological setting is brief, concise and appropriate. Here are some comments/corrections: • The author should start the introduction with a one sentence definition of IFs, mentioning their normal age distribution, and moving the references in line 63 and 64 to the end of the definition sentence. In the current form, the references in the middle of the first sentence clutters it and takes away from the impact of this sentence. If it reads better, the authors can also place the IF definition after the current first sentence. • For line 84, a good reference to add would be Beukes et al., 2016, Episodes v. 39: 285-317. It reviews all the Mn deposits of Africa. • More information needs to be provided on how the age range of the CVSB was determined. • Some minor comments and corrections are noted in the pdf copy. Make sure to address these too. Methodology: The following comments related to section 2 need to be addressed: • It should be stated very early on in the methodology section how many samples were taken, what sample types (lithologies) were taken and the approximate localities of the samples. • The sample preparation subsection (lines 119-125) is insufficient. More detail needs to be provided on how samples were taken, and how they were prepared for the different analyses. The type of mill used is not even mentioned. To refer to previous papers in the manner done here for sample preparation is not sufficient. • Lines 120-121: I am not sure how this fits under the subheading of sample preparation. This will need to be moved and better explained. So yes, it does form the basis for facies analysis, but what was actually done and how? • Raman spectroscopy (lines 132-138 should not be included under the XRD subsection heading (section 2.2.1). • Line 172: The

abbreviations ICP-ES/MS and XRD have not been defined. Also, AcmeLabs has had its name changed to Bureau Veritas. Where is the lab situated? • Lines 226 to 227: To use North American Shale Composite (NASC) for normalization of BIF data is an older way of doing it. Most of the newer publications use Post-Archean Australian Shale (PAAS). The authors should consider rather normalizing to PAAS for better comparison to recent literature. Results: Here are the main points that need to be addressed in section 3: • Lines 266 to 268 contains repetition from what was stated in line 264. The authors should combine and clean this up to remove the repetition. • Line 292: The term "Sh beds" has not been previously defined. What does it mean? • Line 298: The term "Gcm" has not been previously defined. What does it mean? • Lines 398-400: The bracket started in line 398 is never properly closed. Please do so. • Lines 431-434: The phrase "from a relatively shallow and deeper water setting. . . to a relatively deeper quiet water environment" appears to be contradictory at the start. Some rephrasing is required to improve clarity. • Lines 456-459: The granules associated with supergene formation versus those associated with sedimentary reworking will have very different internal structures. Whole rock geochemistry will also be very different. Why is such data not available? These are two very different formational proposals that should be resolved. • Lines 467-469: The way this sentence is phrased does not read accurately. Why do dramatic fluctuations control the Fe to Si ratio? Seems to be overly obvious and at the same time not supported by evidence, as if the lines between correlation and causality are being blurred. I would strongly recommend that this statement gets rephrased to more accurately represent what the data is actually showing, namely an inverse correlation between Fe and Si, which is to be expected when the two components are the only two major ones in the rock! • Lines 520-523: Here the normalization to NASC becomes problematic. The paper that originally presented to calculations and plot to assess true Ce anomalies (Bau and Dulski, 1996), used PAAS as their shale standard. All the calculations in that paper therefore used PAAS-normalized values and NASC. This alone likely justified why the authors should redo the REE data normalized to PAAS. • Lines 522-523: Depending on the

journal's citation format, shouldn't "Bau et al." be "Bau and Dulski"? Discussion Section 4.1 is generally well-written and creates a good depositional framework. Please note, however, some comment below that need to be addressed with regards to this sub-section. Here are the main points that need to be addressed in section 4: • Lines 604-606: I do not believe the conclusion here is completely accurate. One can only state the MFIF deposition preceded the second-stage Mn mineralization. There is no clear evidence that the MFIF and first-stage Mn mineralization was coeval. • Lines 614-615: Please rephrase this so that the clarity is improved. • Lines 615-616: "This uplifting into shallower water event"... Which event is this? The discussion preceding this statement in this paragraph has only been referring to a transgressive (i.e. deep-ening) event. Please rewrite and restructure where necessary to make this paragraph read better and make more sense. • Lines 651-655: Break this sentence into two shorter ones and do some restructuring that it reads better and the content is clearer. • Lines 663-672: This starting sentence for this paragraph reads like it comes out of nowhere. The authors need to lead into the content of this paragraph much better. Also, I am not convinced that this paragraph belongs in the subsection, as it does not directly relate to mineral paragenesis and also reads like it comes out of nowhere. • Lines 685-687: This statement has a few potential problems. Firstly, the lack of a Ce anomaly is not a definite indicator of reducing conditions. This can be buffered by ex-cess $Fe^{2+}$ in the system. Secondly, the Ce anomaly calculations appear to have been done using NASC-normalized REE data instead of PAAS-normalized data. At the very least the latter issue has to be resolved, and thereafter a more convincing argument need to be provided for reducing conditions. • Lines 697-699: This one line is not convincing enough as a mechanism for redox stratified depositional environment. I ad-mit that better and more convincing mechanisms are discussed later in the manuscript, but here the line comes over as unconvincing. Maybe note that possible mechanism for redox stratification are discussed in more details later. • Lines 722-729: I am not following the argument in point 3 very well. It requires some rephrasing and rewriting to convey the argument more clearly and concisely. • Lines 722: The authors need to

re-evaluate all the statements related to Ce anomalies after recalculating the anomalies to PAAS. • Lines 730-738: For point 4 as well, the argument is not coming through clearly. There also appears to be some structuring and grammatical problems in this paragraph. From what I can follow the argument is probably sound, but I cannot be sure as the paragraph is not well written. • Lines 806-809: Many authors agree with the statement that the lack of organic carbon is not due to metamorphism, but for a different reason. The organic matter is also likely destroyed in Precambrian BIF in a redox reaction with Fe3+, leading to the formation of 13C-depleted siderite and ankerite. This should be briefly addressed. See, for example, Smith et al. (2013, Economic Geology, v. 108:111-134) and references therein. • Lines 832-838: The tectonic and sea level mechanisms for changing redox conditions seem plausible. However, it only truly works when the motivation and mechanism for anoxia during IF deposition are more convincing. See main comment nr 6 above. • Lines 851-853: For interest, also see Smith et al. (2013, Economic Geology, v. 108: 111-134). • Lines 857-858: Rephrase and fix the grammar in this sentence. • Line 876-877: The statement "most likely dependent on prevailing redox conditions" is not yet convincing! The accumulation of Mn could also be buffered by the availability of Fe2+. Conclusions • Lines 990-994: Agreed that this is a feasible mechanism. However, Prevalence of available Fe2+ as a redox buffer should also be considered and addressed. • Lines 995-999: This needs to be clarified in the discussion and lead up. Was deep water anoxia semi-permanent or did the venting play a role in establishing it? Here it appears that anoxia was semi-permanent, but the motivation needs to be better conveyed in the lead-up. • Lines 998-999: What about the chemolithoautotrophs (microaerophilic bacteria)? I am not convinced that one can commit to only the photoferrotrophs with the dataset presented here. Figures • Figure 13: The major element oxides on the x-axis of part A should have their numbers changed to subscripts. • Figure 14: The y-axis of part A should be logarithmic! This is the norm for REE spider diagrams and makes the pattern more comparable to published REE data on IFs. • Figure 14: Newer publications normalize the REE data to PAAS. Part B of the figure, which aims to delineate true Ce anomalies, was also developed based on PAAS normalization (Bau and Dulski, 1996). I therefore think it would be better to normalize the REE data to PAAS so that better comparisons can be made to literature. Y, normalized to PAAS, should also be included between Dy and Ho (after Bau and Dulski, 1996). The Bau and Dulski (1996) reference should also be added to the caption for figure 14 B.

More detailed comments and minor corrections can be found in a pdf copy of the manuscript that should be made available to the authors. All these minor comments should also be addressed for the next version of the manuscript.

My recommendation as reviewer is that following what I consider moderate revisions and addressing all the issues raised in this report, that the paper be accepted for publication in Solid Earth.

Regards,

Bertus Smith Senior Lecturer Department of Geology University of Johannesburg

Please also note the supplement to this comment:
https://www.solid-earth-discuss.net/se-2017-113/se-2017-113-RC1-supplement.pdf

---

## Referee Comment (RC2) · Anonymous Referee #2 · 24 Jan 2018

Review of: Modern Banded Iron Sedimentary Rocks on Milos Island, Greece

The manuscript is well written, though it does suffer from a smattering of grammatical mistakes. The layers under study do seem somewhat similar to Precambrian BIFs and are thus worth investigation, though it must always be emphasized that Precambrian ocean chemistry was very different than today's seas. I have not directed many comments at the iron deposits themselves as their description and interpretation is reasonable. This review concentrates mostly on the sedimentological aspects of the manuscript, which are problematic. Basically not enough substantiating data is provided for the interpretations given. Many of the interpretations are very specific and the limited exposures available do not provide the types of data necessary to validate the interpretations. The author's interpretations, in general, could be correct, but there are

also other equally as valid interpretations of the depositional systems possible. This situation is not helped by the inclusion of references after an interpretation is put forward that describe a depositional process or rock unit that was formed in a similar environment to that proposed but appear to bear little in common with the rocks present in this study. This reduces down to the problem that the characteristics of the rocks described in this study are not detailed enough to support specific interpretations. For example: a conglomeratic unit is interpreted as a channelized mass flow deposit in a submarine fan. If it was deposited by a high-density turbidity current it will have certain internal characteristics that are well defined in the literature (see some of Walker and Lowe's older papers). If it was a debris flow it will have other characteristics, such as disorganized clast orientations, matrix support, poor sorting etc., that these conglomerates do not appear to have. However, there is an even bigger problem with this interpretation. Submarine fan channel successions form thick fining upwards successions, commonly over tens of meters vertically. Finally submarine fans are one category of submarine base of slope deposit, a group that also includes ramps and aprons, and no evidence is given why this would not be a ramp or an apron, or simply, and much more likely, a conglomerate bed. I put in the latter as a few conglomerate beds do not make a fan, ramp or apron, which are very much larger features. These are just the problems that exist with one interpretation of depositional environments. Similar problems exist with the others.

It would have been beneficial if the authors delegated more discussion to the deposition of the silica-rich layers as the Fe-rich layers forming from hydrothermal fluids are easy to understand but the deposition of the silica layers in BIF is much more difficult to explain. The use of references is perplexing. Most of them do not have direct bearing on what they are referencing in the text. They are on the same general subject, but many do not reinforce the correctness of the preceding statement.

I recommend that the interpretations of the depositional environments of the siliciclastics be eliminated. They are very problematic and greatly distract from the manuscript.

The description and discussion of the IF can stand alone. Its lack of current formed structures implies a low energy environment and that is about all that can be inferred about physical processes from the IF. Thus, the manuscript would need major revisions.

A more detailed line by line review follows:

Line 73: Rare Earth Elements should not be capitalized. Line 115: Rhyolite is not intrusive. Line 226: It is much more common in work on iron formations to us PAAS to normalize the data. Line 327: Below storm wave base does not necessarily mean below 100 to 200 meters. At present storm wave penetration is deepest in locations such as southeastern Australia and Atlantic Canada where it reaches 120m. But these are very storm prone open ocean facing areas. It is difficult to give an estimate for paleo-storm wave base in the study area, but I doubt that it could be even close to 100m as more would mean waves with greater than 200 meter wavelengths in a sheltered area compared to the open Atlantic. Line 340: Not enough evidence is given to justify the turbidite interpretation. Graded beds just mean they were deposited by powering-down events, which can occur in many different environments. Even if they are turbidites, which I have no idea whether they are or not from the evidence, the setting cannot be termed a fan, ie, why not a ramp or apron or a number of other environments that can have turbidites. Line 344: Slump deposits infers an intact or partially intact block that slid. The conglomerates are not slump deposits. They could be debris flows, but again there is not enough evidence given to say this. Line 347: If a flow is carrying pebbles it is not a low density turbidity current. Line 372: gravel to pebble is not proper terminology. Pebbles are gravel if unconsolidated. Line 395: mm-scale layers are not beds, they are laminae Line 418: Why not below storm wave-base? Line 422: The only evidence for the interpretation that the conglomerates are " a series of channel deposits in an inner turbidite fan-like setting" appears to be that they are conglomerates. A great deal more evidence is necessary to be so specific about the depositional environment. Line 424: No evidence has been given for a tidal environment and little evidence for a shoreface. Line 427: There are also many papers that

describe iron formations in other settings. Line 434: This is an example of a reference that has little bearing on the preceding statement. The Mesoarchean Barberton is not a good analog for the sedimentary environment of the basin described here. Line 439: The sedimentary structures described could have been formed in the environments proposed, but they are not limited to the environments given the lack of evidence. Line 458: The description of these deposits has little in common with GIF. It is also better to reference the originator of the term GIF (Simonson), rather than Bekker, which is just a review article. Line 481: This is circular reasoning. Line 482: Precambrian BIF can be sulfide facies. Line 522: This statement is not correct. Planavsky and others (see authors' references to this statement) put forward that the anomaly for Ce must be less than .95 and greater than 1.05 to be significant, not less than or greater than 1. Line 528: They do not have similar enrichment levels; they are light depleted. Line 574: The positive Eu anomalies are quite small compared to those associated with oceanic hydrothermal vent sediments. Also, volcanic detritus can carry positive Eu anomalies. A plot of Ti vrs Eu* would be useful to distinguish if the anomaly is related to volcanic detritus in the IF. Line 614: What is described as an upward fining trend appears to me to be simply one single graded bed. The fining upwards in the bed is better explained by the depositional mechanism losing energy through time. Also, conglomeratic beds usually represent rapid deposition during a high energy event, ie. storm or mass flow, rather than the slow pebble on pebble accumulation over years. Line 682: Comparing the small Eu anomalies present in this study with the larger Precambrian anomalies should include giving the values for the average Precambrian anomalies. Simply stating the values of Eu anomalies of samples in this study are more similar to Archean anomalies is somewhat misleading. Line 757: If even small amounts of seawater are mixing with the hydrothermal fluid, as previously stated, anoxia could not exist. Line 842: The presence of a conglomeratic bed does not commonly mean deepening of a marine succession. There are literally thousands of papers where the upward transition of sandstones to conglomerates is interpreted as shallowing as energy levels increase with shallowing in a marine setting. Line 848: The presence of a

transgressive conglomeratic lag implies that the area was emergent prior to this and the conglomerate formed by wave reworking in a shore proximal environment. Evidence has not been given to support this, and if I am not mistaken the conglomerate has previously in this manuscript been interpreted as a mass flow. Line 853: In these references the maximum regressive surface is overlain by a transgressive lag and then very shallow shoreline deposits affected by wave activity. A very different scenario to what these authors are proposing. Line 855: The referenced BIFs are not deposited in sandstone/grainstone environments, the IFs are grainstone with very low siliciclastic contents and they are interlayered with chemical muds, but the IFs are not banded. Line 1004: This process would be expected to produce a sharp bottom contact to the Fe-rich layer, which would then mineralogical grade upwards into the silica-rich layer. Is this the way the layers are organized?

---

## Author Comment (AC1) · 12 Mar 2018

Anonymous Referee #2 Review of: Modern Banded Iron Sedimentary Rocks on Milos Island, Greece.

Comments The manuscript is well written, though it does suffer from a smattering of grammatical mistakes.

Responses: We extend thankfulness to this reviewer for taking time off to read and comment on our manuscript. His comments have condensed the sedimentology, tightening up loose parts suffering from lack of clarity. To satisfy the comment about a smattering of grammatical errors, we have taken care to minimize errors that might previously have escaped careful editing. Attached to this document is a PDF file named, supplement, containing the manuscript with changes, including those requested by the reviews, in red.

The layers under study do seem somewhat similar to Precambrian BIFs and are thus worth investigation, though it must always be emphasized that Precambrian ocean chemistry was very different than today's seas. I have not directed many comments at the iron deposits themselves as their description and interpretation is reasonable. This review concentrates mostly on the sedimentological aspects of the manuscript, which are problematic. Basically not enough substantiating data is provided for the interpretations given. Many of the interpretations are very specific and the limited exposures available do not provide the types of data necessary to validate the interpretations. The author's interpretations, in general, could be correct, but there are other equally as valid interpretations of the depositional systems possible. This situation is not helped by the inclusion of references after an interpretation is put forward that describe a depositional process or rock unit that was formed in a similar environment to that proposed but appear to bear little in common with the rocks present in this study. This reduces down to the problem that the characteristics of the rocks described in this study are not detailed enough to support specific interpretations. For example: a conglomeratic unit is interpreted as a channelized mass flow deposit in a submarine fan. If it was deposited by a high-density turbidity current it will have certain internal characteristics that are well defined in the literature (see some of Walker and Lowe's older papers). If it was a debris flow it will have other characteristics, such as disorganized clast orientations, matrix support, poor sorting etc. that these conglomerates do not appear to have. However, there is an even bigger problem with this interpretation. Submarine fan channel successions form thick fining upwards successions, commonly over tens of meters vertically. Finally submarine fans are one category of submarine base of slope deposit, a group that also includes ramps and aprons, and no evidence is given why this would not be a ramp or an apron, or simply, and much more likely, a conglomerate bed. I put in the latter as a few conglomerate beds do not make a fan, ramp or apron, which are very much larger features. These are just the problems that exist with one interpretation of depositional environments. Similar problems exist with the others. It would have been beneficial if the authors delegated more discussion to the deposition of the silica-rich layers as the Fe-rich layers forming from hydrothermal fluids are easy to understand but the deposition of the silica layers in BIF is much more difficult to explain. The use of references is perplexing. Most of them do not have direct bearing on what they are referencing in the text. They are on the same general subject, but many do not reinforce the correctness of the preceding statement. I recommend that the interpretations of the depositional environments of the siliciclastics be eliminated. They are very problematic and greatly distract from the manuscript.

Response: For the sake of clarity, we should indicate that we are not proposing the Milos BIF-type rocks as the exact equivalent of Precambrian BIFs or insinuating that the seawater in which they formed had the exact composition of the Precambrian oceans. But we have found several components of the deposit that have the potential of providing and aiding mechanistic models aimed at understanding how BIFs formed. They may give new insights into the deep past from the present-day seawater biogeochemical perspective. These are some of the challenges we wish to resolve by detailed description of the geological and geochemical processes behind the perplexing deposition of the Milos IF. To enable comparison, we use the simple definition of BIFs as marine sedimentary rocks with alternating layers of Fe-rich and Si-rich bands, containing at least 15% Fe. This definition does not restrict the potential for BIFs to form only in Precambrian oceans, although they are a major feature of this unique period, a time when seawater had extraordinarily high levels of dissolved Fe and Si.

What our data are showing is that these local conditions of elevated and cyclic supply of dissolved Fe and Si and accompanied by strict bottom water anoxic conditions in a localised reservoir cut off from the open ocean, can in principle allow the rare deposition of BIF-type rocks in the modern ocean. The rarity of these types of deposits in the present-day ocean hints that such conditions seldom develop under the existing atmosphere, but that they can indeed occur. Therefore we present these as a rare modern

BIF-type facies, different from the Precambrian BIFs, in the same way the rare Neoproterozoic BIFs are different from the widespread Paleoproterozoic Superior BIFs, which are in turn distinct from the mainly Algoma-type Archean BIF deposits that are limited in scale. This paragraph has been edited and included in our conclusions to highlight the importance of distinguishing this deposit from the Precambrian formations.

We strongly agree that sedimentary features can be difficult to interpret with certainty. We have therefore reduced the degree to which these interpretations have been made for the above reasons laid down by this reviewer. We however believe that it is important to keep solid parts of the interpretations that help explain how anoxic conditions could have apparently developed in the CVSB to enable dissolved Fe enrichment and its oxidation to Fe(III). Further, sedimentology must not be interpreted independently from the geochemistry and redox. This has become even more crucial with the new Figure 13C-D that unambiguously supports the contentious anoxic depositional conditions previously illustrated by REEs. This new information has been acquired using the widely accepted iron extraction redox proxy (See Poulton, S.W. and Canfield, D.E. 2011. Ferruginous conditions: A dominant feature of the ocean through Earth's history. Elements 7, 107–112, for a review).

This reviewer indicates that it would have been more helpful to dedicate more time discussing the Si bands. Our data show that band formation was mainly controlled by the activity of Fe, while Si precipitation was a passive process that cannot be explored beyond the fact. As we have shown and discussed, it is the cyclic release and oxidation of ferrous Fe that in fact controls the enrichment of Fe in the Fe-rich bands and Si in the Si bands. This particular observation provides the first independent modern verification for similar processes suggested to have formed the ancient BIFs (See Bekker et al. 2010 for details and references therein, cited in the main text).

The description and discussion of the IF can stand alone. Its lack of current formed structures implies a low energy environment and that is about all that can be inferred about physical processes from the IF. Thus, the manuscript would need major revisions.

A more detailed line by line review follows:

Response: We agree with this reviewer about the low-energy environment in which the IF formed, which is in agreement with our initial conclusions. However, it would appear that the switch from one redox state to the other was often accompanied by tectonic activity that caused deepening and shallowing.

Line 73: Rare Earth Elements should not be capitalized.

Response: Corrected

Line 115: Rhyolite is not intrusive.

Response: Corrected to extrusive

Line 226: It is much more common in work on iron formations to us PAAS to normalize the data.

Response: The rationale for using the NASC is as follows:

1. The NASC normalization maintains data consistency with the REE data published in our previous papers on the Milos IF ((1) Chi Fru, E., Ivarsson, M., Kilias, S.P., Bengtson, S., Belivanova, V., Marone, F., Fortin, D., Broman, C., and Stampanoni, M.: Fossilized iron bacteria reveal a pathway to the origin banded iron formations. Nat. Comm., 4, 2050 DOI: 10.1038/ncomms3050, 2013. (2) Chi Fru, E., Ivarsson, M., Kilias, S.P., Frings, P.J., Hemmingsson, C., Broman, C., Bengtson, S. and Chatzitheodoridis, E.: Biogenicity of an Early Quaterny iron formation, Milos Island, Greece. Geobiology, 13, 225–44, 2015.

2. There are no scientifically demonstrated discrepancies between the PAAS and NASC.

3. Following the above suggestion, data was normalized to PAAS for comparison with the NASC normalized trends. The results produced the same trend as observed when data are normalized to NASC. See new Figure 14 in the manuscript text, accessible in the attached supplement PDF file. Further explanations are also provided under sections 2.6-2.6.1 in the manuscript supplement text. Line 327: Below storm wave base does not necessarily mean below 100 to 200 meters. At present storm wave penetration is deepest in locations such as southeastern Australia and Atlantic Canada where it reaches 120m. But these are very storm prone open ocean facing areas. It is difficult to give an estimate for paleo-storm wave base in the study area, but I doubt that it could be even close to 100m as more would mean waves with greater than 200 meter wavelengths in a sheltered area compared to the open Atlantic.

Response: We have omitted our attempt at specifying a value in the text for the depth. The new wording has been rephrased to:

The MFIF rests directly on the submarine dacites-andesites that were deposited in a relatively shallow submarine environment (Stewart and McPhie, 2006).

Line 340: Not enough evidence is given to justify the turbidite interpretation. Graded beds just mean they were deposited by powering-down events, which can occur in many different environments. Even if they are turbidites, which I have no idea whether they are or not from the evidence, the setting cannot be termed a fan, ie, why not a ramp or apron or a number of other environments that can have turbidites.

Response: Removed from text.

Line 344: Slump deposits infers an intact or partially intact block that slid. The conglomerates are not slump deposits. They could be debris flows, but again there is not enough evidence given to say this.

Response: Lines 337-361 have been deleted and replaced with this short paragraph:

In the overlying sandstone-conglomerate facies, the presence of sedimentary structures indicative of wave action and currents (e.g. cross-stratification), that signify rapid deposition during a high energy event, are consistent with a switch to a shallow-submarine high energy environment (Stewart and McPhie, 2006; Chi Fru et al., 2015).

This shift in depositional environments may have been controlled by a combination of submarine volcano-constructional processes, synvolcanic rifting and volcano-tectonic uplift known to have formed the CVSB (Papanikolaou et al., 1990; Stewart and McPhie, 2006).

Line 347: If a flow is carrying pebbles it is not a low density turbidity current.

Response: Deleted and replaced with the paragraph above.

372: gravel to pebble is not proper terminology. Pebbles are gravel if unconsolidated.

Response: Lines 370-373 have been deleted and replaced with:

The lower sandstone facies represents the host of the main economic grade Mn oxide ores in the CVSB. This constitutes part of a separate study devoted to the Mn ores and will not be dealt with further here.

Line 395: mm-scale layers are not beds, they are laminae.

Response: Deleted and replaced with laminations.

Line 418: Why not below storm wave-base?

Response: Corrected.

Line 422: The only evidence for the interpretation that the conglomerates are " a series of channel deposits in an inner turbidite fan-like setting" appears to be that they are conglomerates. A great deal more evidence is necessary to be so specific about the depositional environment.

Response: Lines 420-434 have been deleted.

Line 424: No evidence has been given for a tidal environment and little evidence for a shoreface.

Response: Lines 420-434 have been deleted.

Line 427: There are also many papers that describe iron formations in other settings.

Response: Lines 420-434 have been deleted.

Line 434: This is an example of a reference that has little bearing on the preceding statement. The Mesoarchean Barberton is not a good analog for the sedimentary environment of the basin described here.

Response: Lines 420-434 have been deleted.

Line 439: The sedimentary structures described could have been formed in the environments proposed, but they are not limited to the environments given the lack of evidence.

Response: Lines 335-340 deleted.

Line 458: The description of these deposits has little in common with GIF. It is also better to reference the originator of the term GIF (Simonson), rather than Bekker, which is just a review article.

Response: References to GIF have been deleted from the text.

Line 481: This is circular reasoning.

Response: We agree and further demonstrate this in the new Figure 13D. These emphases must be highlighted to show some of the similarities these rare deposit shares with true BIFs.

Line 482: Precambrian BIF can be sulfide facies.

Response: We agree. We are trying to make the statement that these are those type of BIF facies that are sulfide rich. We have therefore peplaced with the text:

Lack of association of the framboidal-iron-rich particles with S, following SEM-EDS analysis, rules out a pyrite affiliation and is consistent with the non-sulfidic depositional model suggested by the sequential iron extraction redox proxy (Fig. 13D).

Line 522: This statement is not correct. Planavsky and others (see authors' references to this statement) put forward that the anomaly for Ce must be less than .95 and greater than 1.05 to be significant, not less than or greater than 1.

Response: This has been corrected.

Line 528: They do not have similar enrichment levels; they are light depleted. Response: Enrichment changed to depleted.

Line 574: The positive Eu anomalies are quite small compared to those associated with oceanic hydrothermal vent sediments. Also, volcanic detritus can carry positive Eu anomalies. A plot of Ti vrs Eu* would be useful to distinguish if the anomaly is related to volcanic detritus in the IF.

Response: We agree, but this effort will not tell us anything more than what we have already shown, since multiple evidence shows that we are dealing with materials being released into the basin intermittently by hydrothermal/volcanic activity as demonstrated by the ash particles in the bands. As we have shown in Figure 13 and from using multiple lines of evidence, the supply of materials from the continent to the basin was not an important source of sediments during the formation of the alternating Fe and Si layers. Our main interpretation is a hydrothermal source, backed by data in our cited publication in Nature Communications and Geobiology, in addition to the present submission.

Line 614: What is described as an upward fining trend appears to me to be simply one single graded bed. The fining upwards in the bed is better explained by the depositional mechanism losing energy through time. Also, conglomeratic beds usually represent rapid deposition during a high energy event, ie. storm or mass flow, rather than the slow pebble on pebble accumulation over years.

Response: This text has been revised as suggested. The new text reads like this:

Geomorphological/chemical reconfiguration orchestrated the deposition of the NFIF in a deeper, small-restricted basin (Fig. 2). The deepening of Basin 3 is reflected in the underlying graded conglomerate bed that exhibits an upward fining trend, followed by transition into the fine-grained NFIF. The conglomerate bed may represent rapid deposition during a high-energy event, i.e. storm or mass flow, whereas the upwards fining in the bed is better explained by the depositional mechanism losing energy through time. These high-energy conditions apparently must have ceased during the deposition of the overlying NFIF, where we interpret that increased abundance of finely laminated IF and decreased evidence of storm and/or mass flow reworking reflects deepening conditions. The hypothesized deepening of Basin 3 is consistent with the interpretation that active rifting was an important mechanism in the formation of the CVSB (Papanikolaou et al., 1990).

Line 682: Comparing the small Eu anomalies present in this study with the larger Precambrian anomalies should include giving the values for the average Precambrian anomalies. Simply stating the values of Eu anomalies of samples in this study are more similar to Archean anomalies is somewhat misleading.

Response: The paragraph has been deleted.

Line 757: If even small amounts of seawater are mixing with the hydrothermal fluid, as previously stated, anoxia could not exist.

Response: See new Figure 13C-D that firmly establishes the anoxic/ferruginous depositional conditions. Moreover the statement made by this reviewer that even if small amounts of seawater are mixing with hydrothermal fluid, anoxia cannot exist, is misleading and a bit perplexing because this argument means that redox gradients should not exist in nature. Following the rules of stoichiometry in chemical reactions, large volumes of highly reduced solutions such as hydrothermal fluids require equally large concentrations of oxidants (especially oxygen) to make the fluid oxidizing. From this reasoning, considerable amounts of oxygen are required to react with the large volumes of the highly reduced chemicals and compounds present in hydrothermal fluids.

This argument is given as an explanation for why it took so long for oxygen to rise in the atmosphere (See Lyons et al., 2014. The rise of oxygen in Earth's early ocean and atmosphere. Nature 506:307–315: and the review by Bekker et al. 2010, cited in the manuscript). The reasoning is that reduced hydrothermal fluids that made up a bulk of the early oceans were eventually overwhelmed stoichiometrically by oxygen (meaning more oxygen was being produced than consumed by the reduced fluids), leading to the rise of oxygen in the atmosphere, c. 2.4 billion years. Even after that, although the Paleoproterozoic surface ocean was oxidized for close to two billion years, complete ocean oxygenation only came at the end of the Precambrian despite the fact that reduced deep ocean hydrothermal fluids continuously mixed with the oxygen-rich ocean surface seawater. If we were to follow the argument given by this reviewer, then the whole ocean would have been oxidized following the mixing of the reducing fluids with the thin layer of oxygen-rich seawater on the ocean surface. This indicates that the sedimentology, geochemistry and redox must be jointly interpreted to understand what occurred at Milos.

Line 842: The presence of a conglomeratic bed does not commonly mean deepening of a marine succession. There are literally thousands of papers where the upward transition of sandstones to conglomerates is interpreted as shallowing as energy levels increase with shallowing in a marine setting.

Response: We strongly agree that the paragraph was not well-phrased, leading to the difficulty in understanding the meaning of the sentence. It has now been revised to:

All of this is feasible with the three-basin-fault-bounded hypothesis as a requirement for movement along fault lines in response to temporal tectonic activation. The upward sequence transition from the Mn-rich sandstone facies, through the pebbly conglomerate and the final termination in the overlying mud-grained NFIF (Fig. 8B), reflect sedimentary features formed during multiple changes in seawater levels (Cattaneo & Steel, 2000). This study proposes that the NFIF that overlies the transgressive-type conglomeratic lag along an erosional contact surface was likely deposited during maximum flooding, when the basin became stagnant and stratified, and subsequently was uplifted to emergence.

Line 848: The presence of a transgressive conglomeratic lag implies that the area was emergent prior to this and the conglomerate formed by wave reworking in a shore proximal environment. Evidence has not been given to support this, and if I am not mistaken the conglomerate has previously in this manuscript been interpreted as a mass flow.

Response: Deleted.

Line 853: In these references the maximum regressive surface is overlain by a transgressive lag and then very shallow shoreline deposits affected by wave activity. A very different scenario to what these authors are proposing.

Response: Paragraph and references removed.

Line 855: The referenced BIFs are not deposited in sandstone/grainstone environments, the IFs are grainstone with very low siliciclastic contents and they are interlayered with chemical muds, but the IFs are not banded.

Response: Because this interpretation is not of immediate relevance to the strength of the paper, the paragraph has been deleted.

Line 1004: This process would be expected to produce a sharp bottom contact to the Fe-rich layer, which would then mineralogical grade upwards into the silica-rich layer. Is this the way the layers are organized?

Response: Yes. We show this in supplementary Figures 8 and 9.

Please also note the supplement to this comment:
https://www.solid-earth-discuss.net/se-2017-113/se-2017-113-AC1-supplement.pdf

[Figure]

**Supplement:**

Anonymous Referee #2 Review of: Modern Banded Iron Sedimentary Rocks on Milos Island, Greece.

Comments
The manuscript is well written, though it does suffer from a smattering of grammatical mistakes.

**Responses:** We extend thankfulness to this reviewer for taking time off to read and comment on our manuscript. His comments have condensed the sedimentology, tightening up loose parts suffering from lack of clarity. To satisfy the comment about a smattering of grammatical errors, we have taken care to minimize errors that might previously have escaped careful editing. Attached to this document is a PDF file named, supplement, containing the manuscript with changes, including those requested by the reviews, in red.

The layers under study do seem somewhat similar to Precambrian BIFs and are thus worth investigation, though it must always be emphasized that Precambrian ocean chemistry was very different than today's seas. I have not directed many comments at the iron deposits themselves as their description and interpretation is reasonable. This review concentrates mostly on the sedimentological aspects of the manuscript, which are problematic. Basically not enough substantiating data is provided for the interpretations given. Many of the interpretations are very specific and the limited exposures available do not provide the types of data necessary to validate the interpretations. The author's interpretations, in general, could be correct, but there are other equally as valid interpretations of the depositional systems possible. This situation is not helped by the inclusion of references after an interpretation is put forward that describe a depositional process or rock unit that was formed in a similar environment to that proposed but appear to bear little in common with the rocks present in this study. This reduces down to the problem that the characteristics of the rocks described in this study are not detailed enough to support specific interpretations. For example: a conglomeratic unit is interpreted as a channelized mass flow deposit in a submarine fan. If it was deposited by a high-density turbidity current it will have certain internal characteristics that are well defined in the literature (see some of Walker and Lowe's older papers). If it was a debris flow it will have other characteristics, such as disorganized clast orientations, matrix support, poor sorting etc. that these conglomerates do not appear to have. However, there is an even bigger problem with this interpretation. Submarine fan channel successions form thick fining upwards successions, commonly over tens of meters vertically. Finally submarine fans are one category of submarine base of slope deposit, a group that also includes ramps and aprons, and no evidence is given why this would not be a ramp or an apron, or simply, and much more likely, a conglomerate bed. I put in the latter as a few conglomerate beds do not make a fan, ramp or apron, which are very much larger features. These are just the problems that exist with one interpretation of depositional environments. Similar problems exist with the others. It would have been beneficial if the authors delegated more discussion to the deposition of the silica-rich layers as the Fe-rich layers forming from hydrothermal fluids are easy to understand but the deposition of the silica layers in BIF is much more difficult to explain. The use of references is perplexing. Most of them do not have direct bearing on what they are referencing in the text. They are on the same general subject, but many do not reinforce the correctness of the preceding statement. I recommend that the interpretations of the depositional environments of the siliciclastics be eliminated. They are very problematic and greatly distract from the manuscript.

**Response:** For the sake of clarity, we should indicate that we are not proposing the Milos BIF-type rocks as the exact equivalent of Precambrian BIFs or insinuating that the seawater in which they formed had the exact composition of the Precambrian oceans. But we have found several components of the deposit that have the potential of providing and aiding mechanistic models aimed at understanding how BIFs formed. They may give insights into the deep past from the present perspective? These are some of the challenges we wish to resolve by detailed description of the geological and geochemical processes behind the perplexing deposition of the Milos IF. To enable comparison, we use the simple definition of BIFs as marine sedimentary rocks with alternating layers of Fe-rich and Si-rich bands, containing at least 15% Fe. This definition does not restrict the potential for BIFs to form only in Precambrian oceans, although they are a major feature of this unique period, a time when seawater had extraordinarily high levels of dissolved Fe and Si.

What our data are showing is that these local conditions of elevated and cyclic supply of dissolved Fe and Si and accompanied by strict bottom water anoxic conditions in a localised reservoir cut off from the open ocean, can in principle allow the rare deposition of BIF-type rocks in the modern ocean. The rarity of these types of deposits in the present-day ocean hints that such conditions seldom develop under the existing atmosphere, but that they can indeed occur. Therefore we present these as a rare modern BIF-type facies, different from the Precambrian BIFs, in the same way the rare Neoproterozoic BIFs are different from the widespread Paleoproterozoic Superior BIFs, which are in turn distinct from the mainly Algoma-type Archean BIF deposits that are limited in scale. This paragraph has been edited and included in our conclusions to highlight the importance of distinguishing this deposit from the Precambrian formations.

We strongly agree that sedimentary features can be difficult to interpret with certainty. We have therefore reduced the degree to which these interpretations have been made for the above reasons laid down by this reviewer. We however believe that it is important to keep solid parts of the interpretations that help explain how anoxic conditions could have apparently developed in the CVSB to enable dissolved Fe enrichment and its oxidation to Fe(III). Further, sedimentology must not be interpreted independently from the geochemistry and redox. This has become even more crucial with the new Figure 13C-D that unambiguously supports the contentious anoxic depositional conditions previously illustrated by REEs. This new information has been acquired using the widely accepted iron extraction redox proxy (See *Poulton, S.W. and Canfield, D.E. 2011. Ferruginous conditions: A dominant feature of the ocean through Earth's history. Elements 7, 107–112, for a review).*

This reviewer indicates that it would have been more helpful to dedicate more time discussing the Si bands. Our data show that band formation was mainly controlled by the activity of Fe, while Si precipitation was a passive process that cannot be explored beyond the fact. As we have shown and discussed, it is the cyclic release and oxidation of ferrous Fe that in fact controls the enrichment of Fe in the Fe-rich bands and Si in the Si bands. This particular observation provides the first independent modern verification for similar processes suggested to have formed the ancient BIFs (See Bekker et al. 2010 for details and references therein, cited in the main text).

The description and discussion of the IF can stand alone. Its lack of current formed structures implies a low energy environment and that is about all that can be inferred about physical processes from the IF. Thus, the manuscript would need major revisions. A more detailed line by line review follows:

**Response:** We agree with this reviewer about the low-energy environment in which the IF formed, which is in agreement with our initial conclusions. However, it would appear that the switch from one redox state to the other was often accompanied by tectonic activity that caused deepening and shallowing.

Line 73: Rare Earth Elements should not be capitalized.

**Response:** Corrected

Line 115: Rhyolite is not intrusive.

**Response:** Corrected to extrusive

Line 226: It is much more common in work on iron formations to us PAAS to normalize the data.

**Response:** The rationale for using the NASC is as follows:

1. The NASC normalization maintains data consistency with the REE data published in our previous papers on the Milos IF ((1) Chi Fru, E., Ivarsson, M., Kilias, S.P., Bengtson, S., Belivanova, V., Marone, F., Fortin, D., Broman, C., and Stampanoni, M.: Fossilized iron bacteria reveal a pathway to the origin banded iron formations. Nat. Comm., 4, 2050 DOI: 10.1038/ncomms3050, 2013. (2) Chi Fru, E., Ivarsson, M., Kilias, S.P., Frings, P.J., Hemmingsson, C., Broman, C., Bengtson, S. and Chatzitheodoridis, E.: Biogenicity of an Early Quaternary iron formation, Milos Island, Greece. Geobiology, 13, 225–44, 2015.

2. There are no scientifically demonstrated discrepancies between the PAAS and NASC.

3. Following the above suggestion, data was normalized to PAAS for comparison with the NASC normalized trends. The results produced the same trend as observed when data are normalized to NASC. See new Figure 14 in the manuscript text, accessible in the attached supplement PDF file. Further explanations are also provided under sections 2.6-2.6.1 in the manuscript supplement text.

Line 327: Below storm wave base does not necessarily mean below 100 to 200 meters. At present storm wave penetration is deepest in locations such as southeastern Australia and Atlantic Canada where it reaches 120m. But these are very storm prone open ocean facing areas. It is difficult to give an estimate for paleo-storm wave base in the study area, but I doubt that it could be even close to 100m as more would mean waves with greater than 200 meter wavelengths in a sheltered area compared to the open Atlantic.

**Response:** We have omitted our attempt at specifying a value in the text for the depth. The new wording has been rephrased to:

*The MFIF rests directly on the submarine dacites-andesites that were deposited in a relatively shallow submarine environment (Stewart and McPhie, 2006).*

Line 340: Not enough evidence is given to justify the turbidite interpretation. Graded beds just mean they were deposited by powering-down events, which can occur in many different environments. Even if they are turbidites, which I have no idea whether they are or not from the evidence, the setting cannot be termed a fan, ie, why not a ramp or apron or a number of other environments that can have turbidites.

**Response:** Removed from text.

Line 344: Slump deposits infers an intact or partially intact block that slid. The conglomerates are not slump deposits. They could be debris flows, but again there is not enough evidence given to say this.

**Response:** Lines 337-361 have been deleted and replaced with this short paragraph:

*In the overlying sandstone-conglomerate facies, the presence of sedimentary structures indicative of wave action and currents (e.g. cross-stratification), that signify rapid deposition during a high energy event, are consistent with a switch to a shallow-submarine high energy environment (Stewart and McPhie, 2006; Chi Fru et al., 2015). This shift in depositional environments may have been controlled by a combination of submarine volcano-constructional processes, synvolcanic rifting and volcano-tectonic uplift known to have formed the CVSB (Papanikolaou et al., 1990; Stewart and McPhie, 2006).*

Line 347: If a flow is carrying pebbles it is not a low density turbidity current.

**Response:** Deleted and replaced with the paragraph above.

372: gravel to pebble is not proper terminology. Pebbles are gravel if unconsolidated.

**Response:** Lines 370-373 have been deleted and replaced with:

*The lower sandstone facies represents the host of the main economic grade Mn oxide ores in the CVSB. This constitutes part of a separate study devoted to the Mn ores and will not be dealt with further here.*

Line 395: mm-scale layers are not beds, they are laminae.

**Response:** Deleted and replaced with laminations.

Line 418: Why not below storm wave-base?

**Response:** Corrected.

Line 422: The only evidence for the interpretation that the conglomerates are " a series of channel deposits in an inner turbidite fan-like setting" appears to be that they are conglomerates. A great deal more evidence is necessary to be so specific about the depositional environment.

**Response:** Lines 420-434 have been deleted.

Line 424: No evidence has been given for a tidal environment and little evidence for a shoreface.

**Response:** Lines 420-434 have been deleted.

Line 427: There are also many papers that describe iron formations in other settings.

Response: Lines 420-434 have been deleted.

Line 434: This is an example of a reference that has little bearing on the preceding statement. The Mesoarchean Barberton is not a good analog for the sedimentary environment of the basin described here.

**Response:** Lines 420-434 have been deleted.

Line 439: The sedimentary structures described could have been formed in the environments proposed, but they are not limited to the environments given the lack of evidence.

**Response:** Lines 335-340 deleted.

Line 458: The description of these deposits has little in common with GIF. It is also better to reference the originator of the term GIF (Simonson), rather than Bekker, which is just a review article.

**Response:** References to GIF have been deleted from the text.

Line 481: This is circular reasoning.

**Response:** We agree and further demonstrate this in the new Figure 13D. These emphases must be highlighted to show some of the similarities these rare deposit shares with true BIFs.

Line 482: Precambrian BIF can be sulfide facies.

**Response: W**e agree. We are trying to make the statement that these are those type of BIF facies that are sulfide rich. We have therefore peplaced with the text:

*Lack of association of the framboidal-iron-rich particles with S, following SEM-EDS analysis, rules out a pyrite affiliation and is consistent with the non-sulfidic depositional model suggested by the sequential iron extraction redox proxy (Fig. 13D).*

Line 522: This statement is not correct. Planavsky and others (see authors' references to this statement) put forward that the anomaly for Ce must be less than .95 and greater than 1.05 to be significant, not less than or greater than 1.

**Response:** This has been corrected.

Line 528: They do not have similar enrichment levels; they are light depleted.
**Response:** Enrichment changed to depleted.

Line 574: The positive Eu anomalies are quite small compared to those associated with oceanic hydrothermal vent sediments. Also, volcanic detritus can carry positive Eu anomalies. A plot of Ti vrs Eu* would be useful to distinguish if the anomaly is related to volcanic detritus in the IF.

**Response:** We agree, but this effort will not tell us anything more than what we have already shown, since multiple evidence shows that we are dealing with materials being released into the basin intermittently by hydrothermal/volcanic activity as demonstrated by the ash particles in the bands. As we have shown in Figure 13 and from using multiple lines of evidence, the supply of materials from the continent to the basin was not an important source of sediments during the formation of the alternating Fe and Si layers. Our main interpretation is a hydrothermal source, backed by data in our cited publication in Nature Communications and Geobiology, in addition to the present submission.

Line 614: What is described as an upward fining trend appears to me to be simply one single graded bed. The fining upwards in the bed is better explained by the depositional mechanism losing energy through time. Also, conglomeratic beds usually represent rapid deposition during a high energy event, ie. storm or mass flow, rather than the slow pebble on pebble accumulation over years.

**Response:** This text has been revised as suggested. The new text reads like this:

*Geomorphological/chemical reconfiguration orchestrated the deposition of the NFIF in a deeper, small-restricted basin (Fig. 2). The deepening of Basin 3 is reflected in the underlying graded conglomerate bed that exhibits an upward fining trend, followed by transition into the fine-grained NFIF. The conglomerate bed may represent rapid deposition during a high-energy event, i.e. storm or mass flow, whereas the upwards fining in the bed is better explained by the depositional mechanism losing energy through time. These high-energy conditions apparently must have ceased during the deposition of the overlying NFIF, where we interpret that increased abundance of finely laminated IF and decreased evidence of storm and/or mass flow reworking reflects deepening conditions. The hypothesized deepening of Basin 3 is consistent with the interpretation that active rifting was an important mechanism in the formation of the CVSB (Papanikolaou et al., 1990).*

Line 682: Comparing the small Eu anomalies present in this study with the larger Precambrian anomalies should include giving the values for the average Precambrian anomalies. Simply stating the values of Eu anomalies of samples in this study are more similar to Archean anomalies is somewhat misleading.

**Response:** The paragraph has been deleted.

Line 757: If even small amounts of seawater are mixing with the hydrothermal fluid, as previously stated, anoxia could not exist.

**Response:** See new Figure 13C-D that firmly establishes the anoxic/ferruginous depositional conditions. Moreover the statement made by this reviewer that even if small amounts of seawater are mixing with hydrothermal fluid, anoxia cannot exist, is misleading and a bit perplexing because this argument means that redox gradients should not exist in nature. Following the rules of stoichiometry in chemical reactions, large volumes of highly reduced solutions such as hydrothermal fluids require equally large concentrations of oxidants (especially oxygen) to make the fluid oxidizing. From this reasoning, considerable amounts of oxygen are required to react with the large volumes of the highly reduced chemicals and compounds present in hydrothermal fluids. This argument is given as an explanation for why it took so long for oxygen to rise in the atmosphere (See Lyons et al., 2014. The rise of oxygen in Earth's early ocean and atmosphere. Nature 506:307–315: and the review by Bekker et al. 2010, cited in the manuscript). The reasoning is that reduced hydrothermal fluids that made up a bulk of the early oceans were eventually overwhelmed stoichiometrically by oxygen (meaning more oxygen was being produced than consumed by the reduced fluids), leading to the rise of oxygen in the atmosphere, c. 2.4 billion years. Even after that, although the Paleoproterozoic surface ocean was oxidized for close to two billion years, complete ocean oxygenation only came at the end of the Precambrian despite the fact that reduced deep ocean hydrothermal fluids continuously mixed with the oxygen-rich ocean surface seawater. If we were to follow the argument given by this reviewer, then the whole ocean would have been oxidized following the mixing of the reducing fluids with the thin layer of oxygen-rich seawater on the ocean surface. This indicates that the sedimentology, geochemistry and redox must be jointly interpreted to understand what occurred at Milos.

Line 842: The presence of a conglomeratic bed does not commonly mean deepening of a marine succession. There are literally thousands of papers where the upward transition of sandstones to conglomerates is interpreted as shallowing as energy levels increase with shallowing in a marine setting.

**Response:** We strongly agree that the paragraph was not well-phrased, leading to the difficulty in understanding the meaning of the sentence. It has now been revised to:

*All of this is feasible with the three-basin-fault-bounded hypothesis as a requirement for movement along fault lines in response to temporal tectonic activation. The upward sequence transition from the Mn-rich sandstone facies, through the pebbly conglomerate and the final termination in the overlying mud-grained NFIF (Fig. 8B), reflect sedimentary features formed during multiple changes in seawater levels (Cattaneo & Steel, 2000). This study proposes that the NFIF that overlies the transgressive-type conglomeratic lag along an erosional contact surface was likely deposited during maximum flooding, when the basin became stagnant and stratified, and subsequently was uplifted to emergence.*

Line 848: The presence of a transgressive conglomeratic lag implies that the area was emergent prior to this and the conglomerate formed by wave reworking in a shore proximal environment. Evidence has not been given to support this, and if I am not mistaken the conglomerate has previously in this manuscript been interpreted as a mass flow.

**Response:** Deleted.

Line 853: In these references the maximum regressive surface is overlain by a transgressive lag and then very shallow shoreline deposits affected by wave activity. A very different scenario to what these authors are proposing.

**Response:** Paragraph and references removed.

Line 855: The referenced BIFs are not deposited in sandstone/grainstone environments, the IFs are grainstone with very low siliciclastic contents and they are interlayered with chemical muds, but the IFs are not banded.

**Response:** Because this interpretation is not of immediate relevance to the strength of the paper, the paragraph has been deleted.

Line 1004: This process would be expected to produce a sharp bottom contact to the Fe-rich layer, which would then mineralogical grade upwards into the silica-rich layer. Is this the way the layers are organized?

**Response:** Yes. We show this in supplementary Figures 8 and 9.

[revised manuscript text omitted]

---

## Author Comment (AC2) · 12 Mar 2018

Reviewer Comments

A.J.B. Smith (Referee) bertuss@uj.ac.za

Dear Editor – Solid Earth This letter serves to summarize my review of the manuscript entitled "Sedimentary mechanisms of a modern banded iron formation on Milos Island, Greece", submitted by E Chi Fru et al. for possible publication in Solid Earth.

Reviewer: The manuscript documents Quaternary Fe-rich chemical sediments from the Cape Vani sedimentary basin (CVSB) on Milos Island, Greece. The Fe-rich units show a close associated with Mn-rich units and two subtypes were identified: i) microfossil rich iron formation (IF); and ii) non-fossiliferous IF. The IFs also occur over a

limited lateral extent. Geochemical and mineralogical data suggest that these units are very similar to Precambrian IFs and could potentially be proxies to the latter. Depositional conditions are also proposed by the authors, which include tectonics, biological activity, changing redox and abiotic Si precipitation. Although there is some overlap with their previous publications in Nature Communications (2013) and Geobiology (2015), the authors state that a major new addition is presenting plausible mechanisms for the temporal and spatial separation between Fe and Mn deposition in the CVSB.

It must be noted upfront that this manuscript covers a fascinating and important geological occurrence, namely a Quaternary, spatially limited IF. With the majority of IF deposited during the Precambrian and, more specifically, prior to 2 Ga, this occurrence shows how unique and isolated conditions can drastically influence localized geology. Further documentation and more field descriptions of this occurrence is therefore always welcome in the literature. The authors should also be commended on the level into which they attempt to present a depositional model for the units, something which is often not even attempted in manuscripts on IFs. The inclusions of the importance of faulting and tectonics is also a great addition into the model, something which often cannot be done in detail on older, Precambrian IF occurrences. The authors also take special care to address a multitude of possible paleoenvironmental conditions, looking at the past and comparing it to the present, and should be complimented on such an even-handed approach. The conclusion on the cause for the banding (main conclusion 5) being caused by episodic hydrothermal intensification is also an important one, and one that I believe is also supported by evidence from Precambrian IFs.

Response: We thank the reviewer (Dr A.J.B Smith) immensely for the enormous time and effort put in reviewing our manuscript. His attention to detail has transformed both the quality of the complex interpretations, better bringing out the remarkable similarities the enigmatic Milos IF share with the Precambrian BIFs and their implications for understanding the past. Below we address his critical comments point by point. Attached to this document is a PDF file named, supplement, containing the manuscript with the

changes requested by the reviews in red.

Point 1

Reviewer: There is inconsistent use of element names and symbols in the manuscript (e.g. line 233 uses "iron" and "Mn" in the same line). The authors should be consistent in their use of either names or symbols for elements. Also, the use of hyphens are also inconsistent and should carefully be revised and updated to ensure format and spelling consistency across the manuscript.

Response: We have amended the text accordingly.

Point 2

Reviewer: It is important that the authors include, even if briefly, the detail on how the age constraints of the CVSB were determined somewhere in the geological setting of the manuscript.

Response: Age constraints and accompanying references are now given in the introductory paragraph in section 1.1, under the geological setting.

Point 3

Reviewer: The results section on geochemistry (section 3.3; line 461 onwards) contains a lot of interspersed petrography and mineralogy, which I believe is inappropriate. I strongly recommend that the two sets of results be clearly separated and presented in their own result sections. This will also require some reshuffling and/or re-editing of figures to properly fit the order and flow of the revised sections.

Response: We agree that this might be confusing to the reader and have therefore divided section 3.3 into four independent subsections entitled:

3.3.1 Geochemistry of the individual Fe-rich and Si-rich bands 3.3.2 Mineralogy of the individual Fe-rich and Si-rich bands 3.3.3 Hydrothermal versus continental weathering 3.3.4 Redox reconstruction

Point 4

Reviewer: The use of North American Shale Composite (NASC) to normalize the REE data is strange and a bit dated. Most new publications on IFs from the mid-1990s onwards use Post-Archean Australian Shale (PAAS) for IF REE normalization. I would recommend the authors rather use this standard as it would make the data more comparable to other IF publications. In addition, the assessment of true Ce anomalies presented in this paper is also based on Bau and Dulski (1996), wherein they used PAAS as the shale standard for normalization. The Ce anomalies therefore need to be recalculated and replotted using PAAS. I do realize this might not make much of a difference, but for comparative purposes and for accuracy relating to the original publication this should be done.

Response: We agree that some recent papers have used PAAS (Post Archean Australian Shale-Taylor and Mclennan 1985) instead of NASC (North American Shale Composite-Gromet et al., 1984) while others use UCC (Multi Element Normalisation-Taylor and Mclennan 1985) and Chondrite-Normalised REE-Thomson 1982), which are all scientifically valid standards. In the case of our study, NASC was used for two main reasons: 1. The NASC normalization maintains data consistency with the REE data published in our previous papers on the Milos IF ((1) Chi Fru, E., Ivarsson, M., Kilias, S.P., Bengtson, S., Belivanova, V., Marone, F., Fortin, D., Broman, C., and Stampanoni, M.: Fossilized iron bacteria reveal a pathway to the origin banded iron formations. Nat. Comm., 4, 2050 DOI: 10.1038/ncomms3050, 2013. (2) Chi Fru, E., Ivarsson, M., Kilias, S.P., Frings, P.J., Hemmingsson, C., Broman, C., Bengtson, S. and Chatzitheodoridis, E.: Biogenicity of an Early Quaternary iron formation, Milos Island, Greece. Geobiology, 13, 225–44, 2015.

2. There are no scientifically demonstrated discrepancies between the PAAS and NASC.

3. Following the instructions given here, data was normalized to PAAS for comparison

with the NASC normalized trends. The results produced the same trend as observed when data are normalized to NASC. See new Figure 14 in the manuscript text, accessible in the attached supplement PDF file. Further explanations are also provided under sections 2.6-2.6.1 in the manuscript supplement text. Reviewer: To make the REE assessment in this paper even more complete, the authors should also plot YSN in the REE diagram between Dy and Ho (REY plots) as proposed by Bau and Dulski (1996) to get a more complete pattern and assessment of the REE trends.

Response: Unlike the paper by Bau and Dulski that was focused on REE analysis, the REE analysis performed in this study was conducted specifically to answer specific questions that we had posed, relevant to our study. These include:

1. What are the redox depositional conditions? In this revision, we have instead included a new Figure 13C-D, which provides independent support from iron extraction for the reducing depositional conditions displayed by the lack of Ce anomaly, which is critical to this paper than the REY plots. 2. What was the source of sediments to the basin? By using the Eu anomaly, coupled to the relationship between LREE and HREE, which produces a unique graphical shape for hydrothermal deposits, we could predict the hydrothermal/volcanic source of sediments, supported by the strong presence of volcanic ash in the Fe-rich bands. This information is further supported by other methods such as the chemical index of alternation (CIA), which shows negligible contribution of land-derived weathered sediments to the deposit. 3. In future work we will strive to give more thought to REY plots, especially if they can help resolve pertinent hypotheses for how the Milos IF formed.

Point 5

Reviewer: The conclusion that the IFs were deposited in reducing conditions in a redox stratified sea/ocean, as indicated in figure 16 and stated in lines 685 to 687, is not currently convincing for the following reasons: 1) The lack of a Ce anomaly is not a definite indicator of reducing conditions. This can be buffered by excess Fe(II) in the system

(see classic Eh-Ph diagrams by Brookins as well as Smith et al., 2013, Economic Geology, v. 108: 111-134). 2) The Ce anomaly calculations appear to have been done using NASC-normalized REE data instead of PAAS-normalized data. Proposed mechanisms for deeper water anoxia is left for very late in the manuscript, hinging on modern analogues (section 4.2.5), making the lead up at times unconvincing. Some of these arguments are also based on the interpretation of geochemical evidence to suggest anoxia, when micro-oxic conditions, in my opinion, cannot be completely ruled out. A more even handed approach regarding the Eh conditions throughout the manuscript, and taking into account the redox buffering effect between $Fe^{2+}$ and $Mn^{2+}$, is likely required throughout. Micro-aerophilic bacteria also likely played a larger role than currently suggested by the authors when considering how recently deposition occurred and how deep the water could have been (well below wave base). I do believe the authors have generally done a good job regarding Eh conditions in their depositional model and that all the necessary information is throughout the manuscript, but in my opinion they are only 70% there in making it convincing and even handed.

Response: We appreciate the detailed insights and agree that although widely used, Ce anomalies have to be interpreted with caution. For this reason, we have analysed the same set of samples by the sequential iron extraction approach; a widely applied proxy for reconstructing Paleo-redox. These data presented in the new Figure 13C-D and supported by the references below and several others in the public literature, confirm the inferred anoxic depositional conditions indicate by the Ce anomalies. Comparative normaliztion with either the PAAS or NASC standards, produced the same outcome, in agreement with the interchageable use of both standards. More information on the analysis can be found under sections 2.6-2.6.1 and in Figure 14.

1. Poulton, S.W., and Canfield, D.E.: Development of a sequential iron extraction procedure for iron: implications for iron partitioning in continentally derived particles. Chem. Geol. 2014, 209–221, 2005. 2. Poulton, S.W. and Canfield, D.E.: Ferruginous conditions: A dominant feature of the ocean through Earth's history. Elements. 7, 107–

112, 2011. Point 6 Reviewer: The formation of the granular Fe-rich beds (lines 456-459) is not satisfactorily addressed. There has to be morphological evidence within the granules for the authors to be able to commit to either a sedimentary or supergene formational mechanism.

Response: This statement resonates with the opinion provided the anonymous reviewer. We have therefore deleted the paragraph regarding GIF.

Point 7

Reviewer: To me, there is some confusion regarding deep ocean anoxia relative to hydrothermal venting in the depositional model (lines 995-999). This needs to be clarified in the discussion and lead up. Was deep water anoxia semi-permanent or did the venting play a role in establishing it? Here it appears that anoxia was semi-permanent, but the motivation needs to be better conveyed in the lead-up.

Response: The new Figure 13C-D, which reports on redox reconstruction by sequential iron extraction, has been discussed extensively in the manuscript. This proxy is perhaps the most widely accepted tool for reconstructing paleoredox depositional conditions (i.e., oxic, anoxic but ferruginous and anoxic but euxinic environments. See references given under point 5 above. The proxy works on the basis that it is capable of delineating the redox conditions in the water column beneath which sediments form. We have discussed a number of procedures in the paper by which this anoxia could have developed in the CVSB. The understanding is that there is a combination of the basin being cut off, either in a crater environment like in the adjacent Kolombo volcano and a combination of hydrothermal activity and $CO_2$ accumulation, creating permanent bottom water anoxia. We have explored all these different pathways in the manuscript. However, what has become more certain with the sequential iron extraction data is that deposition occurred under severe anoxic conditions, which could have been the results of numerous processes. These processes will gradually become evident as our work on this formation expands and as other researchers become genuinely interested in

sampling and exploring its formation mechanisms.

Geochemical REE evidence has shown that Fe was sourced from hydrothermal fluids, and deposition took place beneath anoxic waters (see above). This combined with geological and mineralogical evidence for example the presence of tridymite etc, in Fe-rich layers of the NFIF), indicate that oxidation of Fe(II) in the NFIF corresponded closely in time with major Basin 3-scale intense, possibly episodic, submarine volcanism and hydrothermal activity. We suggest that submarine volcanism/hydrothermalism were responsible for generating a dynamic Basin 3-scale chemocline separating anoxic/suboxic ferruginous deep waters from oxic shallow waters. This is explained by the notion that seawater redox state in a basin with restricted circulation and intense submarine volcanism/hydrothermalism like Basin 3, may be lowered by an enhanced flux of hydrothermally derived reductants, like reduced Fe and Mn, H2 and even CO2-induced stratification, as discussed extensively for various hydrothermal vent fields in the Hellenic Volcanic Arc. These processes would have overpowered the oxidizing potential of seawater (Bekker et al., 2014).

The oxidation of Fe(II) at and below the chemocline, by microaerophilic chemolithoautotrophs and strict anaerobic photoautotrophic Fe(II) oxidation are favored as potential modes of Fe(III)(oxyhydr)oxide precipitation in the NFIF. However, at the current time, such evidence is lacking for the NFIF. We therefore choose not to delve into speculation. One approach had been to use fossil lipid biomarkers, but consistent with the poor organic content of iron formations, this approach proved not to be very successful. Moreover, unlike the MFIF, the NFIF is microfossil-poor, leaving the question of microbial contribution to the deposition of the NFIF wide-opened. On-going stable Fe isotope analysis may help solve this problem. However, Fe isotopes may not be able to differentiate biological activity from abiological processes, because in some cases both can fractionate Fe equally. Nonetheless, our paper provides an intriguing scenario where Precambrian type-rocks are formed in firmly reconstructed anoxic bottom waters under the modern atmosphere. This is unprecedented. We use a multitude of

techniques from REEs, carbon isotopes, Raman analysis, TEM, lipid biomarkers, sequential iron extraction, etc, at a comprehensive scale that is hardly ever seen in one paper, to arrive our conclusions.

Point 8

Reviewer: In the final conclusion (lines 1013-1017) the authors state that "Whether the rocks described here are analogues of Precambrian BIFs or not, and whether the proposed formation mechanisms match those that formed the ancient rocks, is opened to debate." The work here have many similarities to proposed Precambrian BIF depositional models (e.g. Smith et al., 2013; Bekker et al., 2010 and depositional models by Klein and Beukes, Beukes and Gutzmer). The authors should comment on this briefly.

Response: Your comment has been included in the conclusion.

References 1. Bekker, A., Planavsky, N., Rasmussen, B., Krapez, B., Hofmann, A., Slack, J., Rouxel, O. and Konhauser, K., 2014. Iron formations: Their origins and implications for ancient seawater chemistry. In Treatise on geochemistry (Vol. 12, pp. 561-628). Elsevier. 2. Brookins, D.G., 1988, Eh-pH diagrams for geochemistry: Berlin, Springer-Verlag, 176 p. 3. Brookins, D.G., 1989, Aqueous geochemistry of rare-earth elements: Reviews in Mineralogy, v. 21, p. 201–225. 4. Konhauser, K., 2007, Introduction to geomicrobiology: Malden, Blackwell, 425 p. 5. Konhauser, K.O., Hamade, T., Raiswell, R., Morris, R.C., Ferris, F.G., Southam, G., and Canfield, D.E., 2002, Could bacteria have formed the Precambrian banded iron formations?: Geology, v. 30, p. 1079–1082. 6. McCollom, T.M., and Shock, E.L., 1997, Geochemical constraints on chemolithoautotrophic metabolism by microorganisms in seafloor hydrothermal systems: Geochimica et Cosmochimica Acta, v. 61, p. 4375–4391. 7. Smith, A.J., Beukes, N.J. and Gutzmer, J., 2013. The composition and depositional environments of Mesoarchean iron formations of the West Rand Group of the Witwatersrand Supergroup, South Africa. Economic Geology, 108(1), pp.111-134. 8. Taylor SR, McLennan SM (1985) The continental crust: its composition and evolution. Blackwell Scientific Publication, Carlton, 312 p. 9. Gromet PL et al., 1984. The "North American shale composite": Its compilation, major and trace element characteristics. Geochim. Cosmo. Acta 48:2469-2482. 10. Thompson, R.N. (1982) Magmatism of the British Tertiary volcanic province. Scott. J. Geol. 18, 49-107.

Specific Comments:

Abstract Reviewer: The sentence running from lines 37 to 38 should be rephrased. As it is currently written it does not clearly convey the stratigraphic relationship between the Fe- and Mn-rich units. Is the transition to the Mn-rich formation upwards or downwards? I think rephrasing this as two sentences briefly providing the bottom-up stratigraphy would clear this up.

Response: Rephrased as suggested.

Reviewer: The statements in lines 38 to 41 relating to anoxia might not be completely accurate. Refer to point 6 above.

Response: The statement relating to anoxia is strongly supported by new evidence provided in Figure 13C, in agreement with the lack of Ce anomaly, as discussed above. Also see the extensive discussion on how such anoxic conditions might develop in the CVSB, based on, on-going modern processes that form anoxia along the Hellenic Volcanic Arc. Please kindly check the cited references as they contain valuable information.

Reviewer: The summary of depositional conditions in lines 46 to 48 is too vague and brief. The authors need to take a few more lines and properly summarize their proposed depositional model.

Response: This sentence is given in two parts. The first sentence summarizes the basic Si mineralogy of the two deposits, while the concluding sentence sums up the findings in this study, which are indeed too complex (involving a number of processes) to be laid out completely in the abstract. To attempt to explain each process and how

they are linked together is beyond the scope of the abstract, given their complexity. The abstract has been reformatted to read as: An Early Quaternary shallow submarine hydrothermal iron formation (IF) in the Cape Vani sedimentary basin (CVSB) on Milos Island, Greece, displays banded rhythmicity similar to Precambrian banded iron formation (BIF). Sedimentary and stratigraphic reconstruction, coupled to biogeochemical analysis and micro-nanoscale mineralogical characterization, confirm the Milos IF as a modern BIF analogue. Spatial coverage of the BIF-type rocks in relation to the economic grade Mn ore that brought prominence to the CVSB implicates tectonic activity and changing redox in their deposition. Field-wide stratigraphic and biogeochemical reconstruction demonstrate two temporal and spatially isolated iron deposits in the CVSB with distinct sedimentological character. Petrographic screening suggests the previously described photoferrotrophic-like microfossil-rich IF (MFIF), accumulated on basement andesite in a ~150 m wide basin, in the SW margin of the basin. A strongly banded non-fossiliferous IF (NFIF) sits on top the Mn-rich sandstones at the transition to the renowned Mn-rich formation, capping the NFIF unit. Geochemical evidence relates the origin of the NFIF to periodic submarine volcanism and water column oxidation of released Fe(II) in conditions apparently predominated by anoxia, similar to the MFIF. Raman spectroscopy pairs hematite-rich grains in the NFIF with relics of a carbonaceous material carrying an average $\delta$13Corg signature of ~-25‰. However, a similar $\delta$13Corg signature in the MFIF is not directly coupled to hematite by mineralogy. The NFIF, which post dates large-scale Mn deposition in the CVSB, is composed primarily of amorphous Si (opal-SiO2·nH2O) while crystalline quartz (SiO2) predominates the MFIF. An intricate interaction between tectonic processes, changing redox, biological activity and abiotic Si precipitation are proposed to have collectively formed the unmetamorphosed BIF-type deposits in a shallow submarine volcanic center.

Introduction Reviewer: The introduction and geological setting is brief, concise and appropriate. Here are some comments/corrections:

Reviewer: The author should start the introduction with a one sentence definition of IFs,

mentioning their normal age distribution, and moving the references in line 63 and 64 to the end of the definition sentence. In the current form, the references in the middle of the first sentence clutters it and takes away from the impact of this sentence. If it reads better, the authors can also place the IF definition after the current first sentence

Response: Sentence restructured as suggested.

Reviewer: For line 84, a good reference to add would be Beukes et al., 2016, Episodes v. 39: 285-317. It reviews all the Mn deposits of Africa.

Response: Added as suggested.

Reviewer: More information needs to be provided on how the age range of the CVSB was determined.

Response: The text has been updated to: K-Ar radiometric dating of biotite and amphiboles belonging to the dacitic/andesitic lava domes flooring the CVSB basin gave an Upper Pliocene age of 2.38±0.1 Ma (Fytikas et al., 1986; Stewart and McPhie, 2006). The fossiliferous sandstones/sandy tuffs hosting the Mn-rich deposit, which contain the gastropod mollusk guide fossil, Haustator biplicatus sp. (Bronn, 1831), indicate an Upper Pliocene-Lower Pleistocene age.

Reviewer: Some minor comments and corrections are noted in the pdf copy. Make sure to address these too.

Response. We have made all the corrections suggested in the annotated PDF. The responses are highlighted red in the new text appended in the supplementary document.

Methodology

Reviewer: It should be stated very early on in the methodology section how many samples were taken, what sample types (lithologies) were taken and the approximate localities of the samples

Response: The cardinal points from where samples were collected are given in the

paper, such that any individual can follow this direction to the exact location of sampling. This study shows the sawn rocks and discusses how each representative layer was analysed by a variety of methods. The focus of this study is on the fine textures of these rocks, combined with field survey over the entire 1 Km long basin.

Reviewer: The sample preparation subsection (lines 119-125) is insufficient. More detail needs to be provided on how samples were taken, and how they were prepared for the different analyses. The type of mill used is not even mentioned. To refer to previous papers in the manner done here for sample preparation is not sufficient.

Response: Samples were sawn to remove weathered surfaces, and chips of rocks were sent to GeoTech Labs, which is a certified commercial laboratory for preparing thin sections. We are not sure what can be said about sawing a rock to remove weathered surfaces, which is why that part of the sentence has been left unchanged. Usually, it is enough to provide information on commercial providers, since they are accredited and certified. The rock polishing is a service provided by GeoTech Labs and can be requested at a fix rate, the details of which are not needed for this paper. Thirdly, the manuscript is over 15000 words. It is correct scientific practice to reference methods that can be obtained from previous publications. This is the practice advised by some of the most influential scientific journals. We have beefed parts of the acid digestion and removed parts as requested. However, the mechanism by which the rocks were pulverized is inconsequential because any mechanism would work as long as it enables the final dissolution of the powder into a liquid phase whose chemical analysis can be analyzed which is where focus is put. XRD analysis was performed directly on the pulverized samples as discussed in the manuscript.

Reviewer: Lines 120-121: I am not sure how this fits under the subheading of sample preparation. This will need to be moved and better explained. So yes, it does form the basis for facies analysis, but what was actually done and how?

Response. Line120-Line 121 has been removed as advised to Section 3.1, as an introductory sentence under Lithostratigraphy, where the methodology for facies analysis is described in detail.

Reviewer: Raman spectroscopy (lines 132-138 should not be included under the XRD subsection heading (section 2.2.1).

Response: Corrected.

Reviewer: Line 172: The abbreviations ICP-ES/MS and XRD have not been defined. Also, AcmeLabs has had its name changed to Bureau Veritas. Where is the lab situated?

Response: Corrected.

Reviewer: Lines 226 to 227: To use North American Shale Composite (NASC) for normalization of BIF data is an older way of doing it. Most of the newer publications use Post-Archean Australian Shale (PAAS). The authors should consider rather normalizing to PAAS for better comparison to recent literature.

Response: We have provided justification above

Results Reviewer: Lines 266 to 268 contains repetition from what was stated in line 264. The authors should combine and clean this up to remove the repetition.

Response: Lines 266 to 268 have been deleted.

Reviewer: Line 292: The term "Sh beds" has not been previously defined. What does it mean?

Response: It has been removed and named accordingly as explained in the next comment. This abbreviation, like the one below, is an editing oversight.

Line 298: The term "Gcm" has not been previously defined. What does it mean?
Response: All abbreviations relating to lithological successions have been removed and fully spelt out throughout the text. "Sh" has been replaced with "plane-parallellaminated sandstone/sandy tuff", "Gcm" has been replaced with "clast-supported pebble-to-cobble conglomerate".

Reviewer: Lines 398-400: The bracket started in line 398 is never properly closed. Please do so.

Response: Corrected

Reviewer: Lines 431-434: The phrase "from a relatively shallow and deeper water setting: : : to a relatively deeper quiet water environment" appears to be contradictory at the start. Some rephrasing is required to improve clarity.

Response: The paragraph has been shortened to: The hypothesized deepening of Basin 3 is consistent with the interpretation that active rifting was an important mechanism in the formation of the CVSB (Papanikolaou et al., 1990).

Reviewer: Lines 456-459: The granules associated with supergene formation versus those associated with sedimentary reworking will have very different internal structures. Whole rock geochemistry will also be very different. Why is such data not available? These are two very different formational proposals that should be resolved.

Response: We have not completed the mineralogical and geochemical description of these rocks. Therefore at the moment the sedimentary reworking vs. supergene origin cannot be deciphered. However, resolving this problem is beyond the scope of the present paper, and it is the subject of a future communication. Therefore, we have removed the reference to GIF, leaving the supergene possibility, because to our best judgment this better fits the geological and macroscopic characteristics of these ironstones.

Reviewer: Lines 467-469: The way this sentence is phrased does not read accurately. Why do dramatic fluctuations control the Fe to Si ratio? Seems to be overly obvious and at the same time not supported by evidence, as if the lines between correlation and causality are being blurred. I would strongly recommend that this statement gets

rephrased to more accurately represent what the data is actually showing, namely an inverse correlation between Fe and Si, which is to be expected when the two components are the only two major ones in the rock!

Response: Revised as suggested to read:

The laser ablation ICP-MS data further show that dramatic fluctuations in Fe concentrations control the Si to Fe ratio in both types of rocks, despite the thousands of millions of years gap between them. This inverse correlation between Fe and Si is expected because they are the two major elemental components of the rock.

Reviewer: Lines 520-523: Here the normalization to NASC becomes problematic. The paper that originally presented to calculations and plot to assess true Ce anomalies (Bau and Dulski, 1996), used PAAS as their shale standard. All the calculations in that paper therefore used PAAS-normalized values and NASC. This alone likely justified why the authors should redo the REE data normalized to PAAS

Response: We found no difference by normalizing the data either with NASC or PAAS. However, we maintained the NASC data to maintain consistency with recent publications that have used the NASC standard on the Milos IF. See sections 2.6-2.6.1 in the manuscript. Also see Figure 14.

Reviewer: Lines 522-523: Depending on the journal's citation format, shouldn't "Bau et al." be "Bau and Dulski"?

Response: Corrected.

Discussion Reviewer: Lines 604-606: I do not believe the conclusion here is completely accurate. One can only state the MFIF deposition preceded the second-stage Mn mineralization. There is no clear evidence that the MFIF and first-stage Mn mineralization was coeval.

Reponse: Deleted

Reviewer: Lines 614-615: Please rephrase this so that the clarity is improved. Lines 615-616: "This uplifting into shallower water event": Which event is this? The discussion preceding this statement in this paragraph has only been referring to a transgressive (i.e. deepening) event. Please rewrite and restructure where necessary to make this paragraph read better and make more sense.

Reponse: This paragraph from line 614-line 619 has been modified to: The deepening of Basin 3 is reflected in the underlying graded conglomerate bed that exhibits an upward fining trend, and transitions into the NFIF. The conglomerate bed may represent rapid deposition during a high-energy event, i.e. storm or mass flow, whereas the fining upwards in the bed is better explained by the depositional mechanism losing energy through time. These high-energy conditions apparently must have ceased during the deposition of the overlying NFIF, where we interpret that increased abundance of finely laminated IF and decreased evidence of storm and/or mass flow reworking reflects deepening conditions. The hypothesized deepening of Basin 3 is consistent with the interpretation that active rifting was occurring during CVSB evolution (Papanikolaou et al., 1990).

Reviewer: Lines 651-655: Break this sentence into two shorter ones and do some restructuring that it reads better and the content is clearer.

Response: The sentence split into two as suggested. However, samples were sawn to remove exposed layers and only the laminated bands for the NFIF were analyzed. Modern sediments from Spathi bay, located Southeast of Milos Island where hydrothermal activity is presently ensuing at 12.5 m below sea level, revealed similar plant lipids as recorded in the Quaternary IF (Fig. 15G).

Reviewer: Lines 663-672: This starting sentence for this paragraph reads like it comes out of nowhere. The authors need to lead into the content of this paragraph much better. Also, I am not convinced that this paragraph belongs in the subsection, as it does not directly relate to mineral paragenesis and also reads like it comes out of

nowhere

Response: This paragraph has been deleted.

Reviewer: Lines 685-687: This statement has a few potential problems. Firstly, the lack of a Ce anomaly is not a definite indicator of reducing conditions. This can be buffered by excess $Fe2+$ in the system. Secondly, the Ce anomaly calculations appear to have been done using NASC-normalized REE data instead of PAAS-normalized data. At the very least the latter issue has to be resolved, and thereafter a more convincing argument need to be provided for reducing conditions.

Response: We have confirmed the redox conditions using another much accepted method as stated above. And as already stated and shown above, whether NASC or PAAS was used, the outcome was the same.

Reviewer: Lines 697-699: This one line is not convincing enough as a mechanism for redox stratified depositional environment. I admit that better and more convincing mechanisms are discussed later in the manuscript, but here the line comes over as unconvincing. Maybe note that possible mechanism for redox stratification are discussed in more details later.

Response: We have provided more supportive evidence for seafloor anoxia during deposition of both the NFIF and the MFIF using amore reliable redox proxy. Given that bottom water anoxia would have existed beneath a permanently oxygenated atmosphere, surficial waters column would have been oxidized.

Reviewer: Lines 722-729: I am not following the argument in point 3 very well. It requires some rephrasing and rewriting to convey the argument more clearly and concisely.

Response: This response is particularly directed at sedimentologists who according to our experience have attacked our conclusions by proposing that anoxic hydrothermal fluids may have penetrated preformed sediments to form the IF-rich bands as a

diagenetic product. However this is impossible, given that reduced hydrothermal fluids in anoxic sediments deprived of light and oxygen, would lack oxidizing power to precipitate iron oxides. The new text reads as:

The reducing depositional conditions do not support sediment diagenesis as an alternative model for explaining the origin of the Milos IF. This is because the oxidation of ferrous Fe supplied in reduced hydrothermal fluids, must interact with a sizeable pool of oxygen, enabling microaerophilic bacteria oxidation of ferrous iron to Fe(III)(oxyhydr)oxides (Johnson et al., 2008). Otherwise, light-controlled photoferrotrophy—an extremely rare sediment characteristic—precipitates Fe oxides in the absence of oxygen in sunlight environments Weber et al., 2006).

Lines 722: The authors need to re-evaluate all the statements related to Ce anomalies after recalculating the anomalies to PAAS

Response: See above for justification with links to revision in the text and Figure 14.

Reviewer: Lines 730-738: For point 4 as well, the argument is not coming through clearly. There also appears to be some structuring and grammatical problems in this paragraph. From what I can follow the argument is probably sound, but I cannot be sure as the paragraph is not well written

Response: This is equally a comment that has been raised by sedimentologists. But we believe that the paper is clear enough and have deleted this point.

Reviewer: Lines 806-809: Many authors agree with the statement that the lack of organic carbon is not due to metamorphism, but for a different reason. The organic matter is also likely destroyed in Precambrian BIF in a redox reaction with Fe3+, leading to the formation of 13C-depleted siderite and ankerite. This should be briefly addressed. See, for example, Smith et al. (2013, Economic Geology, v. 108:111-134) and references therein.

Response: The text has been updated to include this information:

Importantly, prokaryotic biomarkers are suggested to poorly preserve in these young BIF analogues. This raises the possibility that this may provide an important explanation for why lipid biomarkers are yet to be extracted from Precambrian BIFs. Moreover, the data are compatible with the low Corg recorded in BIFs of all ages, suggesting that the low Corg abundance may not be due to metamorphism as often proposed (Bekker et al., 2010) or to Corg oxidation by dissimilatory iron reducing bacteria to form 13C-depleted siderite and ankerite during diagenesis (Johnson et al., 2008; Bekker et al., 2010). The Milos BIF-type rocks are unmetamorphosed and lack iron carbonate, yet have vanishingly low Corg levels similar to the ancient metamorphosed BIFs. However, an alternative possibility is that the iron oxides may have been reduced through biological oxidation of organic carbon, but carbonate saturation was not reached (Smith et al., 2013).

Reviewer: Lines 832-838: The tectonic and sea level mechanisms for changing redox conditions seem plausible. However, it only truly works when the motivation and mechanism for anoxia during IF deposition are more convincing. See main comment nr 6 above

Response: We have provided a strong evidence for anoxia by the sequential iron proxy.

Reviewer: Lines 851-853: For interest, also see Smith et al. (2013, Economic Geology, v. 108: 111-134).

Response: According to the anonymous reviewer comments, we felt that it is better to delete lines 851-853.

Reviewer: Lines 857-858: Rephrase and fix the grammar in this sentence.

Response: corrected

Reviewer: Line 876-877: The statement "most likely dependent on prevailing redox conditions" is not yet convincing! The accumulation of Mn could also be buffered by the availability of Fe2+

Response: New evidence has been provided that strongly support reducing depositional conditions of both the NFIF and MFIF.

Conclusions

Reviewer: Lines 990-994: Agreed that this is a feasible mechanism. However, Prevalence of available Fe2+ as a redox buffer should also be considered and addressed.

Response: Redox depositional conditions are firmly established.

Reviewer: Lines 995-999: This needs to be clarified in the discussion and lead up. Was deep water anoxia semi permanent or did the venting play a role in establishing it? Here it appears that anoxia was semi-permanent, but the motivation needs to be better conveyed in the lead-up.

Response: We have provided plausible mechanisms that can explain the anoxia recorded in this basin in extensive review of redox processes along the entire volcanic arc. However the change from Mn deposition to the NFIF certainly indicates that redox was changing intermittently. This subject has been thoroughly discussed throughout the manuscript and to avoid repetition we have not brought it up again here.

Lines 998-999: What about the chemolithoautotrophs (microaerophilic bacteria)? I am not convinced that one can commit to only the photoferrotrophs with the dataset presented here

Response: Lines 995-999:

We agree and have within the entire text discussed this point, but we have been careful since all endeavours have fallen short to identify their presence in Milos. Any extensive discussion on this topic is highly speculative as stated above. Inasmuch as we want to have this discussion, we feel that it is reasonable to limit it to the available data at hand. However, the similar depositional conditions as MFIF, give us the opportunity to discuss more about photoferrotrophy, as there is some evidence pointing to this. We have searched for characteristic microaerophilic fossils such as the twisted stalks of

Mariprofundus ferrooxydans but found no evidence. This sentence has been changed to read:

The mechanism of formation of the MFIF and NFIF therefore most likely involved exhalative release of reduced hydrothermal/volcanic fluids into a restricted and deoxygenated seafloor water column where the oxidation of reduced Fe to Fe(III)(oxyhydr)oxides occurred, most likely by the activity of photoferrotrophs (Chi Fru et al., 2013). Microaerophilic oxidation of Fe(II) was likely critical, but that remains to be shown.

Please also note the supplement to this comment:
https://www.solid-earth-discuss.net/se-2017-113/se-2017-113-AC2-supplement.pdf

---

## Editor Decision (ED1)

[revised manuscript text omitted]

---

## Author Response (AR2)

Ernest Chi Fru
Senior Lecturer in Geomicrobiology
Earth and Ocean Sciences
Cardiff University
CF10 4AT Cardiff, UK
Email: ChiFruE@cardiff.ac.uk
02/04/2018

Dear Dr Samankassou,

Thank you immensely for editing our paper, *Sedimentary mechanisms of a modern banded iron formation on Milos Island, Greece*, which is currently in revision for final publication in Solid Earth.
I have gone through the manuscript to effect the minor changes you recommended. I am therefore delighted to resubmit the revised version of the manuscript for your consideration, hoping that you will find the changes adequate to consider our paper suitable for final publication in Solid Earth.

Thank you very much for your time.

Kind regards,
Ernest

**Sedimentary mechanisms of a modern banded iron formation on Milos Island, Greece**

[1,2]Ernest Chi Fru*, [3]Stephanos Kilias, [4,5]Magnus Ivarsson, [1]Jayne E. Rattray,

[3]Katerina Gkika, [2]Iain McDonald, [6]Qian He, [1]Curt Broman

[1]Department of Geological Sciences, 10691, Stockholm University, Sweden.

[2]School of Earth and Ocean Sciences, Cardiff University, Park Place, CF10  3AT,

Cardiff, UK.

[3]Department of Economic Geology and Geochemistry, Faculty of Geology and

Geoenvironment, National and Kapodistrian University of Athens, Panepistimiopolis,

Zographou, 15784, Athens, Greece.

[4]Department of Biology, University of Southern Denmark, Campusvej 55, Odense M,

DK5230, Denmark

[5]Department of Palaeobiology, Swedish Museum of Natural History, Box 50007,

Stockholm, Sweden.

[6]School of Chemistry, Cardiff University, Park Place, CF10 3AT, Cardiff, UK.

*Corresponding author

Tel: +44(0) 29 208 70058

Email: ChiFruE@cardiff.ac.uk

Short title: A modern banded iron formation

**Abstract.** An Early Quaternary shallow submarine hydrothermal iron formation (IF) in the Cape Vani sedimentary basin (CVSB) on Milos Island, Greece, displays banded rhythmicity similar to Precambrian banded iron formation (BIF). Field-wide stratigraphic and biogeochemical reconstruction show two temporal and spatially isolated iron deposits in the CVSB with distinct sedimentological character. Petrographic screening suggests the photoferrotrophic-like microfossil-rich IF (MFIF), accumulated on a basement consisting of andesites, in a ~150 m wide basin, in the SW margin of the basin. A banded non-fossiliferous IF (NFIF) sits on top of the Mn-rich sandstones at the transition to the renowned Mn-rich formation, capping the NFIF unit. Geochemical data relates the origin of the NFIF to periodic submarine volcanism and water column oxidation of released Fe(II) in conditions predominated by anoxia, similar to the MFIF. Raman spectroscopy pairs hematite-rich grains in the NFIF with relics of a carbonaceous material carrying an average $\delta^{13}C_{org}$ signature of ~-25‰. A similar $\delta^{13}C_{org}$ signature in the MFIF could not be directly coupled to hematite by mineralogy. The NFIF, which post dates large-scale Mn deposition in the CVSB, is composed primarily of amorphous Si (opal-$SiO_2 \cdot nH_2O$) while crystalline quartz ($SiO_2$) predominates the MFIF. An intricate interaction between tectonic processes, changing redox, biological activity and abiotic Si precipitation are proposed to have collectively formed the unmetamorphosed BIF-type deposits. Despite the differences in Precambrian ocean-atmosphere chemistry and the present geologic time, these formation mechanisms coincide with those believed to have formed Algoma-type BIFs proximal to active seafloor volcanic centers.

**Keywords:** Banded iron formation; BIF analog; Hydrothermal activity; Iron cycling; Silica cycling.

**1 Introduction**

Banded iron formations (BIFs) are marine sedimentary deposits formed predominantly during the Precambrian, containing at least 15% bulk Fe content and

Fe-rich bands alternating with Si-rich layers (James, 1954; Gross, 1980; Simonson,

[revised manuscript text omitted]

Unknown

Fig. 14. Rare earth element (REE) distribution in samples and calculated Ce and Eu
anomalies for NFIF bands and MFIF. (A), NASC normalized REE distribution in
various rock facies. (B), Ce anomalies. (C), Eu anomalies and light REE (LREE) vs.
heavy REE (HREE) ratio in the NFIF bands and MFIF. Similar trends were
reproduced for Post Archean Australian Shale (PAAS) normalized REE (McLennan,
1989; Bau and Dulski, 1986), exemplified by the inset in B.

[Figure]

Fig. 15. GC/MS chromatogram sections of total lipid extracts of the BIF-type rocks (A-F). Data are for individual bands excised from the sawn rock in Figure 7E. Panel G illustrates total lipid extract for the modern shallow submarine hydrothermal sediments at Spathi Bay, south east on the coast of Milos Island. Peak values indicate the lipid-specific $\delta^{13}$C values per mil. Because of the low intensity of the lipids recovered, it was not possible to obtain $\delta^{13}$C values specific for all peaks. Peaks are annotated as; FAME = fatty acid methyl ester; Me = methyl group; TMS = trimethylsilyl; TMSE = trimethylsilyl ester. (1) $C_{14:0}$ FAME, (1a) $C_{14:0}$ 13Me FAME, (2) $C_{15:0}$ FAME, (3) $C_{16:0}$ FAME, (3a) $C_{16:9}$ FAME, (3b) C16:0 TMS, (3c) 10Me $C_{16:0}$ FAME, (3d) $C_{16:9}$ FAME, (3e) $C_{16:0}$ TMSE, (4) $C_{17:0}$ TMS, (5) $C_{18:0}$ FAME, (5a) $C_{18:9}$

FAME, (5b) $C_{18:0}$ TMS, (5c) $C_{18:0}$ TMSE, (6) $C_{19:0}$ FAME, (6a) $C_{19:0}$ 18Me TMS, (7)
$C_{21:0}$ TMS, (8) $C_{22:0}$ TMS, (9) Cholesterol TMS, (10) Stigmasterol TMS, (11) beta-
Sitosterol (*) contaminants (e.g., phthalates).

[Figure]

Fig. 16. Conceptual model of the mechanism of band formation of the NFIF, related
to changes in the intensity of hydrothermal activity and chemical oxidation of Fe(II)
to Fe(III) in the water column, inferred from the data. See Chi Fru et al. (2013) for a
biological model for the formation of the MFIF.